# Mode-seeking for inverse problems with diffusion models

## Abstract

A pre-trained unconditional diffusion model, combined with posterior sampling or maximum a posteriori (MAP) estimation techniques, can solve arbitrary inverse problems without task-specific training or fine-tuning. However, existing posterior sampling and MAP estimation methods often rely on modeling approximations and can be computationally demanding. In this work, we propose the variational mode-seeking loss (VML), which, when minimized during each reverse diffusion step, guides the generated sample towards the MAP estimate. VML arises from a novel perspective of minimizing the Kullback-Leibler (KL) divergence between the diffusion posterior $p(\mathbf{x}_0|\mathbf{x}_t)$ and the measurement posterior $p(\mathbf{x}_0|\mathbf{y})$, where $\mathbf{y}$ denotes the measurement. Importantly, for linear inverse problems, VML can be analytically derived and need not be approximated. Based on further theoretical insights, we propose VML-MAP, an empirically effective algorithm for solving inverse problems, and validate its efficacy over existing methods in both performance and computational time, through extensive experiments on diverse image-restoration tasks across multiple datasets.

## 1 Introduction

Solving an inverse problem essentially involves estimation of the original data sample $\mathbf{x}$ based on a given partially degraded measurement $\mathbf{y}$. Formally, Equation (1) relates the degraded measurement with the original data sample, where $\mathcal{A}$ denotes the degradation operator and $\eta$ is a random variable denoting measurement noise, which is typically assumed to be Gaussian distributed with known standard deviation $\sigma_{\mathbf{y}}$, i.e., $\eta \sim \mathcal{N}(0, \sigma_{\mathbf{y}}^2 \mathbf{I})$. This implies that $p(\mathbf{y}|\mathbf{x}) = \mathcal{N}(\mathcal{A}(\mathbf{x}), \sigma_{\mathbf{y}}^2 \mathbf{I})$. For linear inverse problems, $\mathcal{A}$ is linear and can be denoted with a matrix instead, i.e., $\mathcal{A}(\mathbf{x}) = \mathrm{H}\mathbf{x}$, where we use the matrix $\mathrm{H}$ to denote a linear degradation operator throughout the paper.

$$\mathbf{y} = \mathcal{A}(\mathbf{x}) + \eta. \tag{1}$$

Inverse problems are commonly ill-posed, where many plausible data samples could correspond to a given degraded measurement, rendering a probabilistic approach essential. In a Bayesian framework, solving an inverse problem amounts to estimating (or sampling from) the posterior distribution $p(\mathbf{x}|\mathbf{y})$. As diffusion models (Ho et al., 2020; Song et al., 2021; Rombach et al., 2022) gain prominence in generative modeling, leveraging pre-trained unconditional diffusion models to solve inverse problems in a plug-and-play fashion is becoming increasingly attractive (Lugmayr et al., 2022; Kawar et al., 2022; Chung et al., 2023; Song et al., 2023a; Zhu et al., 2023; Rout et al., 2023; Song et al., 2024; Mardani et al., 2024; Janati et al., 2024; Gutha et al., 2025; Zhang et al., 2025; Moufad et al., 2025; Zilberstein et al., 2025). In this work, we advance this line of research by proposing a mode-seeking loss based inference-time guidance strategy for solving inverse problems with pre-trained unconditional diffusion models.

In a diffusion process, noise is added progressively to the samples of the input data distribution, i.e., $\mathbf{x}_0 \sim p_{data}(\mathbf{x}_0)$ to convert these into noisy samples gradually over a time horizon $t \in (0, T]$. This process can be reversed stochastically with a Stochastic Differential Equation (SDE) or deterministically with Probability Flow Ordinary Differential Equation (PF ODE), both of which require the score function of the marginal distribution at time $t$, i.e., $\nabla_{\mathbf{x}_t} \log p(\mathbf{x}_t)$ for all $t \in (0, T]$ (Song et al., 2021). The score functions are typically intractable, so a diffusion model $s_\theta(\mathbf{x}_t, t)$ is trained using score-matching loss (Vincent, 2011; Song et al., 2020) to approximate these.

**Conditional generation and Related works.** To sample from a posterior $p(\mathbf{x}_0|\mathbf{y})$ using diffusion models, where $\mathbf{y}$ is a given condition or measurement, it suffices to replace the unconditional score function $\nabla_{\mathbf{x}_t} \log p(\mathbf{x}_t)$ with the conditional score $\nabla_{\mathbf{x}_t} \log p(\mathbf{x}_t|\mathbf{y})$ in the reverse diffusion process mentioned above. Since $\nabla_{\mathbf{x}_t} \log p(\mathbf{x}_t|\mathbf{y}) = \nabla_{\mathbf{x}_t} \log p(\mathbf{x}_t) + \nabla_{\mathbf{x}_t} \log p(\mathbf{y}|\mathbf{x}_t)$, the former term can be replaced with the unconditional diffusion model $s_\theta(\mathbf{x}_t, t)$ but the latter term remains intractable due to the intractability of $p(\mathbf{y}|\mathbf{x}_t)$ (Chung et al., 2023), so several works (Chung et al., 2023; Song et al., 2023a; Peng et al., 2024; Boys et al., 2024; Zhang et al., 2025) rely on Gaussian approximations. Other approaches for posterior sampling (Trippe et al., 2022; Cardoso et al., 2023; Achituve et al., 2025) use sequential monte-carlo sampling techniques, while some others (Wang et al., 2024; Gutha et al., 2025; Xu et al., 2025) circumvent the need to estimate the conditional score by instead aiming for the MAP estimate, however, relying on modeling approximations which typically require solving the PF ODE (Song et al., 2021) in reverse time. This task is computationally feasible with a pre-trained consistency model (Song et al., 2023b). However, with only having access to a diffusion model, it quickly becomes prohibitively expensive, since it requires several neural function evaluations. To avoid this, in practice, a few-step Euler discretization is used to solve the PF ODE for MAP estimation (Wang et al., 2024; Gutha et al., 2025) or for posterior sampling in the case of Zhang et al. (2025). We refer to the survey by Daras et al. (2024) for a more comprehensive categorization of these methods.Unlike previous methods, our approach is based on minimizing a mode-seeking loss *at each reverse diffusion step*, which aligns the diffusion posterior with the measurement posterior.

**Contributions.** Our main contributions in this work are summarized below.

- We introduce the variational mode-seeking loss (VML), which, when minimized during each reverse diffusion step, steers the intermediate sample $\mathbf{x}_t$ towards the MAP estimate as $t \to 0$.

- For linear inverse problems, we derive a closed-form expression for VML without any approximations and also demonstrate the redundancy of certain terms by further theoretical analysis.

- Based on the previous insight, we propose a practically effective algorithm (VML-MAP) for solving inverse problems, and also a preconditioner for ill-conditioned linear operators.

- We demonstrate VML-MAP's effectiveness over other approaches through extensive experiments on diverse image-restoration tasks across multiple datasets.

## 2 BACKGROUND ON DIFFUSION MODELS

The forward process in a diffusion model corrupts a clean sample of the input data distribution i.e., $\mathbf{x}_0 \sim P(\mathbf{x}_0)$ into intermediate noisy sample $\mathbf{x}_t$, $t \in (0, T]$, modeled by the forward SDE given by Equation (2), where $\mathbf{f}(\cdot, t) : \mathbb{R}^n \to \mathbb{R}^n$, and $g : \mathbb{R} \to \mathbb{R}$ are the drift and diffusion coefficients, respectively, and $\mathbf{w}_t$ denotes a standard Wiener process (Song et al., 2021). The drift, diffusion coefficients, and $T$ are chosen such that the distribution of $\mathbf{x}_T$ is tractable to sample from and is typically independent of the input data distribution.

$$\mathrm{d}\mathbf{x}_t = \mathbf{f}(\mathbf{x}_t, t)\mathrm{d}t + g(t)\mathrm{d}\mathbf{w}_t. \tag{2}$$

The reverse SDE in Equation (3) converts a noisy sample $\mathbf{x}_t$ into a clean data sample $\mathbf{x}_0$, where $\bar{\mathbf{w}}_t$ denotes a standard Wiener process in reverse time. $p(\mathbf{x}_t)$ denotes the marginal distribution at time $t$. Its score function, i.e., $\nabla_{\mathbf{x}_t} \log p(\mathbf{x}_t)$, is usually intractable, so a neural network $s_\theta(\mathbf{x}_t, t)$ is trained using score-matching loss (Vincent, 2011; Song et al., 2020) to approximate this for all $t$.

$$\mathrm{d}\mathbf{x}_t = \{\mathbf{f}(\mathbf{x}_t, t) - g^2(t)\nabla_{\mathbf{x}_t} \log p(\mathbf{x}_t)\}\mathrm{d}t + g(t)\mathrm{d}\bar{\mathbf{w}}_t. \tag{3}$$

For a given choice of $\mathbf{f}, g$, the PF ODE in Equation (4) describes a deterministic process where, an intermediate sample $\mathbf{x}_t$ generated by the ODE share the same marginal probability $p(\mathbf{x}_t)$ as that simulated by the forward SDE for all $t \in (0, T]$.

$$\mathrm{d}\mathbf{x}_t = \{\mathbf{f}(\mathbf{x}_t, t) - \frac{1}{2}g^2(t)\nabla_{\mathbf{x}_t} \log p(\mathbf{x}_t)\}\mathrm{d}t. \tag{4}$$

To sample $\mathbf{x}_0 \sim p(\mathbf{x}_0)$, we first sample $\mathbf{x}_T \sim p(\mathbf{x}_T)$ and solve either the reverse SDE with SDE solvers or the PF ODE with ODE solvers using the learned score function $s_\theta(\mathbf{x}_t, t)$. Corresponding to the Variance Exploding (VE) SDE formulation from Song et al. (2021), throughout this paper, we fix $\mathbf{f}(\mathbf{x}_t, t) = \mathbf{0}$, $g(t) = \sqrt{\mathrm{d}\sigma^2(t)/\mathrm{d}t}$, where $\sigma(t)$ denotes the noise schedule for $t \in [0, T]$. We use $\sigma(t)$ and $\sigma_t$ interchangeably to denote the noise level at time $t$. With the above choice of $\mathbf{f}$, and $g$, the corresponding perturbation kernel is given by $p(\mathbf{x}_t|\mathbf{x}_0) = \mathcal{N}(\mathbf{x}_0, \sigma_t^2\mathbf{I})$, and $p(\mathbf{x}_T) \approx \mathcal{N}(0, \sigma_T^2\mathbf{I})$. Note that setting $\sigma(t) = t$ recovers the case of EDM preconditioning (Karras et al., 2022).

# 3 VARIATIONAL MODE-SEEKING LOSS

**Motivation.** Given a measurement $\mathbf{y}$, and an unconditional diffusion model that generates samples from $p(\mathbf{x}_0)$, we are primarily interested in finding the MAP estimate, i.e., $\arg\max_{\mathbf{x}_0} \log p(\mathbf{x}_0|\mathbf{y})$. The main motivation for VML stems from the following observation. Starting the reverse diffusion process from a fixed noisy sample $\mathbf{x}_t$ at time $t$ results in a distribution over $\mathbf{x}_0$, i.e., $p(\mathbf{x}_0|\mathbf{x}_t)$. If we find an optimal $\mathbf{x}_t^*$ such that $p(\mathbf{x}_0|\mathbf{x}_t^*)$ shares modes, i.e., high-density regions, with the posterior $p(\mathbf{x}_0|\mathbf{y})$, then, by starting the reverse diffusion process from $\mathbf{x}_t^*$ at time $t$, one may expect to generate a probable sample of the posterior. If we repeat the task of finding such $\mathbf{x}_t^*$ at each diffusion time step $t$, then, as $t \to 0$, $\mathbf{x}_t^*$ converges[1] to the MAP estimate as explained in the rest of this section.

For a fixed value of $\mathbf{x}_t$, say $\mathbf{x}_t = \boldsymbol{\gamma}$, the behavior of $p(\mathbf{x}_0|\mathbf{x}_t = \boldsymbol{\gamma})$ along various time steps of a diffusion process is shown in Figure 1. As $p(\mathbf{x}_0|\mathbf{x}_t = \boldsymbol{\gamma}) \propto p(\mathbf{x}_t = \boldsymbol{\gamma}|\mathbf{x}_0)p(\mathbf{x}_0)$, and with $p(\mathbf{x}_t|\mathbf{x}_0) = \mathcal{N}(\mathbf{x}_0, \sigma_t^2\mathbf{I})$, the distribution $p(\mathbf{x}_0|\mathbf{x}_t = \boldsymbol{\gamma})$ is essentially proportional to the product of $p(\mathbf{x}_0)$ and a Gaussian kernel with variance $\sigma_t^2$, centered at $\boldsymbol{\gamma}$ (due to the symmetrical form of $p(\mathbf{x}_t|\mathbf{x}_0)$). Since $\sigma_t \to 0$ as $t \to 0$, the dependence of $p(\mathbf{x}_0|\mathbf{x}_t = \boldsymbol{\gamma})$ on $\boldsymbol{\gamma}$ grows stronger as $t$ decreases, with $p(\mathbf{x}_0|\mathbf{x}_t = \boldsymbol{\gamma})$ converging to the Dirac delta function $\delta(\mathbf{x}_0 - \boldsymbol{\gamma})$ as $t \to 0$, for any $\boldsymbol{\gamma}$.

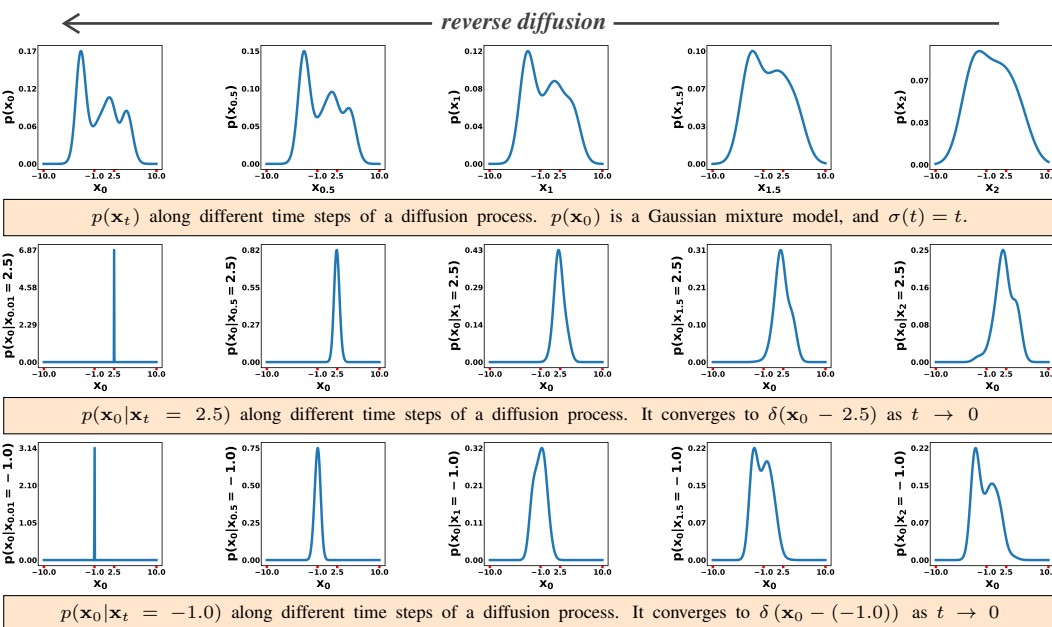

Figure 1: The figure depicts how the functional form of $p(\mathbf{x}_0|\mathbf{x}_t)$ gets peaky around $\mathbf{x}_t$ as $t \to 0$.

Suppose, at each time step $t$, we find an optimal $\boldsymbol{\gamma}_t^*$ such that $p(\mathbf{x}_0|\mathbf{x}_t = \boldsymbol{\gamma}_t^*)$ shares modes i.e., high-density regions with $p(\mathbf{x}_0|\mathbf{y})$. Note that for $t$ arbitrarily close to 0, $p(\mathbf{x}_0|\mathbf{x}_t = \boldsymbol{\gamma}_t^*)$ is an extremely peaky distribution around $\boldsymbol{\gamma}_t^*$. For this distribution to share modes with the posterior $p(\mathbf{x}_0|\mathbf{y})$, it is ideal for $\boldsymbol{\gamma}_t^*$ to be closer, and converging to the MAP estimate as $t \to 0$, since the MAP estimate is the highest posterior mode, i.e., the sample with the highest posterior probability density.

**Formalism.** From a variational perspective, at a diffusion time step $t$, $p(\mathbf{x}_0|\mathbf{x}_t)$ is a parameterized distribution of $\mathbf{x}_t$. During each reverse diffusion time step $t$, we aim to find a specific distribution $p(\mathbf{x}_0|\mathbf{x}_t^*)$ from the class of parameterized distributions $\{p(\mathbf{x}_0|\mathbf{x}_t)\}_{\mathbf{x}_t}$ such that $p(\mathbf{x}_0|\mathbf{x}_t^*)$ shares modes i.e., high-density regions with the posterior. The functional form of $p(\mathbf{x}_0|\mathbf{x}_t)$ getting arbitrarily peaky as $t \to 0$ implies that the optimal $\mathbf{x}_t^*$ ideally converges to the MAP estimate as $t \to 0$. The reverse KL divergence is known to promote this mode-matching behavior of distributions, so we choose $D_{\mathrm{KL}}(p(\mathbf{x}_0|\mathbf{x}_t)||p(\mathbf{x}_0|y))$ as the minimization objective at each time step, which we refer to as the variational mode-seeking loss (VML). In practice, however, finding the exact MAP estimate is extremely challenging, as the VML can be highly non-convex, rendering optimization approaches ineffective. Instead, we settle for the modes of the posterior found by the VML optimizer in practice.

---

[1]May require assumptions on $p(\mathbf{x}_t)$ and the convergence of $p(\mathbf{x}_0|\mathbf{x}_t)$. Note that $p(\mathbf{x}_t|\mathbf{x}_0) = \mathcal{N}(0, \sigma_t^2\mathbf{I})$

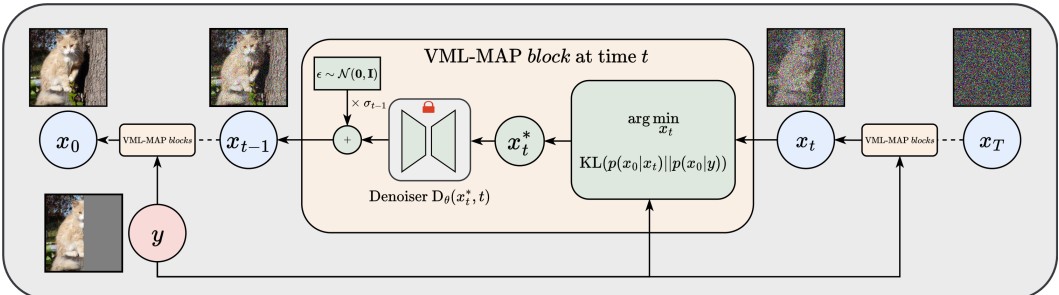

Figure 2: VML-MAP overview. Initially, $\mathbf{x}_T$ is sampled from Gaussian noise and passed to a VML-MAP block, optimizing for $\mathbf{x}_T^*$ that minimizes $D_{\mathrm{KL}}(p(\mathbf{x}_0|\mathbf{x}_T)||p(\mathbf{x}_0|\mathbf{y}))$ followed by a standard reverse diffusion step to output $\mathbf{x}_{T-1}$. The process is repeated for each diffusion step until $t = 0$.

**Proposition 1.** *The variational mode-seeking loss (VML) at diffusion time $t$, for a degradation operator $\mathcal{A}$, measurement $\mathbf{y}$, and measurement noise variance $\sigma_{\mathbf{y}}^2$ is given by*

$$\mathrm{VML} = D_{\mathrm{KL}}(p(\mathbf{x}_0|\mathbf{x}_t)||p(\mathbf{x}_0|\mathbf{y})) = -\log p(\mathbf{x}_t) - \frac{\|\mathrm{D}(\mathbf{x}_t, t) - \mathbf{x}_t\|^2}{2\sigma_t^2} - \frac{1}{2\sigma_t^2}\mathrm{Tr}\left\{\mathrm{Cov}[\mathbf{x}_0|\mathbf{x}_t]\right\}$$

$$+ \frac{1}{2\sigma_{\mathbf{y}}^2}\left(-2\mathbf{y}^\top \int_{\mathbf{x}_0} \mathcal{A}(\mathbf{x}_0)p(\mathbf{x}_0|\mathbf{x}_t)\mathrm{d}\mathbf{x}_0 + \int_{\mathbf{x}_0} \|\mathcal{A}(\mathbf{x}_0)\|^2 p(\mathbf{x}_0|\mathbf{x}_t)\mathrm{d}\mathbf{x}_0\right) + \mathrm{C}$$

*where $\mathrm{C}$ is a constant, independent of $\mathbf{x}_t$. $\mathrm{Tr}$ denotes the matrix trace, $\mathrm{Cov}$ denotes the covariance matrix, and $\mathrm{D}(\cdot, \cdot)$ denotes the true denoiser (see Appendix A.1).*

**Proposition 2.** *The variational mode-seeking loss (VML) at diffusion time $t$, for a **linear** degradation matrix $\mathrm{H}$, measurement $\mathbf{y}$, and measurement noise variance $\sigma_{\mathbf{y}}^2$ is given by*

$$\mathrm{VML} = D_{\mathrm{KL}}(p(\mathbf{x}_0|\mathbf{x}_t)||p(\mathbf{x}_0|\mathbf{y})) = -\log p(\mathbf{x}_t) - \frac{\|\mathrm{D}(\mathbf{x}_t, t) - \mathbf{x}_t\|^2}{2\sigma_t^2} - \frac{1}{2\sigma_t^2}\mathrm{Tr}\left\{\mathrm{Cov}[\mathbf{x}_0|\mathbf{x}_t]\right\}$$

$$+ \underbrace{\frac{\|\mathbf{y} - \mathrm{HD}(\mathbf{x}_t, t)\|^2}{2\sigma_{\mathbf{y}}^2}}_{\text{measurement consistency}} + \frac{1}{2\sigma_{\mathbf{y}}^2}\mathrm{Tr}\left\{\mathrm{HCov}[\mathbf{x}_0|\mathbf{x}_t]\mathrm{H}^\top\right\} + \mathrm{C}$$

*where $\mathrm{C}$ is a constant, independent of $\mathbf{x}_t$. $\mathrm{Tr}$ denotes the matrix trace, $\mathrm{Cov}$ denotes the covariance matrix, and $\mathrm{D}(\cdot, \cdot)$ denotes the true denoiser (see Appendix A.1).*

Proofs are provided in Appendix A.4. In the case of a linear degradation operator, VML has a measurement consistency term (see Proposition 2) which resembles the widely used approximation of the guidance term $\log p(\mathbf{y}|\mathbf{x}_t)$ in the literature (Chung et al., 2023; Song et al., 2023a), typically in the context of posterior sampling. However, VML arrives at this term without any modeling approximation, and in the context of MAP estimation. The remaining terms within VML are referred to as the prior terms, as they do not involve the measurement $\mathbf{y}$.

Note that the higher-order terms of VML involving $\mathrm{Cov}[\mathbf{x}_0|\mathbf{x}_t]$ are computationally demanding, especially when VML has to be differentiable, to use gradient-based optimization for minimization. Based on further theoretical insights from Appendix B, we hypothesize that these higher-order terms may not be crucial in practice, and propose the simplified VML (see Equation 5) for linear inverse problems denoted as $\mathrm{VML_S}$, where the higher-order terms are excluded.

$$\mathrm{VML_S} = \frac{\|\mathbf{y} - \mathrm{HD}(\mathbf{x}_t, t)\|^2}{2\sigma_{\mathbf{y}}^2} - \log p(\mathbf{x}_t) - \frac{\|\mathrm{D}(\mathbf{x}_t, t) - \mathbf{x}_t\|^2}{2\sigma_t^2} + \mathrm{C} \tag{5}$$

$$\nabla_{\mathbf{x}_t}\mathrm{VML_S} = \underbrace{-\frac{\partial \mathrm{D}^\top(\mathbf{x}_t, t)}{\partial \mathbf{x}_t}\frac{\mathrm{H}^\top(\mathbf{y} - \mathrm{HD}(\mathbf{x}_t, t))}{\sigma_{\mathbf{y}}^2}}_{\text{measurement consistency gradient}} - \underbrace{\frac{\partial \mathrm{D}^\top(\mathbf{x}_t, t)}{\partial \mathbf{x}_t}\frac{(\mathrm{D}(\mathbf{x}_t, t) - \mathbf{x}_t)}{\sigma_t^2}}_{\text{prior gradient}} \tag{6}$$

Equation (6) shows the gradient of $\mathrm{VML_S}$, which we use in practice during optimization. Figure 2 provides an overview, and Algorithm 1 shows the exact implementation details of our proposed

---

**Algorithm 1:** VML-MAP / VML-MAP$_{pre}$

---

**Input:** $\mathrm{D}_\theta(\cdot, \cdot), \mathrm{H}, \mathbf{y}, \sigma_{\mathbf{y}}, \sigma(\cdot), t_{i \in \{0, \ldots N\}}, K, \gamma$
**Output:** $\mathbf{x}_{t_0}$
**Initialize** $\mathbf{x}_{t_N} \sim \mathcal{N}(\mathbf{0}, \sigma_{t_N}^2 \mathbf{I})$
**for** $i \leftarrow N$ **to** $1$ **do**
    **for** $j \leftarrow 1$ **to** $K$ **do**
        **if** *VML-MAP* **then**
            $\mathbf{x}_{t_i} \leftarrow \mathbf{x}_{t_i} - \gamma \cdot \nabla_{\mathbf{x}_{t_i}} \mathrm{VML_S}$            `/* see Equation (6) */`
        **else if** *VML-MAP$_{pre}$* **then**
            $\mathbf{x}_{t_i} \leftarrow \mathbf{x}_{t_i} - \gamma \cdot \nabla_{\mathbf{x}_{t_i}} \mathrm{VML_{S_{pre}}}$      `/* see Equation (9) */`
    **end**
    $\mathbf{x}_{t_{i-1}} \sim \mathcal{N}(\mathrm{D}_\theta(\mathbf{x}_{t_i}, t_i), \sigma_{t_{i-1}}^2 \mathbf{I})$
**end**
**Return** $\mathbf{x}_{t_0}$

---

approach, which we refer to as VML-MAP. The inputs to Algorithm 1 consists of the diffusion denoiser $\mathrm{D}_\theta(\cdot, \cdot)$, the linear degradation matrix $\mathrm{H}$, the measurement $\mathbf{y}$ with noise variance $\sigma_{\mathbf{y}}^2$, the diffusion noise schedule $\sigma(\cdot)$, the total number of reverse diffusion steps $N$ with the discretized time step schedule specified by $t_{i \in \{0, \ldots N\}}$, where $t_0 = 0$, the gradient descent iterations per step given by $K$, and the learning rate $\gamma$. We use the notations $\sigma(t)$ and $\sigma_t$ interchangeably.

## 4 IMPROVED OPTIMIZATION

### 4.1 VML-MAP FOR IMAGE RESTORATION

Several image restoration tasks in computer vision, such as inpainting, super-resolution, deblurring, etc., can be modeled as linear inverse problems. In this section, we apply VML-MAP to the afore-mentioned image restoration tasks to understand its effectiveness in practice. In our experiments, we use 1000 images (each from a different class) from the ImageNet (Russakovsky et al., 2015) validation set with a resolution of $64 \times 64$ for evaluation using the corresponding pre-trained class-conditional diffusion model from Ho et al. (2020). We consider the challenging tasks of image inpainting with a half-mask (where the right half of the image is masked), $4\times$ super-resolution, and uniform deblurring with a $16 \times 16$ kernel. We make the deblurring task even more challenging by setting the singular values below a high threshold to zero. See Appendix C for further details. Using LPIPS and FID metrics, which capture measurement consistency and perceptual quality of the restored images, we report the performance of existing methods (DDRM (Kawar et al., 2022), $\Pi$GDM (Song et al., 2023a), and MAPGA (Gutha et al., 2025)) and compare against VML-MAP.

Table 1: Evaluation of baselines, VML-MAP and VML-MAP$_{pre}$ for noiseless inverse problems of half-mask inpainting, $4\times$ super-resolution, and deblurring on 1000 images of ImageNet64 validation set. Best values in **bold**, second best values underlined.

| Method | Inpainting | | $4\times$ Super-res | | Deblurring | |
|---|---|---|---|---|---|---|
| | LPIPS↓ | FID↓ | LPIPS↓ | FID↓ | LPIPS↓ | FID↓ |
| DDRM | 0.262 | 56.97 | 0.235 | 78.16 | 0.466 | 198.0 |
| $\Pi$GDM | 0.242 | 55.14 | 0.241 | 88.96 | 0.436 | 166.0 |
| MAPGA | 0.172 | 46.33 | 0.203 | 83.95 | 0.323 | 114.3 |
| **VML-MAP** | **0.146** | **38.70** | 0.136 | 61.90 | 0.356 | 105.5 |
| **VML-MAP**$_{pre}$ | **0.146** | **38.70** | **0.128** | **59.42** | **0.263** | **78.07** |

Quantitative evaluation results from Table 1 indicate the effectiveness of VML-MAP in practice over existing baselines for inpainting and super-resolution. Figure 3 presents a qualitative comparison of the same. Note that while DDRM, $\Pi$GDM, and MAPGA require access to the Singular Value

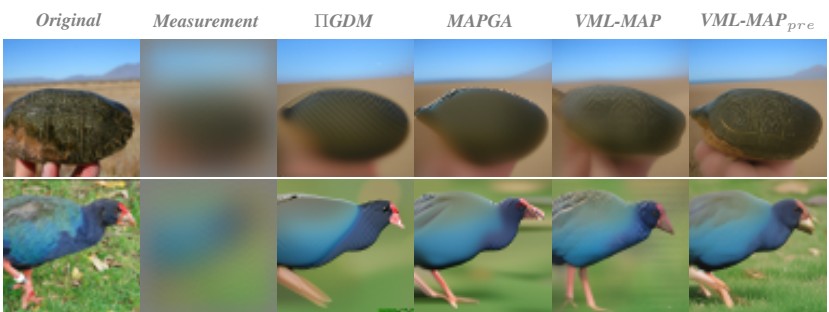

Figure 3: Half-mask inpainting, $4\times$ super-resolution, and deblurring tasks from Table 1.

Figure 4: Deblurring task from Table 1. Zoom in for the best view.

Decomposition (SVD) of the linear degradation matrix H to find the pseudoinverse of terms involved therein, VML-MAP only requires the forward operation of H. However, when H is ill-conditioned, VML-MAP can struggle with the optimization, as seems to be the case with deblurring. In the next section, we introduce a preconditioner to alleviate this problem, which also requires the SVD of H.

## 4.2 PRECONDITIONER

Preconditioners help accelerate convergence and thereby improve the effectiveness of the optimizer, especially when dealing with ill-conditioned loss objectives. Assuming a linear degradation matrix, H with SVD given by $H = U\Sigma V^\top$ (U,V denote the left and right singular orthogonal matrices respectively, with $\Sigma$ denoting the singular values matrix), we use the preconditioner P in Equation (7) to essentially replace the gradient $\nabla_{\mathbf{x}_t}\text{VML}_S$ in VML-MAP with the preconditioned gradient $P\nabla_{\mathbf{x}_t}\text{VML}_S$. We refer to this method as VML-MAP$_{pre}$, also presented in Algorithm 1.

$$P = \left(\frac{\partial D^\top(\mathbf{x}_t, t)}{\partial \mathbf{x}_t}\right) M^{-1} \left(\frac{\partial D^\top(\mathbf{x}_t, t)}{\partial \mathbf{x}_t}\right)^{-1} \tag{7}$$

$$M = (\mathbf{I} - \Sigma^+\Sigma) + H^\top H \tag{8}$$

$$\nabla_{\mathbf{x}_t}\text{VML}_{S_{pre}} = -\frac{\partial D^\top(\mathbf{x}_t, t)}{\partial \mathbf{x}_t} M^{-1} \frac{H^\top(\mathbf{y} - HD(\mathbf{x}_t, t))}{\sigma_{\mathbf{y}}^2} - \frac{\partial D^\top(\mathbf{x}_t, t)}{\partial \mathbf{x}_t} M^{-1} \frac{(D(\mathbf{x}_t, t) - \mathbf{x}_t)}{\sigma_t^2} \tag{9}$$

$\Sigma^+$ above denotes the pseudoinverse of $\Sigma$. Equation (9) further expands the preconditioned gradient $P\nabla_{\mathbf{x}_t}\text{VML}_S$ (that we hereon denote with $\nabla_{\mathbf{x}_t}\text{VML}_{S_{pre}}$). Note that $\frac{\partial D^\top(\mathbf{x}_t,t)}{\partial \mathbf{x}_t} = \sigma_t^2 \text{Cov}[\mathbf{x}_0|\mathbf{x}_t]$ (see Appendix A.2). Assuming $\text{Cov}[\mathbf{x}_0|\mathbf{x}_t] \succ 0$ (i.e., positive definite), implies the positive definiteness and hence the invertibility of $\frac{\partial D^\top(\mathbf{x}_t,t)}{\partial \mathbf{x}_t}$, which further implies the invertibility of P.

We use the SVD of H to compute $M^{-1}$ efficiently, which in turn is used to compute $\nabla_{\mathbf{x}_t}\text{VML}_{S_{pre}}$ for VML-MAP$_{pre}$ (see Equation 9). Note that DDRM, $\Pi$GDM, and MAPGA also require SVD of

H, which makes it a fair comparison against VML-MAP$_{pre}$. Results from Table 1 also indicate the effectiveness of the preconditioner on $4\times$ super-resolution and deblurring tasks as VML-MAP$_{pre}$ shows significant improvements in LPIPS and FID over VML-MAP and other baselines, denoting higher perceptual quality of the restored images. For inpainting, VML-MAP and VML-MAP$_{pre}$ are essentially equivalent since $M = P = I$. Figure 4 presents a qualitative comparison of the restored samples with different baselines, VML-MAP, and VML-MAP$_{pre}$ for the deblurring task.

## 5 MAIN EXPERIMENTS AND RESULTS

Our experiments in Table 2 include half-mask inpainting, $4\times$ super-resolution, and the deblurring tasks previously mentioned in Section 4.1. In Table 3, we focus on the image inpainting task with several masks. In all our experiments, we evaluate on 100 validation images of ImageNet (Deng et al., 2009) with a resolution of $256 \times 256$, using the unconditional ImageNet256 pre-trained diffusion model from Ho et al. (2020), and on 100 images of FFHQ (Karras et al., 2019), with a resolution of $256 \times 256$, using the FFHQ256 pre-trained diffusion model from Chung et al. (2023). With FID and LPIPS as evaluation metrics, we compare VML-MAP and VML-MAP$_{pre}$ against several baselines such as DDRM (Kawar et al., 2022), ΠGDM (Song et al., 2023a), MAPGA (Gutha et al., 2025), and DAPS (Zhang et al., 2025). In all our experiments, we fix a budget of approximately 1000 neural function evaluations of the diffusion model for both VML-MAP and VML-MAP$_{pre}$. For DAPS, we consider two configurations, with 1000, 4000 neural function evaluations denoted DAPS-1K, DAPS-4K, respectively, see Zhang et al. (2025). We refer to Appendix C for further details regarding the experiment setup, hyperparameters, and runs across different seeds.

Table 2: Evaluation of several image restoration methods on noiseless inverse problems of half-mask inpainting, $4\times$ super-resolution, and deblurring on 100 validation images of ImageNet256, and on 100 images of FFHQ256. Best values in **bold**, second best values underlined.

| Dataset | Method | Inpainting | | $4\times$ Super-res | | Deblurring | |
|---|---|---|---|---|---|---|---|
| | | LPIPS↓ | FID↓ | LPIPS↓ | FID↓ | LPIPS↓ | FID↓ |
| ImageNet | DDRM | 0.391 | 102.8 | 0.289 | 90.50 | 0.618 | 234.6 |
| | ΠGDM | 0.373 | 103.8 | 0.292 | 83.80 | 0.560 | 231.3 |
| | MAPGA | 0.289 | 83.13 | 0.274 | 76.13 | 0.459 | 194.7 |
| | DAPS-1K | 0.385 | 98.72 | 0.256 | 73.33 | 0.607 | 220.3 |
| | DAPS-4K | 0.371 | 94.98 | 0.243 | 69.98 | 0.593 | 224.4 |
| | **VML-MAP** | **0.265** | **69.21** | **0.192** | 59.91 | 0.511 | 205.2 |
| | **VML-MAP**$_{pre}$ | **0.265** | **69.21** | 0.194 | **56.60** | **0.371** | **166.7** |
| FFHQ | DDRM | 0.243 | 71.21 | 0.154 | 70.47 | 0.307 | 117.5 |
| | ΠGDM | 0.234 | 71.02 | 0.147 | 68.65 | 0.293 | 114.3 |
| | MAPGA | 0.206 | 65.29 | 0.132 | 64.45 | 0.235 | 120.9 |
| | DAPS-1K | 0.232 | 59.93 | 0.113 | 61.59 | 0.259 | 100.2 |
| | DAPS-4K | 0.223 | 60.20 | 0.100 | 58.38 | 0.230 | 95.05 |
| | **VML-MAP** | **0.180** | **51.94** | **0.099** | 56.58 | 0.247 | 99.84 |
| | **VML-MAP**$_{pre}$ | **0.180** | **51.94** | **0.099** | **52.04** | **0.183** | **93.99** |

The quantitative results from Table 2 and the corresponding qualitative comparisons from Figures 5 and 6 highlight the effectiveness of VML-MAP and VML-MAP$_{pre}$ over existing methods. Also from Figure 7, which shows the tradeoff between runtime and perceptual quality of reconstructed images for several baselines, VML-MAP achieves better perceptual quality with lower runtime than other methods, highlighting its computational efficiency. Also, VML-MAP$_{pre}$ has an almost similar runtime as VML-MAP, as we compute $M^{-1}$ in Equation (9) with negligible overhead using SVD.

As mentioned in Section 4.2, DDRM, ΠGDM, MAPGA, and VML-MAP$_{pre}$ require SVD of H, while DAPS and VML-MAP only require the forward operation of H. To ensure a fair comparison

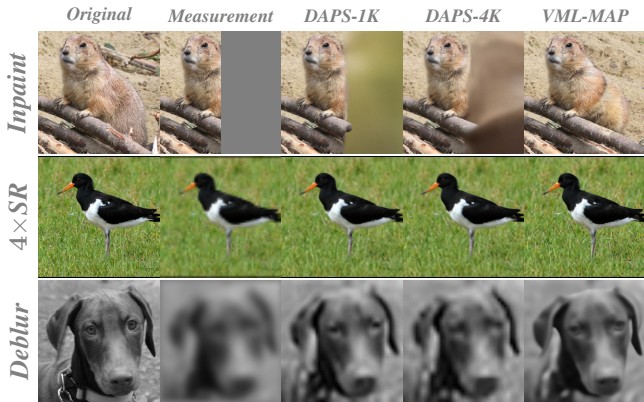

Figure 5: Half-mask inpainting, $4\times$ super-resolution, and deblurring tasks from Table 2. Note that DAPS-1K, DAPS-4K, and VML-MAP only require the forward operation of the linear degradation operator and not its SVD. Zoom in for the best view.

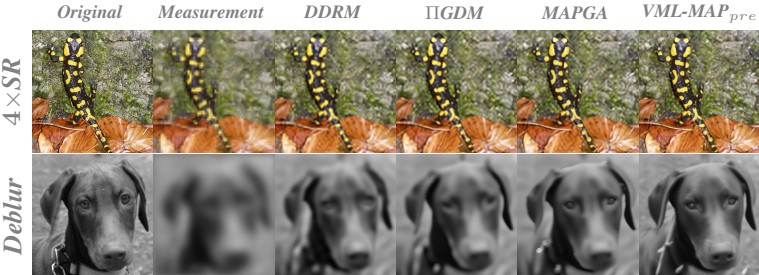

Figure 6: $4\times$ Super-resolution, and Deblurring tasks from Table 2. Note that DDRM, ΠGDM, MAPGA, and VML-MAP$_{pre}$ require the SVD of the linear degradation operator.

Table 3: Evaluation of several image restoration methods on noiseless inpainting with expand mask, box mask, super-resolution mask, and random mask on 100 validation images of ImageNet256, and on 100 images of FFHQ256. Best values in **bold**, second best values underlined.

| Dataset | Method | Expand mask | | Box mask | | Sup-res mask | | Random mask | |
|---------|--------|-------------|------|----------|------|--------------|------|-------------|------|
| | | LPIPS↓ | FID↓ | LPIPS↓ | FID↓ | LPIPS↓ | FID↓ | LPIPS↓ | FID↓ |
| ImageNet | DDRM | 0.548 | 152.0 | 0.211 | 133.1 | 0.089 | 29.24 | 0.052 | 21.81 |
| | ΠGDM | 0.523 | 155.7 | 0.203 | 123.9 | 0.080 | 26.54 | 0.054 | 24.48 |
| | MAPGA | 0.466 | 126.9 | 0.150 | 91.14 | 0.087 | 25.02 | 0.051 | 20.31 |
| | DAPS-1K | 0.550 | 167.2 | 0.201 | 112.6 | 0.132 | 53.46 | 0.092 | 35.93 |
| | DAPS-4K | 0.521 | 154.0 | 0.187 | 102.8 | 0.114 | 45.65 | 0.082 | 31.67 |
| | **VML-MAP** | **0.434** | **116.2** | **0.138** | **75.80** | **0.068** | **19.96** | **0.044** | **16.14** |
| FFHQ | DDRM | 0.426 | 151.1 | 0.087 | 50.60 | 0.031 | **20.75** | 0.027 | 18.79 |
| | ΠGDM | 0.415 | 146.3 | 0.084 | 46.95 | 0.033 | 23.98 | 0.027 | 19.93 |
| | MAPGA | 0.393 | 129.3 | 0.070 | 41.34 | 0.036 | 27.01 | 0.027 | 20.88 |
| | DAPS-1K | 0.423 | 126.7 | 0.076 | 33.17 | 0.057 | 47.21 | 0.059 | 45.55 |
| | DAPS-4K | 0.398 | 117.5 | 0.077 | 33.82 | 0.051 | 42.32 | 0.051 | 40.06 |
| | **VML-MAP** | **0.365** | **112.9** | **0.057** | **28.38** | **0.027** | 21.40 | **0.022** | **16.30** |

in this regard, we evaluate all the methods on the image inpainting task with several masks, where the SVD of $\mathbb{H}$ is trivial. The quantitative results from Table 3 and the qualitative comparison in Appendix C.3 reveal the superior performance of VML-MAP in all the inpainting tasks.

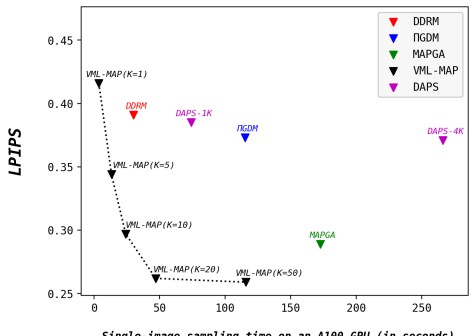 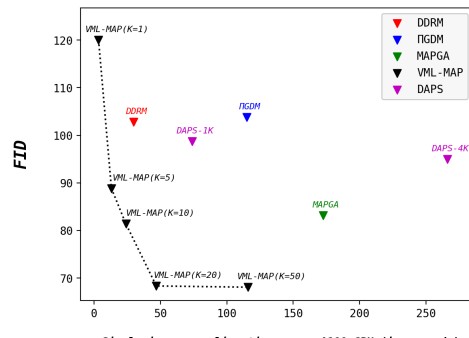

Figure 7: Runtime vs Perceptual quality for the half-mask inpainting experiment in Table 2. DDRM and ΠGDM use 500 and 1000 reverse diffusion steps, respectively, to achieve their best results. For VML-MAP, we fix the reverse diffusion steps ($N$) to 20, and vary the number of gradient-descent iterations per step ($K$) across $\{1, 5, 10, 20, 50\}$. See Appendix C.1 for more details. VML-MAP achieves better perceptual quality with lower compute than other methods.

Note that the effectiveness of the preconditioner for $4\times$ super-resolution and deblurring tasks reinforces the need for efficient optimizers in boosting the performance of VML-MAP in practice. In Appendix D, we extend VML-MAP to Latent Diffusion Models (LDM), and compare with other baselines. In practice, even for linear inverse problems, we observe that the non-linearity of the LDM decoder makes the optimization highly challenging, which again highlights the need for better optimizers. By treating the LDM encoder and the decoder as identity mappings, this can also serve as an extension of VML-MAP to non-linear inverse problems using pixel diffusion models.

## 6 CONCLUSION

In this work, we proposed a training-free guidance method that steers the intermediate sample $\mathbf{x}_t$ of an unconditional diffusion model towards the MAP estimate, i.e., $\arg\max_{\mathbf{x}_0} \log p(\mathbf{x}_0|\mathbf{y})$, for a given measurement $\mathbf{y}$, thereby enabling the solution of downstream inverse problems. The core of our approach is a novel formulation based on minimizing the KL divergence between $p(\mathbf{x}_0|\mathbf{x}_t)$ and $p(\mathbf{x}_0|\mathbf{y})$, which we define as the variational mode-seeking loss (VML). We derived VML in a closed form for linear inverse problems without any modeling approximations and use it within our proposed algorithm (VML-MAP), which optimizes VML at each reverse diffusion step. To address the optimization difficulties arising from ill-conditioned linear degradation operators, we proposed a preconditioned variant (VML-MAP$_{pre}$) that offers a simple yet effective remedy. Finally, we demonstrated the effectiveness of our approach through extensive experiments on several image restoration inverse problems across multiple real-world datasets.

## 7 LIMITATIONS

Although this paper primarily focuses on developing a principled framework and establishing the theoretical foundations of VML, the availability of a practically effective optimizer for minimizing the VML objective is equally critical. In our experiments, we found that gradient descent performs sufficiently well to validate the proposed framework empirically. Nonetheless, approximate higher-order methods and advanced optimization strategies have the potential to improve performance further, as illustrated by our proposed preconditioner for ill-conditioned linear inverse problems. We also observed that the VML objective exhibits notable sensitivity to measurement noise ($\sigma_{\mathbf{y}}$) in practice. For instance, in inpainting, while increasing $\sigma_{\mathbf{y}}$ from $0.001$ to $0.01$ still preserves most of the perceptual content of $\mathbf{y}$, the measurement consistency term in VML is downweighted by a factor of $100$, which in turn introduces blurry artifacts in the reconstructed images. In the case of LDMs, the nonlinearity of the decoder exacerbates these optimization challenges, even when addressing linear inverse problems. Importantly, while advanced optimization techniques may enhance performance, they must not come at the expense of prohibitive computational costs. Designing optimizers that are both efficient and practically feasible remains an essential direction for future work.

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

# A APPENDIX

## A.1 TWEEDIE'S FORMULA

$$D(\mathbf{x}_t, t) = \mathbb{E}[\mathbf{x}_0 | \mathbf{x}_t] = \int_{\mathbf{x}_0} \mathbf{x}_0 p(\mathbf{x}_0 | \mathbf{x}_t) \mathrm{d}\mathbf{x}_0 = \mathbf{x}_t + \sigma_t^2 \nabla_{\mathbf{x}_t} \log p(\mathbf{x}_t) \approx \mathbf{x}_t + \sigma_t^2 S_\theta(\mathbf{x}_t, t)$$

where, $D(\mathbf{x}_t, t)$ denotes the true denoiser, and $S_\theta(\cdot, \cdot)$ denotes the learned score function.

*Proof.*

$$\nabla_{\mathbf{x}_t} p(\mathbf{x}_t) = \nabla_{\mathbf{x}_t} \int_{\mathbf{x}_0} p(\mathbf{x}_t | \mathbf{x}_0) p(\mathbf{x}_0) \mathrm{d}\mathbf{x}_0 = \int_{\mathbf{x}_0} p(\mathbf{x}_0) \nabla_{\mathbf{x}_t} p(\mathbf{x}_t | \mathbf{x}_0) \mathrm{d}\mathbf{x}_0$$

$$\nabla_{\mathbf{x}_t} p(\mathbf{x}_t) = \int_{\mathbf{x}_0} p(\mathbf{x}_0) p(\mathbf{x}_t | \mathbf{x}_0) \frac{\mathbf{x}_0 - \mathbf{x}_t}{\sigma_t^2} \mathrm{d}\mathbf{x}_0 \quad \{\text{Note that } p(\mathbf{x}_t | \mathbf{x}_0) = \mathcal{N}(\mathbf{x}_0, \sigma_t^2 \mathbf{I})\}$$

$$\sigma_t^2 \nabla_{\mathbf{x}_t} p(\mathbf{x}_t) = \left( \int_{\mathbf{x}_0} \mathbf{x}_0 p(\mathbf{x}_0) p(\mathbf{x}_t | \mathbf{x}_0) \mathrm{d}\mathbf{x}_0 \right) - \mathbf{x}_t p(\mathbf{x}_t)$$

$$\mathbf{x}_t p(\mathbf{x}_t) + \sigma_t^2 \nabla_{\mathbf{x}_t} p(\mathbf{x}_t) = \int_{\mathbf{x}_0} \mathbf{x}_0 p(\mathbf{x}_0) p(\mathbf{x}_t | \mathbf{x}_0) \mathrm{d}\mathbf{x}_0 = p(\mathbf{x}_t) \mathbb{E}[\mathbf{x}_0 | \mathbf{x}_t]$$

$$\mathbf{x}_t + \sigma_t^2 \nabla_{\mathbf{x}_t} \log p(\mathbf{x}_t) = \mathbb{E}[\mathbf{x}_0 | \mathbf{x}_t] \approx \mathbf{x}_t + \sigma_t^2 S_\theta(\mathbf{x}_t, t)$$

$\square$

## A.2 COVARIANCE FORMULA

$$\mathrm{Cov}[\mathbf{x}_0 | \mathbf{x}_t] = \int_{\mathbf{x}_0} (\mathbf{x}_0 - \mathbb{E}[\mathbf{x}_0 | \mathbf{x}_t])(\mathbf{x}_0 - \mathbb{E}[\mathbf{x}_0 | \mathbf{x}_t])^\top p(\mathbf{x}_0 | \mathbf{x}_t) \mathrm{d}\mathbf{x}_0 = \sigma_t^2 \frac{\partial D(\mathbf{x}_t, t)}{\partial \mathbf{x}_t} \approx \sigma_t^2 \frac{\partial D_\theta(\mathbf{x}_t, t)}{\partial \mathbf{x}_t}$$

where, $D(\mathbf{x}_t, t) = \mathbb{E}[\mathbf{x}_0 | \mathbf{x}_t]$ is the true denoiser, and $D_\theta(\mathbf{x}_t, t)$ is the learned denoiser.

*Proof.*

$$D(\mathbf{x}_t, t) = \mathbb{E}[\mathbf{x}_0 | \mathbf{x}_t] = \int_{\mathbf{x}_0} p(\mathbf{x}_0 | \mathbf{x}_t) \mathrm{d}\mathbf{x}_0$$

$$\frac{\partial D(\mathbf{x}_t, t)}{\partial \mathbf{x}_t} = \frac{\partial}{\partial \mathbf{x}_t} \int_{\mathbf{x}_0} \mathbf{x}_0 p(\mathbf{x}_0 | \mathbf{x}_t) = \int_{\mathbf{x}_0} \mathbf{x}_0 p(\mathbf{x}_0) \frac{\partial}{\partial \mathbf{x}_t} \left( \frac{p(\mathbf{x}_t | \mathbf{x}_0)}{p(\mathbf{x}_t)} \right) \mathrm{d}\mathbf{x}_0$$

$$\{\text{Note that } p(\mathbf{x}_t | \mathbf{x}_0) = \mathcal{N}(\mathbf{x}_0, \sigma_t^2 \mathbf{I})\}$$

$$\frac{\partial D(\mathbf{x}_t, t)}{\partial \mathbf{x}_t} = \int_{\mathbf{x}_0} \mathbf{x}_0 p(\mathbf{x}_0) \left( \frac{p(\mathbf{x}_t) p(\mathbf{x}_t | \mathbf{x}_0) \frac{(\mathbf{x}_0 - \mathbf{x}_t)^\top}{\sigma_t^2} - p(\mathbf{x}_t | \mathbf{x}_0) \frac{\partial p(\mathbf{x}_t)}{\partial \mathbf{x}_t}}{p(\mathbf{x}_t)^2} \right) \mathrm{d}\mathbf{x}_0$$

$$\frac{\partial D(\mathbf{x}_t, t)}{\partial \mathbf{x}_t} = \left( \int_{\mathbf{x}_0} \mathbf{x}_0 p(\mathbf{x}_0) \frac{p(\mathbf{x}_t | \mathbf{x}_0) \frac{(\mathbf{x}_0 - \mathbf{x}_t)^\top}{\sigma_t^2}}{p(\mathbf{x}_t)} \mathrm{d}\mathbf{x}_0 \right) - \left( \int_{\mathbf{x}_0} \mathbf{x}_0 p(\mathbf{x}_0) \frac{p(\mathbf{x}_t | \mathbf{x}_0) \frac{\partial p(\mathbf{x}_t)}{\partial \mathbf{x}_t}}{p(\mathbf{x}_t)^2} \mathrm{d}\mathbf{x}_0 \right)$$

$$\frac{\partial D(\mathbf{x}_t, t)}{\partial \mathbf{x}_t} = \left( \int_{\mathbf{x}_0} \mathbf{x}_0 \frac{(\mathbf{x}_0 - \mathbf{x}_t)^\top}{\sigma_t^2} p(\mathbf{x}_0 | \mathbf{x}_t) \mathrm{d}\mathbf{x}_0 \right) - \left( \int_{\mathbf{x}_0} \mathbf{x}_0 \frac{\partial \log p(\mathbf{x}_t)}{\partial \mathbf{x}_t} p(\mathbf{x}_0 | \mathbf{x}_t) \mathrm{d}\mathbf{x}_0 \right)$$

$$\frac{\partial D(\mathbf{x}_t, t)}{\partial \mathbf{x}_t} = \int_{\mathbf{x}_0} \mathbf{x}_0 \left( \frac{(\mathbf{x}_0 - \mathbf{x}_t)^\top}{\sigma_t^2} - \frac{\partial \log p(\mathbf{x}_t)}{\partial \mathbf{x}_t} \right) p(\mathbf{x}_0 | \mathbf{x}_t) \mathrm{d}\mathbf{x}_0$$

$$\frac{\partial D(\mathbf{x}_t, t)}{\partial \mathbf{x}_t} = \int_{\mathbf{x}_0} \mathbf{x}_0 \frac{(\mathbf{x}_0 - D(\mathbf{x}_t, t))^\top}{\sigma_t^2} p(\mathbf{x}_0 | \mathbf{x}_t) \mathrm{d}\mathbf{x}_0$$

$$\sigma_t^2 \frac{\partial D(\mathbf{x}_t, t)}{\partial \mathbf{x}_t} = \int_{\mathbf{x}_0} \mathbf{x}_0 (\mathbf{x}_0 - D(\mathbf{x}_t, t))^\top p(\mathbf{x}_0 | \mathbf{x}_t) \mathrm{d}\mathbf{x}_0$$

$$\sigma_t^2 \frac{\partial D(\mathbf{x}_t, t)}{\partial \mathbf{x}_t} = \int_{\mathbf{x}_0} (\mathbf{x}_0 - D(\mathbf{x}_t, t))(\mathbf{x}_0 - D(\mathbf{x}_t, t))^\top p(\mathbf{x}_0 | \mathbf{x}_t) \mathrm{d}\mathbf{x}_0$$

$$\sigma_t^2 \frac{\partial D(\mathbf{x}_t, t)}{\partial \mathbf{x}_t} = \mathrm{Cov}[\mathbf{x}_0 | \mathbf{x}_t] \approx \sigma_t^2 \frac{\partial D_\theta(\mathbf{x}_t, t)}{\partial \mathbf{x}_t}$$

$\square$

### A.3 LEMMAS

**Lemma 1.** $\int_{\mathbf{x}_0} \|\mathbf{x}_0\|^2 p(\mathbf{x}_0|\mathbf{x}_t)\mathrm{d}\mathbf{x}_0 = \mathrm{Tr}\left\{\mathrm{Cov}[\mathbf{x}_0|\mathbf{x}_t]\right\} + \|\mathrm{D}(\mathbf{x}_t, t)\|^2$

*Proof.*

$$
\int_{\mathbf{x}_0} \|\mathbf{x}_0\|^2 p(\mathbf{x}_0|\mathbf{x}_t)\mathrm{d}\mathbf{x}_0 = \int_{\mathbf{x}_0} \mathrm{Tr}\left\{\mathbf{x}_0\mathbf{x}_0^\top\right\} p(\mathbf{x}_0|\mathbf{x}_t)\mathrm{d}\mathbf{x}_0 = \mathrm{Tr}\left\{\int_{\mathbf{x}_0} \mathbf{x}_0\mathbf{x}_0^\top p(\mathbf{x}_0|\mathbf{x}_t)\mathrm{d}\mathbf{x}_0\right\}
$$

$$
\int_{\mathbf{x}_0} \|\mathbf{x}_0\|^2 p(\mathbf{x}_0|\mathbf{x}_t)\mathrm{d}\mathbf{x}_0 = \mathrm{Tr}\left\{\mathrm{Cov}[\mathbf{x}_0|\mathbf{x}_t] + \mathrm{D}(\mathbf{x}_t, t)\mathrm{D}(\mathbf{x}_t, t)^\top\right\}
$$

$$
\int_{\mathbf{x}_0} \|\mathbf{x}_0\|^2 p(\mathbf{x}_0|\mathbf{x}_t)\mathrm{d}\mathbf{x}_0 = \mathrm{Tr}\left\{\mathrm{Cov}[\mathbf{x}_0|\mathbf{x}_t]\right\} + \|\mathrm{D}(\mathbf{x}_t, t)\|^2
$$

$\square$

**Lemma 2.** $\int_{\mathbf{x}_0} \|\mathrm{H}\mathbf{x}_0\|^2 p(\mathbf{x}_0|\mathbf{x}_t)\mathrm{d}\mathbf{x}_0 = \mathrm{Tr}\left\{\mathrm{H}\mathrm{Cov}[\mathbf{x}_0|\mathbf{x}_t]\mathrm{H}^\top\right\} + \|\mathrm{H}\mathrm{D}(\mathbf{x}_t, t)\|^2$

*Proof.*

$$
\int_{\mathbf{x}_0} \|\mathrm{H}\mathbf{x}_0\|^2 p(\mathbf{x}_0|\mathbf{x}_t)\mathrm{d}\mathbf{x}_0 = \int_{\mathbf{x}_0} \mathrm{Tr}\left\{\mathrm{H}\mathbf{x}_0\mathbf{x}_0^\top\mathrm{H}^\top\right\} p(\mathbf{x}_0|\mathbf{x}_t)\mathrm{d}\mathbf{x}_0
$$

$$
\int_{\mathbf{x}_0} \|\mathrm{H}\mathbf{x}_0\|^2 p(\mathbf{x}_0|\mathbf{x}_t)\mathrm{d}\mathbf{x}_0 = \mathrm{Tr}\left\{\mathrm{H}\left(\int_{\mathbf{x}_0} \mathbf{x}_0\mathbf{x}_0^\top p(\mathbf{x}_0|\mathbf{x}_t)\mathrm{d}\mathbf{x}_0\right)\mathrm{H}^\top\right\}
$$

$$
\int_{\mathbf{x}_0} \|\mathrm{H}\mathbf{x}_0\|^2 p(\mathbf{x}_0|\mathbf{x}_t)\mathrm{d}\mathbf{x}_0 = \mathrm{Tr}\left\{\mathrm{H}\mathrm{Cov}[\mathbf{x}_0|\mathbf{x}_t]\mathrm{H}^\top + \mathrm{H}\mathrm{D}(\mathbf{x}_t, t)\mathrm{D}(\mathbf{x}_t, t)^\top\mathrm{H}^\top\right\}
$$

$$
\int_{\mathbf{x}_0} \|\mathrm{H}\mathbf{x}_0\|^2 p(\mathbf{x}_0|\mathbf{x}_t)\mathrm{d}\mathbf{x}_0 = \mathrm{Tr}\left\{\mathrm{H}\mathrm{Cov}[\mathbf{x}_0|\mathbf{x}_t]\mathrm{H}^\top\right\} + \|\mathrm{H}\mathrm{D}(\mathbf{x}_t, t)\|^2
$$

$\square$

### A.4 PROOFS

**Proposition 1.** *The variational mode-seeking-loss (VML) at diffusion time $t$, for a non-linear degradation operator $\mathcal{A}$, measurement $\mathbf{y}$, and measurement noise variance $\sigma_{\mathbf{y}}^2$ is given by*

$$
\mathrm{VML} = D_{\mathrm{KL}}(p(\mathbf{x}_0|\mathbf{x}_t)||p(\mathbf{x}_0|\mathbf{y})) = -\log p(\mathbf{x}_t) - \frac{\|\mathrm{D}(\mathbf{x}_t, t) - \mathbf{x}_t\|^2}{2\sigma_t^2} - \frac{1}{2\sigma_t^2}\mathrm{Tr}\left\{\mathrm{Cov}[\mathbf{x}_0|\mathbf{x}_t]\right\}
$$

$$
+ \frac{1}{2\sigma_{\mathbf{y}}^2}\left(-2\mathbf{y}^\top \int_{\mathbf{x}_0} \mathcal{A}(\mathbf{x}_0)p(\mathbf{x}_0|\mathbf{x}_t)\mathrm{d}\mathbf{x}_0 + \int_{\mathbf{x}_0} \|\mathcal{A}(\mathbf{x}_0)\|^2 p(\mathbf{x}_0|\mathbf{x}_t)\mathrm{d}\mathbf{x}_0\right) + \mathrm{C}
$$

*where $\mathrm{C}$ is a constant, independent of $\mathbf{x}_t$. $\mathrm{Tr}$ denotes the matrix trace, $\mathrm{Cov}$ denotes the covariance matrix, and $\mathrm{D}(\cdot, \cdot)$ denotes the denoiser.*

*Proof.*

$$
D_{\mathrm{KL}}(p(\mathbf{x}_0|\mathbf{x}_t)||p(\mathbf{x}_0|\mathbf{y})) = \int_{\mathbf{x}_0} p(\mathbf{x}_0|\mathbf{x}_t)\log \frac{p(\mathbf{x}_0|\mathbf{x}_t)}{p(\mathbf{x}_0|\mathbf{y})}\mathrm{d}\mathbf{x}_0
$$

$$
D_{\mathrm{KL}}(p(\mathbf{x}_0|\mathbf{x}_t)||p(\mathbf{x}_0|\mathbf{y})) = \int_{\mathbf{x}_0} p(\mathbf{x}_0|\mathbf{x}_t)\log \frac{p(\mathbf{x}_t|\mathbf{x}_0)\cancel{p(\mathbf{x}_0)}p(\mathbf{y})}{p(\mathbf{x}_t)p(\mathbf{y}|\mathbf{x}_0)\cancel{p(\mathbf{x}_0)}}\mathrm{d}\mathbf{x}_0
$$

$$
D_{\mathrm{KL}}(p(\mathbf{x}_0|\mathbf{x}_t)||p(\mathbf{x}_0|\mathbf{y})) = \log p(\mathbf{y}) - \log p(\mathbf{x}_t) + \int_{\mathbf{x}_0} p(\mathbf{x}_0|\mathbf{x}_t)\log \frac{p(\mathbf{x}_t|\mathbf{x}_0)}{p(\mathbf{y}|\mathbf{x}_0)}\mathrm{d}\mathbf{x}_0
$$

$$
D_{\mathrm{KL}}(p(\mathbf{x}_0|\mathbf{x}_t)||p(\mathbf{x}_0|\mathbf{y})) = \log p(\mathbf{y}) - \log p(\mathbf{x}_t) + \left(\int_{\mathbf{x}_0} p(\mathbf{x}_0|\mathbf{x}_t)\log p(\mathbf{x}_t|\mathbf{x}_0)\mathrm{d}\mathbf{x}_0\right)
$$

$$
- \left(\int_{\mathbf{x}_0} p(\mathbf{x}_0|\mathbf{x}_t)\log p(\mathbf{y}|\mathbf{x}_0)\mathrm{d}\mathbf{x}_0\right)
$$

$\left\{\text{Note that } p(\mathbf{x}_t|\mathbf{x}_0) = \mathcal{N}(\mathbf{x}_0, \sigma_t^2\mathbf{I}) \text{ and } p(\mathbf{y}|\mathbf{x}_0) = \mathcal{N}(\mathcal{A}(\mathbf{x}_0), \sigma_{\mathbf{y}}^2\mathbf{I}). \text{ Also, let } \mathbf{x}_0 \in \mathbb{R}^n \text{ and } \mathbf{y} \in \mathbb{R}^m\right\}$

$$D_{\mathrm{KL}}(p(\mathbf{x}_0|\mathbf{x}_t)||p(\mathbf{x}_0|\mathbf{y})) = -\log p(\mathbf{x}_t) - \frac{1}{2}\left(\int_{\mathbf{x}_0} p(\mathbf{x}_0|\mathbf{x}_t)\frac{\|\mathbf{x}_t - \mathbf{x}_0\|^2}{\sigma_t^2}\mathrm{d}\mathbf{x}_0\right)$$

$$+\frac{1}{2}\left(\int_{\mathbf{x}_0} p(\mathbf{x}_0|\mathbf{x}_t)\frac{\|\mathbf{y} - \mathcal{A}(\mathbf{x}_0)\|^2}{\sigma_{\mathbf{y}}^2}\mathrm{d}\mathbf{x}_0\right) + \underbrace{\log p(\mathbf{y}) - \log\frac{\sigma_t^n}{\sigma_{\mathbf{y}}^m} - \frac{n-m}{2}\log 2\pi}_{\mathrm{C}}$$

$$D_{\mathrm{KL}}(p(\mathbf{x}_0|\mathbf{x}_t)||p(\mathbf{x}_0|\mathbf{y})) = -\log p(\mathbf{x}_t) - \frac{1}{2\sigma_t^2}\left(\|\mathbf{x}_t\|^2 - 2\mathbf{x}_t^\top \mathrm{D}(\mathbf{x}_t,t) + \int_{\mathbf{x}_0}\|\mathbf{x}_0\|^2 p(\mathbf{x}_0|\mathbf{x}_t)\mathrm{d}\mathbf{x}_0\right)$$

$$+\frac{1}{2\sigma_{\mathbf{y}}^2}\left(-2\mathbf{y}^\top\int_{\mathbf{x}_0}\mathcal{A}(\mathbf{x}_0)p(\mathbf{x}_0|\mathbf{x}_t)\mathrm{d}\mathbf{x}_0 + \int_{\mathbf{x}_0}\|\mathcal{A}(\mathbf{x}_0)\|^2 p(\mathbf{x}_0|\mathbf{x}_t)\mathrm{d}\mathbf{x}_0\right) + \mathrm{C}$$

$$\{\text{By Lemma 1}\}$$

$$D_{\mathrm{KL}}(p(\mathbf{x}_0|\mathbf{x}_t)||p(\mathbf{x}_0|\mathbf{y})) = -\log p(\mathbf{x}_t) - \frac{1}{2\sigma_t^2}\Big(\|\mathbf{x}_t\|^2 - 2\mathbf{x}_t^\top \mathrm{D}(\mathbf{x}_t,t) + \mathrm{Tr}\{\mathrm{Cov}[\mathbf{x}_0|\mathbf{x}_t]\}$$

$$+\|\mathrm{D}(\mathbf{x}_t,t)\|^2\Big) + \frac{1}{2\sigma_{\mathbf{y}}^2}\left(-2\mathbf{y}^\top\int_{\mathbf{x}_0}\mathcal{A}(\mathbf{x}_0)p(\mathbf{x}_0|\mathbf{x}_t)\mathrm{d}\mathbf{x}_0 + \int_{\mathbf{x}_0}\|\mathcal{A}(\mathbf{x}_0)\|^2 p(\mathbf{x}_0|\mathbf{x}_t)\mathrm{d}\mathbf{x}_0\right) + \mathrm{C}$$

$$D_{\mathrm{KL}}(p(\mathbf{x}_0|\mathbf{x}_t)||p(\mathbf{x}_0|\mathbf{y})) = -\log p(\mathbf{x}_t) - \frac{\|\mathrm{D}(\mathbf{x}_t,t) - \mathbf{x}_t\|^2}{2\sigma_t^2} - \frac{1}{2\sigma_t^2}\mathrm{Tr}\{\mathrm{Cov}[\mathbf{x}_0|\mathbf{x}_t]\}$$

$$+\frac{1}{2\sigma_{\mathbf{y}}^2}\left(-2\mathbf{y}^\top\int_{\mathbf{x}_0}\mathcal{A}(\mathbf{x}_0)p(\mathbf{x}_0|\mathbf{x}_t)\mathrm{d}\mathbf{x}_0 + \int_{\mathbf{x}_0}\|\mathcal{A}(\mathbf{x}_0)\|^2 p(\mathbf{x}_0|\mathbf{x}_t)\mathrm{d}\mathbf{x}_0\right) + \mathrm{C}$$

$$\square$$

**Proposition 2.** *The variational mode-seeking-loss (VML) at diffusion time t, for a linear degradation matrix* H, *measurement* $\mathbf{y}$, *and measurement noise variance* $\sigma_{\mathbf{y}}^2$ *is given by*

$$\mathrm{VML} = D_{\mathrm{KL}}(p(\mathbf{x}_0|\mathbf{x}_t)||p(\mathbf{x}_0|\mathbf{y})) = -\log p(\mathbf{x}_t) - \frac{\|\mathrm{D}(\mathbf{x}_t,t) - \mathbf{x}_t\|^2}{2\sigma_t^2} - \frac{1}{2\sigma_t^2}\mathrm{Tr}\{\mathrm{Cov}[\mathbf{x}_0|\mathbf{x}_t]\}$$

$$+\underbrace{\frac{\|\mathbf{y} - \mathrm{HD}(\mathbf{x}_t,t)\|^2}{2\sigma_{\mathbf{y}}^2}}_{\textit{measurement consistency}} + \frac{1}{2\sigma_{\mathbf{y}}^2}\mathrm{Tr}\left\{\mathrm{HCov}[\mathbf{x}_0|\mathbf{x}_t]\mathrm{H}^\top\right\} + \mathrm{C}$$

*where* C *is a constant, independent of* $\mathbf{x}_t$. Tr *denotes the matrix trace,* Cov *denotes the covariance matrix, and* $\mathrm{D}(\cdot,\cdot)$ *denotes the denoiser.*

*Proof.*

Substituting $\mathcal{A}$ with H in Proposition 1

$$D_{\mathrm{KL}}(p(\mathbf{x}_0|\mathbf{x}_t)||p(\mathbf{x}_0|\mathbf{y})) = -\log p(\mathbf{x}_t) - \frac{\|\mathrm{D}(\mathbf{x}_t,t) - \mathbf{x}_t\|^2}{2\sigma_t^2} - \frac{1}{2\sigma_t^2}\mathrm{Tr}\{\mathrm{Cov}[\mathbf{x}_0|\mathbf{x}_t]\}$$

$$+\frac{1}{2\sigma_{\mathbf{y}}^2}\left(-2\mathbf{y}^\top\int_{\mathbf{x}_0}\mathrm{H}\mathbf{x}_0 p(\mathbf{x}_0|\mathbf{x}_t)\mathrm{d}\mathbf{x}_0 + \int_{\mathbf{x}_0}\|\mathrm{H}\mathbf{x}_0\|^2 p(\mathbf{x}_0|\mathbf{x}_t)\mathrm{d}\mathbf{x}_0\right) + \mathrm{C}$$

$$\{\text{By Lemma 2}\}$$

$$D_{\mathrm{KL}}(p(\mathbf{x}_0|\mathbf{x}_t)||p(\mathbf{x}_0|\mathbf{y})) = -\log p(\mathbf{x}_t) - \frac{\|\mathrm{D}(\mathbf{x}_t,t) - \mathbf{x}_t\|^2}{2\sigma_t^2} - \frac{1}{2\sigma_t^2}\mathrm{Tr}\{\mathrm{Cov}[\mathbf{x}_0|\mathbf{x}_t]\}$$

$$+\frac{1}{2\sigma_{\mathbf{y}}^2}\left(-2\mathbf{y}^\top\mathrm{HD}(\mathbf{x}_t,t) + \mathrm{Tr}\left\{\mathrm{HCov}[\mathbf{x}_0|\mathbf{x}_t]\mathrm{H}^\top\right\} + \|\mathrm{HD}(\mathbf{x}_t,t)\|^2\right) + \mathrm{C}$$

$$D_{\mathrm{KL}}(p(\mathbf{x}_0|\mathbf{x}_t)||p(\mathbf{x}_0|\mathbf{y})) = -\log p(\mathbf{x}_t) - \frac{\|\mathrm{D}(\mathbf{x}_t,t) - \mathbf{x}_t\|^2}{2\sigma_t^2} - \frac{1}{2\sigma_t^2}\mathrm{Tr}\{\mathrm{Cov}[\mathbf{x}_0|\mathbf{x}_t]\}$$

$$+\frac{\|\mathbf{y} - \mathrm{HD}(\mathbf{x}_t,t)\|^2}{2\sigma_{\mathbf{y}}^2} + \frac{1}{2\sigma_{\mathbf{y}}^2}\mathrm{Tr}\left\{\mathrm{HCov}[\mathbf{x}_0|\mathbf{x}_t]\mathrm{H}^\top\right\} + \mathrm{C}$$

$$\left\{\text{Note that } \mathrm{C} = \log p(\mathbf{y}) - \log\frac{\sigma_t^n}{\sigma_{\mathbf{y}}^m} - \frac{n-m}{2}\log 2\pi, \text{ see the proof of Proposition 1}\right\}$$

$$\square$$

**Simplified VML gradient.** The gradient of the simplified VML (i.e., $\text{VML}_\text{S}$) for a linear degradation matrix H is given by

$$\text{VML}_\text{S} = -\log p(\mathbf{x}_t) - \frac{\|\text{D}(\mathbf{x}_t, t) - \mathbf{x}_t\|^2}{2\sigma_t^2} + \frac{\|\mathbf{y} - \text{HD}(\mathbf{x}_t, t)\|^2}{2\sigma_\mathbf{y}^2}$$

$$\nabla_{\mathbf{x}_t}\text{VML}_\text{S} = \underbrace{-\frac{\partial \text{D}^\top(\mathbf{x}_t, t)}{\partial \mathbf{x}_t}\frac{\text{H}^\top(\mathbf{y} - \text{HD}(\mathbf{x}_t, t))}{\sigma_\mathbf{y}^2}}_{\textit{measurement consistency gradient}} \underbrace{-\frac{\partial \text{D}^\top(\mathbf{x}_t, t)}{\partial \mathbf{x}_t}\frac{(\text{D}(\mathbf{x}_t, t) - \mathbf{x}_t)}{\sigma_t^2}}_{\textit{prior gradient}}$$

*Proof.*

$$\nabla_{\mathbf{x}_t}\text{VML}_\text{S} = \{-\nabla_{\mathbf{x}_t}\log p(\mathbf{x}_t)\} - \left\{\nabla_{\mathbf{x}_t}\frac{\|\text{D}(\mathbf{x}_t, t) - \mathbf{x}_t\|^2}{2\sigma_t^2}\right\} + \left\{\nabla_{\mathbf{x}_t}\frac{\|\mathbf{y} - \text{HD}(\mathbf{x}_t, t)\|^2}{2\sigma_\mathbf{y}^2}\right\}$$

$$\nabla_{\mathbf{x}_t}\text{VML}_\text{S} = \left\{-\frac{\text{D}(\mathbf{x}_t, t) - \mathbf{x}_t}{\sigma_t^2}\right\} - \left\{\left(\frac{\partial \text{D}^\top(\mathbf{x}_t, t)}{\partial \mathbf{x}_t}\frac{(\text{D}(\mathbf{x}_t, t) - \mathbf{x}_t)}{\sigma_t^2}\right) - \frac{\text{D}(\mathbf{x}_t, t) - \mathbf{x}_t}{\sigma_t^2}\right\}$$

$$+ \left\{-\frac{\partial \text{D}^\top(\mathbf{x}_t, t)}{\partial \mathbf{x}_t}\frac{\text{H}^\top(\mathbf{y} - \text{HD}(\mathbf{x}_t, t))}{\sigma_\mathbf{y}^2}\right\}$$

$$\nabla_{\mathbf{x}_t}\text{VML}_\text{S} = -\frac{\partial \text{D}^\top(\mathbf{x}_t, t)}{\partial \mathbf{x}_t}\frac{\text{H}^\top(\mathbf{y} - \text{HD}(\mathbf{x}_t, t))}{\sigma_\mathbf{y}^2} - \frac{\partial \text{D}^\top(\mathbf{x}_t, t)}{\partial \mathbf{x}_t}\frac{(\text{D}(\mathbf{x}_t, t) - \mathbf{x}_t)}{\sigma_t^2}$$

$\square$

# B    EXCLUDING HIGHER-ORDER TERMS IN THE VML

The true denoiser $\text{D}(\mathbf{x}_t, t) = \mathbb{E}[\mathbf{x}_0 | \mathbf{x}_t]$ is related to the true score function $\nabla_{\mathbf{x}_t}\log p(\mathbf{x}_t)$ by Tweedie's formula $\text{D}(\mathbf{x}_t, t) = \mathbf{x}_t + \sigma_t^2 \nabla_{\mathbf{x}_t}\log p(\mathbf{x}_t)$ (see Appendix A.1). Applying the derivative to this equation gives $\frac{\partial \text{D}(\mathbf{x}_t, t)}{\partial \mathbf{x}_t} = \mathbf{I} + \sigma_t^2 \nabla_{\mathbf{x}_t}^2 \log p(\mathbf{x}_t) = \frac{1}{\sigma_t^2}\text{Cov}[\mathbf{x}_0 | \mathbf{x}_t]$ (see Appendix A.2). With these reformulations, the higher-order terms of VML can be equivalently expressed in terms of $\nabla_{\mathbf{x}_t}\log p(\mathbf{x}_t)$ and $\nabla_{\mathbf{x}_t}^2 \log p(\mathbf{x}_t)$ as follows.

**Reformulating the higher-order terms of VML:** From Proposition 2, the higher-order terms of the VML (involving $\text{Cov}[\mathbf{x}_0 | \mathbf{x}_t]$), denoted by ($\text{VML}_\text{High}$), for a linear degradation matrix H is

$$\text{VML}_\text{High} = -\frac{1}{2\sigma_t^2}\text{Tr}\{\text{Cov}[\mathbf{x}_0 | \mathbf{x}_t]\} + \frac{1}{2\sigma_\mathbf{y}^2}\text{Tr}\{\text{HCov}[\mathbf{x}_0 | \mathbf{x}_t]\text{H}^\top\}$$

where, $\mathbf{x}_t \in \mathbb{R}^n \; \forall \; t \geq 0$, $\mathbf{y} \in \mathbb{R}^m$ and $\text{H} \in \mathbb{R}^{m \times n}$. Reformulating $\text{VML}_\text{High}$ in terms of $\nabla_{\mathbf{x}_t}\log p(\mathbf{x}_t)$, and $\nabla_{\mathbf{x}_t}^2 \log p(\mathbf{x}_t)$ gives

$$\text{VML}_\text{High} = -\frac{\sigma_t^2}{2}\text{Tr}\{\nabla_{\mathbf{x}_t}^2 \log p(\mathbf{x}_t)\} + \frac{\sigma_t^4}{2\sigma_\mathbf{y}^2}\text{Tr}\{\text{H}\nabla_{\mathbf{x}_t}^2 \log p(\mathbf{x}_t)\text{H}^\top\} + \frac{\sigma_t^2}{2\sigma_\mathbf{y}^2}\text{Tr}\{\text{HH}^\top\} + \text{C}_{\text{VML}_\text{High}}$$

where, $\text{C}_{\text{VML}_\text{High}} = -\frac{n}{2}$.

**Proposition 3.** *Let $p_0(\cdot)$ denote the input data distribution and $p_t(\cdot)$ denote the intermediate marginal distributions of a diffusion process for $t > 0$. Let $\exists \; \tau > 0, d > 0$ such that $p_t \in C^2$ (twice continuously differentiable) and $\|\mathbf{x}\| \leq d \; \forall \; t < \tau$ (i.e., $\forall \; t < \tau$, $\mathbf{x}$ lies in a compact ball, and $p_t \in C^2$). The function $\text{VML}_{\text{High}_t}(\mathbf{x})$ denoting the higher-order terms (involving $\text{Cov}[\mathbf{x}_0 | \mathbf{x}_t]$) of VML, for a linear degradation operator matrix H, measurement $\mathbf{y}$, and measurement noise variance $\sigma_\mathbf{y}^2$ converges uniformly to $\text{C}_{\text{VML}_\text{High}}$ in the limit as $t \to 0$. (Note that $\mathbf{x}_t \in \mathbb{R}^n \; \forall t \geq 0$, $\mathbf{y} \in \mathbb{R}^m$ and $\text{C}_{\text{VML}_\text{High}} = -\frac{n}{2}$ as previously mentioned)*

$$\text{VML}_{\text{High}_t}(\mathbf{x}) = -\frac{\sigma_t^2}{2}\text{Tr}\{\nabla_\mathbf{x}^2 \log p_t(\mathbf{x})\} + \frac{\sigma_t^4}{2\sigma_\mathbf{y}^2}\text{Tr}\{\text{H}\nabla_\mathbf{x}^2 \log p_t(\mathbf{x})\text{H}^\top\}$$

$$+ \frac{\sigma_t^2}{2\sigma_\mathbf{y}^2}\text{Tr}\{\text{HH}^\top\} + \text{C}_{\text{VML}_\text{High}}$$

*and,* $\underset{t \to 0}{\text{unif lim}}\, \text{VML}_{\text{High}_t}(\mathbf{x}) = \text{C}_{\text{VML}_\text{High}} \quad \forall \; \mathbf{x} \; s.t. \; \|\mathbf{x}\| \leq d$

*proof sketch.* For $t < \tau$, $p_t \in C^2$ implies $\nabla_{\mathbf{x}}^2 \log p_t(\mathbf{x})$ is continuous, which further implies continuity of its component functions, i.e., $\frac{\partial^2 \log p_t(\mathbf{x})}{\partial \mathbf{x}^{(i)} \partial \mathbf{x}^{(j)}} \; \forall \; i,j \; \in \; [1, 2, \ldots, n]$. Since $\mathbf{x}$ lies in a compact ball (i.e., $\|\mathbf{x}\| \leq d$), it implies that the component functions are bounded $\forall \; t < \tau$, which further implies boundedness of $\mathrm{Tr}\left\{\nabla_{\mathbf{x}}^2 \log p_t(\mathbf{x})\right\}$, and $\mathrm{Tr}\left\{\mathrm{H}\nabla_{\mathbf{x}}^2 \log p_t(\mathbf{x})\mathrm{H}^\top\right\}$. Note that $\mathrm{Tr}\left\{\mathrm{HH}^\top\right\}$ is also bounded. With $\sigma_t \to 0$, as $t \to 0$, it is apparent that $\mathrm{VML}_{\mathrm{High}_t}$ converges uniformly (since the bounds are global and hold for all $\mathbf{x}$ s.t $\|\mathbf{x}\| \leq d$) to $\mathrm{C}_{\mathrm{VML}_{\mathrm{High}}}$. $\qquad\square$

**Reformulating the VML:** From Proposition 2, the VML for a linear degradation matrix H is

$$\mathrm{VML} = D_{\mathrm{KL}}(p(\mathbf{x}_0|\mathbf{x}_t)||p(\mathbf{x}_0|\mathbf{y})) = -\log p(\mathbf{x}_t) - \frac{\|\mathrm{D}(\mathbf{x}_t, t) - \mathbf{x}_t\|^2}{2\sigma_t^2} - \frac{1}{2\sigma_t^2}\mathrm{Tr}\left\{\mathrm{Cov}[\mathbf{x}_0|\mathbf{x}_t]\right\}$$

$$+ \underbrace{\frac{\|\mathbf{y} - \mathrm{HD}(\mathbf{x}_t, t)\|^2}{2\sigma_{\mathbf{y}}^2}}_{\text{measurement consistency}} + \frac{1}{2\sigma_{\mathbf{y}}^2}\mathrm{Tr}\left\{\mathrm{HCov}[\mathbf{x}_0|\mathbf{x}_t]\mathrm{H}^\top\right\} + \mathrm{C}$$

where, $\mathbf{x}_t \in \mathbb{R}^n \; \forall \; t \geq 0$, $\mathbf{y} \in \mathbb{R}^m$ and $\mathrm{C} = \log p(\mathbf{y}) - \log \frac{\sigma_t^n}{\sigma_{\mathbf{y}}^m} - \frac{n-m}{2} \log 2\pi$ (see the proof of Proposition 2). Reformulating VML in terms of $\nabla_{\mathbf{x}_t} \log p(\mathbf{x}_t)$, and $\nabla_{\mathbf{x}_t}^2 \log p(\mathbf{x}_t)$ gives

$$\mathrm{VML} = D_{\mathrm{KL}}(p(\mathbf{x}_0|\mathbf{x}_t)||p(\mathbf{x}_0|\mathbf{y})) = -\log p(\mathbf{x}_t) - \frac{\sigma_t^2}{2}\|\nabla_{\mathbf{x}_t} \log p(\mathbf{x}_t)\|^2 - \frac{\sigma_t^2}{2}\mathrm{Tr}\left\{\nabla_{\mathbf{x}_t}^2 \log p(\mathbf{x}_t)\right\}$$

$$+ \underbrace{\frac{\|\mathbf{y} - \mathrm{HD}(\mathbf{x}_t, t)\|^2}{2\sigma_{\mathbf{y}}^2}}_{\text{measurement consistency}} + \frac{\sigma_t^4}{2\sigma_{\mathbf{y}}^2}\mathrm{Tr}\left\{\mathrm{H}\nabla_{\mathbf{x}_t}^2 \log p(\mathbf{x}_t)\mathrm{H}^\top\right\} + \frac{\sigma_t^2}{2\sigma_{\mathbf{y}}^2}\mathrm{Tr}\left\{\mathrm{HH}^\top\right\} + \log p(\mathbf{y}) - n\log \sigma_t + \mathrm{C}_{\mathrm{VML}}$$

where, $\mathrm{C}_{\mathrm{VML}} = -\frac{n}{2} + m\log \sigma_{\mathbf{y}} - \frac{n-m}{2}\log 2\pi$.

**Proposition 4.** *Let $p_0(\cdot)$ denote the input data distribution and $p_t(\cdot)$ denote the intermediate marginal distributions of a diffusion process for $t > 0$. Let $\exists \; \tau > 0, d > 0$, such that $p_t \in C^2$ (twice continuously differentiable) and $\|\mathbf{x}\| \leq d \; \forall \; t < \tau$ (In other words, $\exists \; \tau > 0, d > 0$, such that $p_t \in C^2$, and $\|\mathbf{x}_t\| \leq d \; \forall t < \tau$ where $\mathbf{x}_t \in \mathcal{M}_t$ i.e., the intermediate diffusion manifold at time t). Assuming sufficient conditions for $\lim_{t \to 0} \log p_t(\mathbf{x}) = \log p_0(\mathbf{x})$, the function $\mathrm{VML}_t(\mathbf{x}) + n\log \sigma_t$, for a linear degradation operator matrix H, measurement $\mathbf{y}$, and measurement noise variance $\sigma_{\mathbf{y}}^2$ converges pointwise to $-\log p_0(\mathbf{x}|\mathbf{y}) + \hat{\mathrm{C}}_{\mathrm{VML}}$ in the limit as $t \to 0$. (Note that $\mathbf{x}_t \in \mathbb{R}^n \; \forall t \geq 0$, $\mathbf{y} \in \mathbb{R}^m$ and $\hat{\mathrm{C}}_{\mathrm{VML}} = -\frac{n}{2} - \frac{n}{2}\log 2\pi$)*

$$\mathrm{VML}_t(\mathbf{x}) = -\log p_t(\mathbf{x}) - \frac{\sigma_t^2}{2}\|\nabla_{\mathbf{x}} \log p_t(\mathbf{x})\|^2 - \frac{\sigma_t^2}{2}\mathrm{Tr}\left\{\nabla_{\mathbf{x}}^2 \log p_t(\mathbf{x})\right\} + \frac{\|\mathbf{y} - \mathrm{HD}(\mathbf{x}, t)\|^2}{2\sigma_{\mathbf{y}}^2}$$

$$+ \frac{\sigma_t^4}{2\sigma_{\mathbf{y}}^2}\mathrm{Tr}\left\{\mathrm{H}\nabla_{\mathbf{x}}^2 \log p_t(\mathbf{x})\mathrm{H}^\top\right\} + \frac{\sigma_t^2}{2\sigma_{\mathbf{y}}^2}\mathrm{Tr}\left\{\mathrm{HH}^\top\right\} + \log p(\mathbf{y}) - n\log \sigma_t + \mathrm{C}_{\mathrm{VML}}$$

$$\textit{and, } \lim_{t \to 0} \mathrm{VML}_t(\mathbf{x}) + n\log \sigma_t = -\log p_0(\mathbf{x}|\mathbf{y}) + \hat{\mathrm{C}}_{\mathrm{VML}} \; \forall \; \mathbf{x} \; s.t. \; \|\mathbf{x}\| \leq d$$

*proof sketch.* It suffices to show that $\lim_{t \to 0} \left\{\mathrm{VML}_t(\mathbf{x}) + n\log \sigma_t + \log p_0(\mathbf{x}|\mathbf{y}) - \hat{\mathrm{C}}_{\mathrm{VML}}\right\} = 0 \; \forall \; \mathbf{x} \; s.t. \; \|\mathbf{x}\| \leq d$.

$$\mathrm{VML}_t(\mathbf{x}) + n\log \sigma_t + \log p_0(\mathbf{x}|\mathbf{y}) - \hat{\mathrm{C}}_{\mathrm{VML}}$$

$$= \mathrm{VML}_t(\mathbf{x}) + n\log \sigma_t + \log p_0(\mathbf{y}|\mathbf{x}) + \log p_0(\mathbf{x}) - \log p(\mathbf{y}) - \mathrm{C}_{\mathrm{VML}} + \left\{m\log \sigma_{\mathbf{y}} + \frac{m}{2}\log 2\pi\right\}$$

$$= \underbrace{\left\{-\frac{\sigma_t^2}{2}\|\nabla_{\mathbf{x}} \log p_t(\mathbf{x})\|^2 - \frac{\sigma_t^2}{2}\mathrm{Tr}\left\{\nabla_{\mathbf{x}}^2 \log p_t(\mathbf{x})\right\} + \frac{\sigma_t^4}{2\sigma_{\mathbf{y}}^2}\mathrm{Tr}\left\{\mathrm{H}\nabla_{\mathbf{x}}^2 \log p_t(\mathbf{x})\mathrm{H}^\top\right\} + \frac{\sigma_t^2}{2\sigma_{\mathbf{y}}^2}\mathrm{Tr}\left\{\mathrm{HH}^\top\right\}\right\}}_{T_A}$$

$$+ \underbrace{\left\{-\log p_t(\mathbf{x}) + \log p_0(\mathbf{x})\right\}}_{T_B} + \underbrace{\left\{m\log \sigma_y + \frac{m}{2}\log 2\pi + \frac{\|\mathbf{y} - \mathrm{HD}(\mathbf{x}, t)\|^2}{2\sigma_{\mathbf{y}}^2} + \log p_0(\mathbf{y}|\mathbf{x})\right\}}_{T_C}$$

To show that $\lim_{t \to 0}\{\mathrm{VML}_t(\mathbf{x}) + n\log \sigma_t + \log p_0(\mathbf{x}|\mathbf{y}) - \hat{\mathrm{C}}_{\mathrm{VML}}\} = 0 \; \forall \; \mathbf{x} \; s.t. \; \|\mathbf{x}\| \leq d$, we need to show that $\lim_{t \to 0} T_A = \lim_{t \to 0} T_B = \lim_{t \to 0} T_C = 0$. Under sufficient conditions assumed for

$\lim_{t \to 0} \log p_t = \log p_0$, it implies that $\lim_{t \to 0} T_B = 0$. Considering $T_A$: as $p_t(\mathbf{x}) \in C^2$ and $\mathbf{x}$ lies in a compact set, it implies that $\nabla_{\mathbf{x}} \log p_t(\mathbf{x})$ and $\nabla_{\mathbf{x}}^2 \log p_t(\mathbf{x})$ are bounded for all $t < \tau$. With $\sigma_t \to 0$ as $t \to 0$, $\lim_{t \to 0} T_A = 0$. Considering $T_C$: Note that $p_0(\mathbf{y}|\mathbf{x}) = \mathcal{N}(\mathbf{Hx}, \sigma_{\mathbf{y}}^2 \mathbf{I})$ (see Equation 1) and $\log p_0(\mathbf{y}|\mathbf{x}) = -m \log \sigma_y - \frac{m}{2} \log 2\pi - \frac{\|\mathbf{y} - \mathbf{Hx}\|^2}{2\sigma_{\mathbf{y}}^2}$. It can be seen that, as $t \to 0$, $\mathrm{D}(\mathbf{x}, t) \to \mathbf{x}$, since $\mathrm{D}(\mathbf{x}, t) = \mathbf{x} + \sigma_t^2 \nabla_{\mathbf{x}} \log p_t(\mathbf{x})$ (Appendix A.1). This further implies $\lim_{t \to 0} T_c = 0$. $\qquad \square$

**Reformulating the Simplified-VML:** From Equation (5), the Simplified-VML ($\mathrm{VML_S}$) for a linear degradation matrix H is

$$\mathrm{VML_S} = -\log p(\mathbf{x}_t) - \frac{\|\mathrm{D}(\mathbf{x}_t, t) - \mathbf{x}_t\|^2}{2\sigma_t^2} + \underbrace{\frac{\|\mathbf{y} - \mathrm{HD}(\mathbf{x}_t, t)\|^2}{2\sigma_{\mathbf{y}}^2}}_{\text{measurement consistency}} + \mathrm{C}$$

where, $\mathbf{x}_t \in \mathbb{R}^n \ \forall \ t \geq 0$, $\mathbf{y} \in \mathbb{R}^m$ and $\mathrm{C} = \log p(\mathbf{y}) - \log \frac{\sigma_t^n}{\sigma_{\mathbf{y}}^m} - \frac{n-m}{2} \log 2\pi$. Reformulating $\mathrm{VML_S}$ in terms of $\nabla_{\mathbf{x}_t} \log p(\mathbf{x}_t)$, and $\nabla_{\mathbf{x}_t}^2 \log p(\mathbf{x}_t)$ gives

$$\mathrm{VML_S} = -\log p(\mathbf{x}_t) - \frac{\sigma_t^2}{2} \|\nabla_{\mathbf{x}_t} \log p(\mathbf{x}_t)\|^2 + \underbrace{\frac{\|\mathbf{y} - \mathrm{HD}(\mathbf{x}_t, t)\|^2}{2\sigma_{\mathbf{y}}^2}}_{\text{measurement consistency}} + \log p(\mathbf{y}) - n \log \sigma_t + \mathrm{C_{VML_S}}$$

where, $\mathrm{C_{VML_S}} = m \log \sigma_{\mathbf{y}} - \frac{n-m}{2} \log 2\pi$.

**Proposition 5.** *Let $p_0(\cdot)$ denote the input data distribution and $p_t(\cdot)$ denote the intermediate marginal distributions of a diffusion process for $t > 0$. Let $\exists \ \tau > 0, d > 0$ such that $p_t \in C^1$ (once continuously differentiable) and $\|\mathbf{x}\| \leq d \ \forall \ t < \tau$ (i.e., $\forall \ t < \tau$, $\mathbf{x}$ lies in a compact ball, and $p_t \in C^1$). Assuming sufficient conditions for $\lim_{t \to 0} \log p_t(\mathbf{x}) = \log p_0(\mathbf{x}) \ \forall \ \mathbf{x}$, the function given by $\mathrm{VML_{S_t}}(\mathbf{x}) + n \log \sigma_t$, (where $\mathrm{VML_{S_t}}(\mathbf{x})$ denotes the Simplified-VML) for a linear degradation operator matrix H, measurement $\mathbf{y}$, and measurement noise variance $\sigma_{\mathbf{y}}^2$ converges pointwise to $-\log p_0(\mathbf{x}|\mathbf{y}) + \hat{\mathrm{C}}_{\mathrm{VML_S}}$ in the limit as $t \to 0$. (Note that $\mathbf{x}_t \in \mathbb{R}^n \ \forall t \geq 0$, $\mathbf{y} \in \mathbb{R}^m$ and $\hat{\mathrm{C}}_{\mathrm{VML_S}} = -\frac{n}{2} \log 2\pi$)*

$$\mathrm{VML_{S_t}}(\mathbf{x}) = -\log p_t(\mathbf{x}) - \frac{\sigma_t^2}{2} \|\nabla_{\mathbf{x}} \log p_t(\mathbf{x})\|^2 + \frac{\|\mathbf{y} - \mathrm{HD}(\mathbf{x}, t)\|^2}{2\sigma_{\mathbf{y}}^2} + \log p(\mathbf{y}) - n \log \sigma_t + \mathrm{C_{VML_S}}$$

*and,* $\lim_{t \to 0} \mathrm{VML_{S_t}}(\mathbf{x}) + n \log \sigma_t = -\log p_0(\mathbf{x}|\mathbf{y}) + \hat{\mathrm{C}}_{\mathrm{VML_S}} \ \forall \ \mathbf{x} \ s.t. \ \|\mathbf{x}\| \leq d$

*proof sketch.* By arguments similar to those in the proof of Proposition 4 $\qquad \square$

**Remark 1.** *Note that the limit of $\mathrm{VML}_t(\mathbf{x})$ as $t \to 0$ doesn't exist. However, for a given arbitrary $t$, a global minimizer of $\mathrm{VML}_t(\mathbf{x})$ is also a global minimizer of $\mathrm{VML}_t(\mathbf{x}) + n \log \sigma_t$ (for $n \log \sigma_t$ is a constant given $t$) and vice-versa. From Proposition 4, $\mathrm{VML}_t(\mathbf{x}) + n \log \sigma_t$ converges pointwise to $-\log p_0(\mathbf{x}|\mathbf{y}) + \hat{\mathrm{C}}_{\mathrm{VML}}$ in the limit as $t \to 0$.*

**Remark 2.** *Note that the limit of $\mathrm{VML_{S_t}}(\mathbf{x})$ as $t \to 0$ doesn't exist. However, for a given arbitrary $t$, a global minimizer of $\mathrm{VML_{S_t}}(\mathbf{x})$ is also a global minimizer of $\mathrm{VML_{S_t}}(\mathbf{x}) + n \log \sigma_t + \mathrm{C_{VML_{High}}}$ (for $n \log \sigma_t + \mathrm{C_{VML_{High}}}$ is a constant given $t$) and vice-versa. From Proposition 5, $\mathrm{VML_{S_t}}(\mathbf{x}) + n \log \sigma_t + \mathrm{C_{VML_{High}}}$ converges pointwise to $-\log p_0(\mathbf{x}|\mathbf{y}) + \hat{\mathrm{C}}_{\mathrm{VML_S}} + \mathrm{C_{VML_{High}}} = -\log p_0(\mathbf{x}|\mathbf{y}) + \hat{\mathrm{C}}_{\mathrm{VML}}$ in the limit as $t \to 0$.*

**Remark 3.** *From Proposition 3, the function $(\mathrm{VML_{High_t}} - \mathrm{C_{VML_{High}}})$ converges uniformly to the zero function in the limit as $t \to 0$. Note that $\mathrm{VML_{High_t}} - \mathrm{C_{VML_{High}}} = (\mathrm{VML}_t + n \log \sigma_t) - (\mathrm{VML_{S_t}} + n \log \sigma_t + \mathrm{C_{VML_{High}}})$, i.e., the difference of essentially equivalent (in terms of global minimizers) functions of VML and Simplified-VML respectively (see Remarks 1 and 2). It implies that the difference of these functions becomes arbitrarily small as $t \to 0$. In practice, this approximation of VML with Simplified-VML may not be critical, as the errors arising due to the imperfect optimizer and numerical errors from discretizing the reverse SDE or PF ODE typically dominate early in the reverse diffusion process.*

# C  EXPERIMENTAL SETUP, IMPLEMENTATION DETAILS, QUALITATIVE VISUALIZATIONS, AND MORE

## C.1  EXPERIMENTS IN TABLE 1 AND TABLE 2

In these experiments, we considered image restoration inverse problems with severe enough degradations to make it more challenging. However, we do not resort to extreme degradations, as the corresponding measurements typically do not provide strong guidance for recovering the ground truth image, since extreme degradations make the posterior highly multimodal to an extent that the restored image is perceptually dissimilar to the ground truth image, which makes it challenging to assess the performance using the usual LPIPS/FID metrics. Our experiments included half-mask inpainting, $4\times$ super-resolution, and uniform deblurring with a $16 \times 16$ kernel. We utilize the SVD-based super-resolution and uniform deblurring operators from Kawar et al. (2022) to ensure that the preconditioner can be computed efficiently. For uniform deblurring, we observed that the degradations are not severe enough, as the pseudoinverse solution already gives an almost perfect reconstruction. To make it more challenging, we zero out the singular values below a high enough threshold (0.2) as opposed to the default threshold (0.03) used in Kawar et al. (2022).

Table 4: Best learning rate configuration for experiments in Table 1 and Table 2. Note that the learning rate is $\gamma_0 \cdot \sigma_y^2$, with $\gamma_0$ as reported in the table.

| Dataset | Method | Inpainting | $4\times$ Super-res | Deblurring |
|---|---|---|---|---|
| ImageNet64 | VML-MAP | $\gamma_0 = 1.5$ | $\gamma_0 = 30.0$ | $\gamma_0 = 2.25$ |
| | VML-MAP$_{pre}$ | $\gamma_0 = 1.5$ | $\gamma_0 = 1.75$ | $\gamma_0 = 2.0$ |
| ImageNet256 | VML-MAP | $\gamma_0 = 1.25$ | $\gamma_0 = 25.0$ | $\gamma_0 = 2.0$ |
| | VML-MAP$_{pre}$ | $\gamma_0 = 1.25$ | $\gamma_0 = 1.5$ | $\gamma_0 = 1.5$ |
| FFHQ256 | VML-MAP | $\gamma_0 = 1.25$ | $\gamma_0 = 30.0$ | $\gamma_0 = 2.0$ |
| | VML-MAP$_{pre}$ | $\gamma_0 = 1.25$ | $\gamma_0 = 1.5$ | $\gamma_0 = 1.5$ |

For DDRM, ΠGDM, MAPGA, VML-MAP, and VML-MAP$_{pre}$, we use the EDM noise schedule from Karras et al. (2022), with $\sigma_{min} = 0.002$, $\sigma_{max} = 140$, and $\rho = 7$. Note that MAPGA requires a consistency model by default, so throughout this paper, we use the variant MAPGA(D) from Gutha et al. (2025), which replaces the consistency model with a single-step denoiser approximation. We use the EDM schedule for DDRM and ΠGDM as it performs the best compared to the default schedules used in their original repositories. For DAPS, we use the default DAPS-1K and DAPS-4K configurations mentioned in the original paper, and observed that the default hyperparameters used in the paper for box inpainting, super-resolution, and Gaussian deblurring also perform the best for our half-mask inpainting, super-resolution, and uniform deblurring tasks.

For each experiment with DDRM and ΠGDM, we search for $N$ (i.e., the number of reverse diffusion steps) over $\{20.50, 100, 200, 500, 1000\}$ and report the best result. For each experiment using MAPGA, VML-MAP, and VML-MAP$_{pre}$, we search for $(N, K)$ ($N$ denotes the total number of diffusion time steps, and $K$ denotes the number of gradient ascent/descent iterations per step) over $\{(20, 50), (50, 20), (100, 10), (200, 5), (500, 2), (1000, 1)\}$ and report the best performance (this keeps the total budget for MAP-GA, VML-MAP and VML-MAP$_{pre}$ within 1000 optimization steps in total). In every case, we find the best configuration to be $(N, K)$=(20, 50). We set $\sigma_y = 0$ for DDRM, ΠGDM, and MAPGA, while for DAPS, VML-MAP and VML-MAP$_{pre}$, we set $\sigma_y = 1e$-9. For MAPGA, the default learning rate from the original repository was used, while for VML-MAP and VML-MAP$_{pre}$, we report the best learning rate configuration for each task as $\gamma_0 \cdot \sigma_y^2$, with $\gamma_0$ shown in Table 4. Our implementation of DDRM, ΠGDM, and MAPGA is based on the following original repositories ddrm, pgdm, mapga, respectively.

## C.2 EXPERIMENTS IN TABLE 3

In these experiments, we focused on the image inpainting task with different types of masks. As mentioned in Sections 4.2 and 5, DDRM, ΠGDM, MAPGA, and VML-MAP$_{pre}$ require the SVD of H, while DAPS and VML-MAP only require the forward operation of H. For inpainting, H is a diagonal matrix with zeros for indices corresponding to masked pixels, and ones for indices corresponding to observed pixels. The SVD of H in this case is trivial, since H itself is the singular value matrix, with the left and the right singular matrices being identity. This ensures a fair comparison among all the methods, irrespective of whether a method requires the SVD of H or not.

Different types of inpainting masks that were considered in these experiments include

- **Expand mask:** Pixels outside the $128 \times 128$ square center-crop are masked
- **Box mask:** Pixels within the $128 \times 128$ square center-crop are masked
- **Super-resolution mask:** Alternative pixels are masked
- **Random mask:** 70% of the pixels are randomly masked

We use $\gamma_0 = 1.0$ (i.e., a learning rate of $1.0 \cdot \sigma_y^2$) for all four tasks on both ImageNet256 and FFHQ256, and follow the same settings in Appendix C.1 for other hyperparameter configurations.

## C.3 QUALITATIVE VISUALIZATIONS

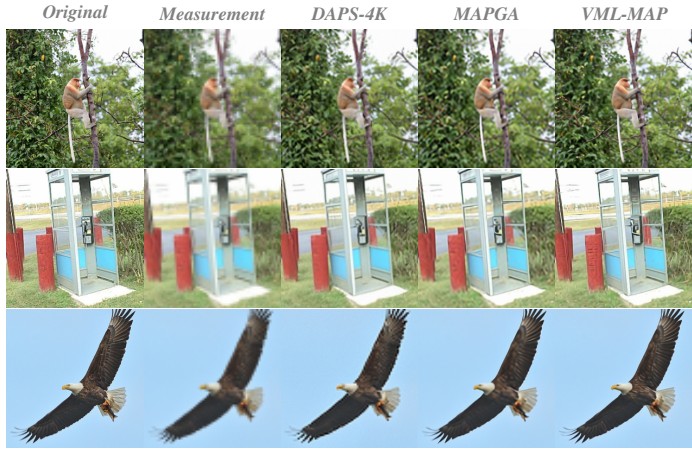

Figure 8: $4\times$ Super-resolution task from Table 2. Zoom in for the best view.

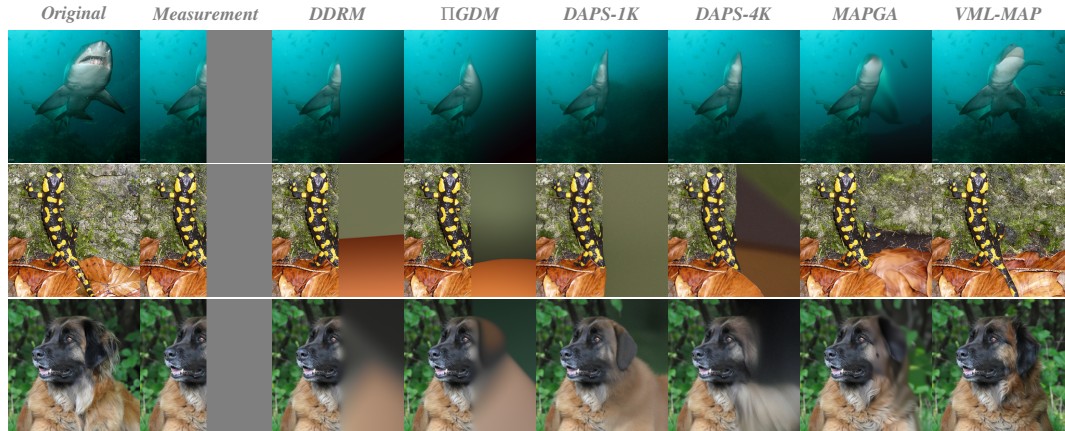

Figure 9: Half-mask inpainting task from Table 2. Zoom in for the best view.

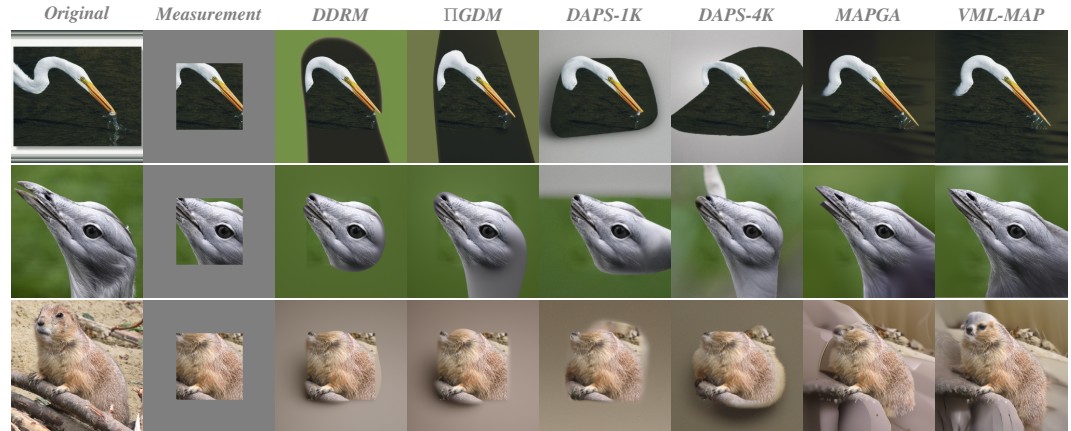

Figure 10: Expand-mask inpainting task from Table 3. Zoom in for the best view.

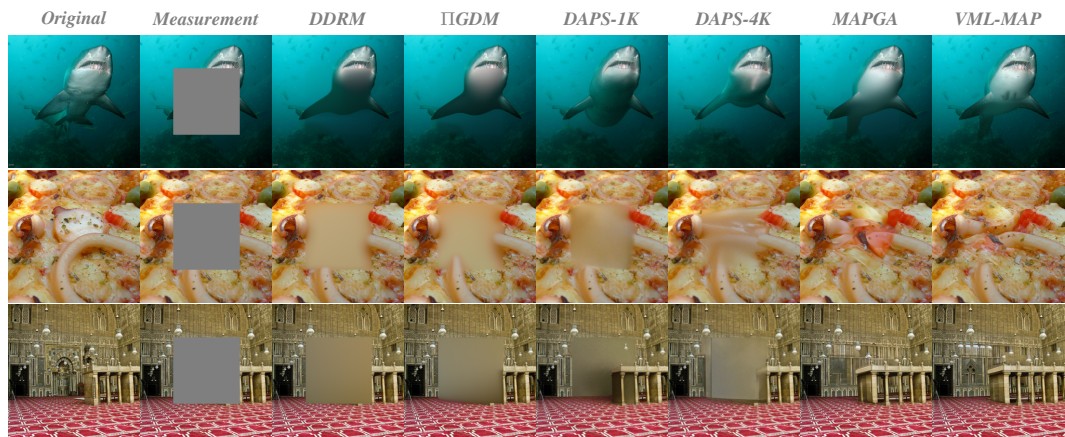

Figure 11: Box-mask inpainting task from Table 3. Zoom in for the best view.

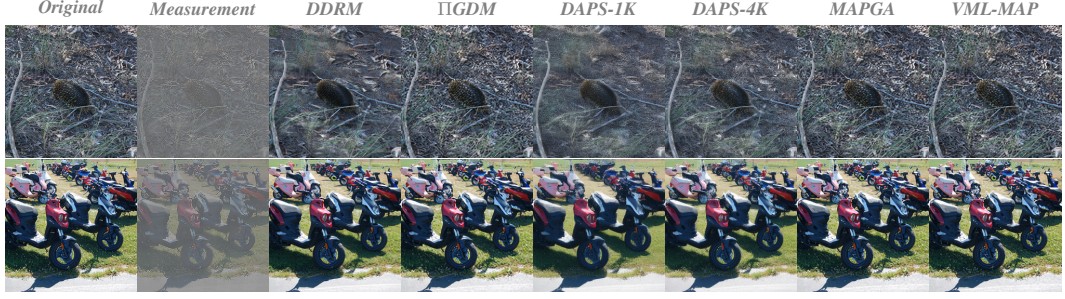

Figure 12: Super-res-mask inpainting task from Table 3. Zoom in for the best view.

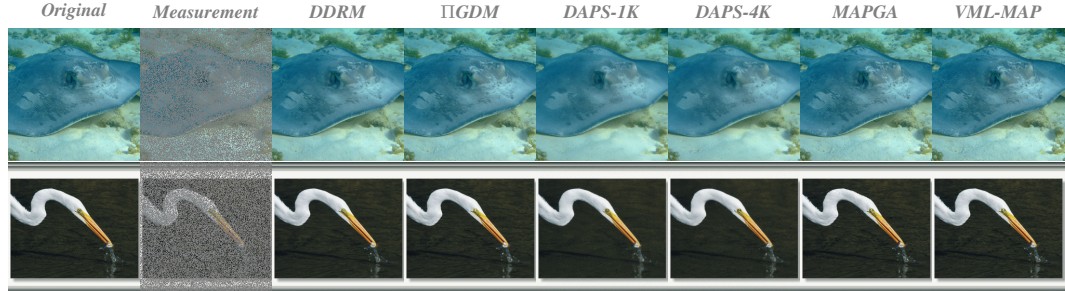

Figure 13: Random-mask inpainting task from Table 3. Zoom in for the best view.

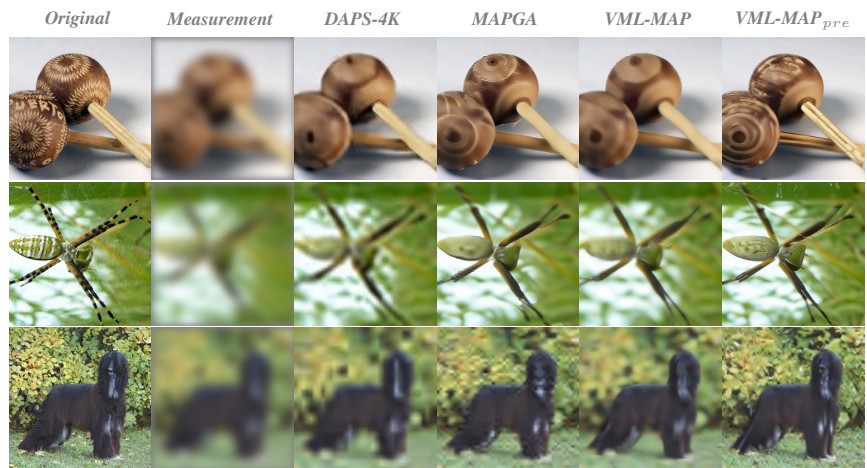

| Original | Measurement | DAPS-4K | MAPGA | VML-MAP | VML-MAP$_{pre}$ |

Figure 14: Deblurring task from Table 2. Zoom in for the best view.

## C.4 Experiments across different seeds

Table 5: Experiments from Table 1 (i.e., Half-mask inpainting, $4\times$ Super-resolution, and Uniform deblurring on 1000 images of ImageNet64 validation set) are repeated across 4 different seeds. The mean and standard deviation across these runs are reported.

| Method | Inpainting | | $4\times$ Super-res | | Deblurring | |
|---|---|---|---|---|---|---|
| | LPIPS↓ | FID↓ | LPIPS↓ | FID↓ | LPIPS↓ | FID↓ |
| DDRM | $0.263_{\pm0.000}$ | $57.15_{\pm0.241}$ | $0.234_{\pm0.000}$ | $77.82_{\pm0.526}$ | $0.467_{\pm0.001}$ | $197.7_{\pm0.600}$ |
| ΠGDM | $0.242_{\pm0.000}$ | $54.69_{\pm0.334}$ | $0.241_{\pm0.000}$ | $88.63_{\pm0.472}$ | $0.439_{\pm0.001}$ | $164.7_{\pm1.098}$ |
| MAPGA | $0.172_{\pm0.000}$ | $46.43_{\pm0.150}$ | $0.204_{\pm0.000}$ | $84.72_{\pm0.779}$ | $\underline{0.322}_{\pm0.001}$ | $113.9_{\pm0.753}$ |
| **VML-MAP** | $\mathbf{0.146}_{\pm0.000}$ | $\mathbf{38.84}_{\pm0.233}$ | $\underline{0.136}_{\pm0.001}$ | $62.19_{\pm0.298}$ | $0.356_{\pm0.001}$ | $\underline{106.6}_{\pm0.682}$ |
| **VML-MAP$_{pre}$** | $\mathbf{0.146}_{\pm0.000}$ | $\mathbf{38.84}_{\pm0.233}$ | $\mathbf{0.129}_{\pm0.000}$ | $\mathbf{60.01}_{\pm0.400}$ | $\mathbf{0.266}_{\pm0.001}$ | $\mathbf{77.46}_{\pm0.424}$ |

Table 6: Experiments from Table 2 (i.e., Half-mask inpainting, $4\times$ Super-resolution, and Uniform deblurring on 100 validation images of ImageNet256 and 100 images of FFHQ256) are repeated across 4 different seeds. The mean and standard deviation across these runs are reported.

| Dataset | Method | Inpainting | | $4\times$ Super-res | | Deblurring | |
|---|---|---|---|---|---|---|---|
| | | LPIPS↓ | FID↓ | LPIPS↓ | FID↓ | LPIPS↓ | FID↓ |
| ImageNet | DDRM | $0.393_{\pm0.001}$ | $103.5_{\pm0.657}$ | $0.289_{\pm0.000}$ | $89.40_{\pm0.962}$ | $0.618_{\pm0.000}$ | $233.4_{\pm2.033}$ |
| | ΠGDM | $0.373_{\pm0.000}$ | $103.7_{\pm0.694}$ | $0.292_{\pm0.001}$ | $83.88_{\pm0.226}$ | $0.562_{\pm0.000}$ | $231.8_{\pm0.631}$ |
| | MAPGA | $\underline{0.290}_{\pm0.001}$ | $\underline{80.76}_{\pm2.013}$ | $0.273_{\pm0.001}$ | $76.12_{\pm0.587}$ | $\underline{0.459}_{\pm0.002}$ | $\underline{194.9}_{\pm1.548}$ |
| | DAPS-1K | $0.384_{\pm0.001}$ | $98.45_{\pm1.879}$ | $0.254_{\pm0.001}$ | $71.60_{\pm1.010}$ | $0.605_{\pm0.001}$ | $217.8_{\pm3.042}$ |
| | DAPS-4K | $0.365_{\pm0.004}$ | $93.62_{\pm0.856}$ | $0.244_{\pm0.000}$ | $69.71_{\pm1.554}$ | $0.593_{\pm0.001}$ | $227.9_{\pm2.991}$ |
| | **VML-MAP** | $\mathbf{0.262}_{\pm0.002}$ | $\mathbf{74.21}_{\pm3.209}$ | $\mathbf{0.194}_{\pm0.001}$ | $\underline{60.08}_{\pm0.687}$ | $0.509_{\pm0.003}$ | $200.4_{\pm2.812}$ |
| | **VML-MAP$_{pre}$** | $\mathbf{0.262}_{\pm0.002}$ | $\mathbf{74.21}_{\pm3.209}$ | $\underline{0.196}_{\pm0.003}$ | $\mathbf{58.60}_{\pm2.076}$ | $\mathbf{0.367}_{\pm0.002}$ | $\mathbf{165.2}_{\pm2.037}$ |
| FFHQ | DDRM | $0.246_{\pm0.001}$ | $71.23_{\pm0.521}$ | $0.154_{\pm0.000}$ | $70.07_{\pm0.495}$ | $0.307_{\pm0.000}$ | $117.9_{\pm0.427}$ |
| | ΠGDM | $0.237_{\pm0.001}$ | $70.40_{\pm0.494}$ | $0.147_{\pm0.000}$ | $68.38_{\pm0.605}$ | $0.293_{\pm0.000}$ | $114.0_{\pm0.302}$ |
| | MAPGA | $\underline{0.206}_{\pm0.000}$ | $64.15_{\pm0.660}$ | $0.132_{\pm0.000}$ | $63.82_{\pm0.455}$ | $0.235_{\pm0.000}$ | $119.2_{\pm1.045}$ |
| | DAPS-1K | $0.233_{\pm0.001}$ | $61.22_{\pm0.759}$ | $0.113_{\pm0.000}$ | $60.89_{\pm1.028}$ | $0.260_{\pm0.000}$ | $100.8_{\pm0.951}$ |
| | DAPS-4K | $0.224_{\pm0.000}$ | $\underline{60.22}_{\pm0.738}$ | $\underline{0.100}_{\pm0.000}$ | $58.29_{\pm0.398}$ | $\underline{0.230}_{\pm0.001}$ | $\underline{93.71}_{\pm1.217}$ |
| | **VML-MAP** | $\mathbf{0.180}_{\pm0.001}$ | $\mathbf{52.76}_{\pm0.526}$ | $\mathbf{0.100}_{\pm0.000}$ | $\underline{57.55}_{\pm1.414}$ | $0.247_{\pm0.000}$ | $99.40_{\pm1.799}$ |
| | **VML-MAP$_{pre}$** | $\mathbf{0.180}_{\pm0.001}$ | $\mathbf{52.76}_{\pm0.526}$ | $\mathbf{0.100}_{\pm0.000}$ | $\mathbf{52.20}_{\pm0.814}$ | $\mathbf{0.182}_{\pm0.000}$ | $\mathbf{93.48}_{\pm0.681}$ |

Here, we repeat the experiments from Table 1 and Table 2 across 4 different seeds and report the mean and standard deviation of LPIPS and FID. The variation of these quantitative metrics across different seeds is quite insignificant (see Table 5 and Table 6), which also reinforces the validity of the original conclusions drawn from Table 1 and Table 2 in Section 4 and Section 5, respectively.

While the quantitative metrics are not sensitive to different seeds, the qualitative results do show variations across seeds. Note that VML-MAP involves optimization of VML (using gradient descent) during each reverse diffusion step (see Figure 2, and Algorithm 1). Since gradient descent is not a perfect optimizer in practice, the results of the optimization can widely differ based on the initialization $\mathbf{x}_T$ and also the stochastic components within VML-MAP (such as the sampling operation for renoising after each optimization block). Since the random seed directly influences this, the qualitative results differ across different seeds. While it's probably not equivalent to posterior sampling, this behaviour, however, can be leveraged to produce diverse samples (see Figure 15).

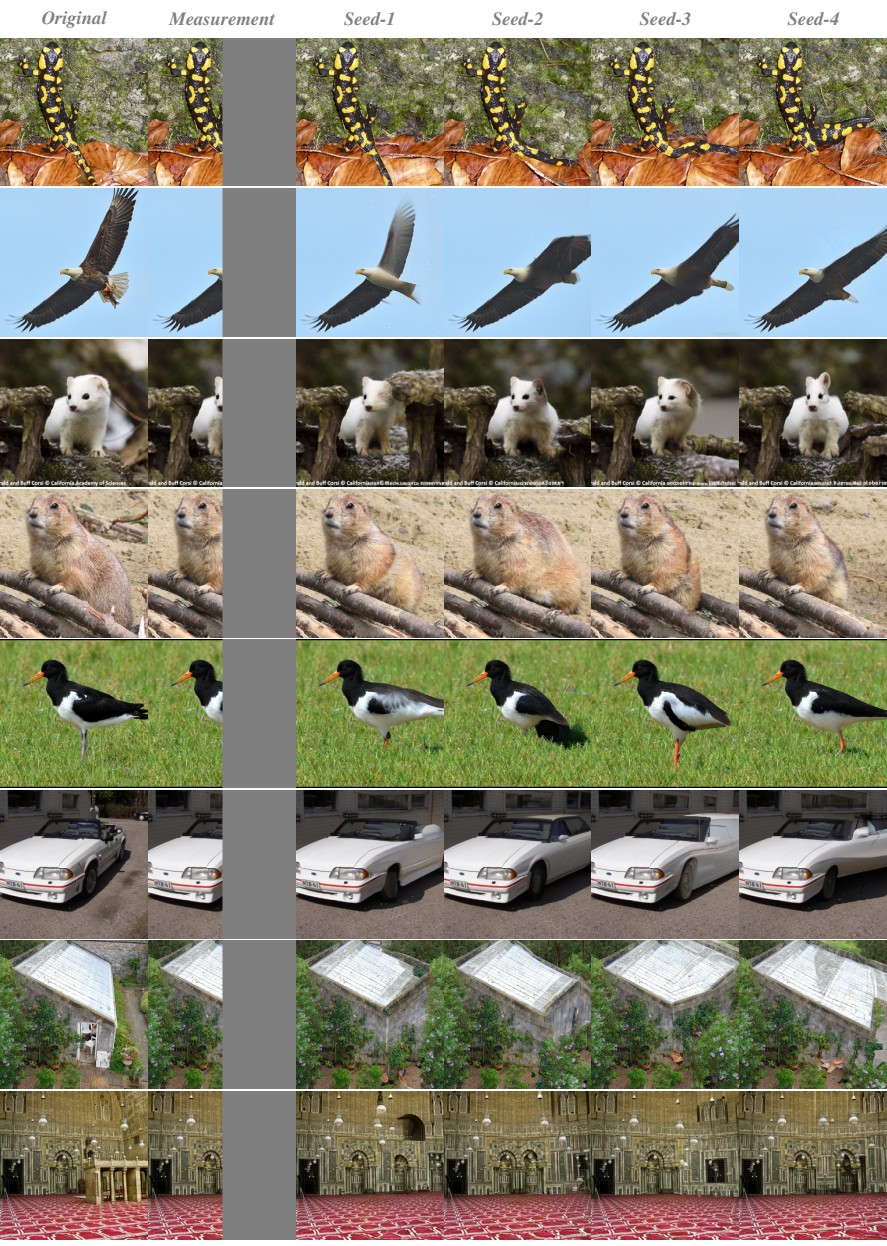

Figure 15: Restored images of VML-MAP across different seeds for the half-mask inpainting task. Left to right: original, measurement, and restored images of VML-MAP with 4 different seeds.

## D EXTENSION TO LATENT DIFFUSION MODELS (LDM)

### D.1 APPROXIMATING THE VML FOR LDMS

Here, we provide an extension of the VML objective to Latent Diffusion Models (LDM) for solving inverse problems. In LDMs, we treat both the encoder $\mathcal{E}$ and the decoder $\mathcal{D}$ as deterministic mappings from a clean image $\mathbf{x}_0$ to a clean latent variable $\mathbf{z}_0$ and vice-versa, respectively. To solve an inverse problem with a pre-trained LDM and a given measurement $\mathbf{y}$, we first aim to solve for the MAP estimate $\mathbf{z}_0^* = \arg\max_{\mathbf{z}_0} \log p(\mathbf{z}_0|\mathbf{y})$ with the VML objective extended to LDMs and later use the decoder to predict a clean image in a deterministic manner i.e., $\mathbf{x}_0^* = \mathcal{D}(\mathbf{z}_0^*)$. First, we define the VML objective in LDMs, which we term $\text{VML}_{\text{LDM}}$, as the KL divergence between $p(\mathbf{z}_0|\mathbf{z}_t)$ and $p(\mathbf{z}_0|\mathbf{y})$, and further approximate it as given below.

**Proposition 6.** *The variational mode-seeking-loss for an LDM* ($\text{VML}^{\text{LDM}}$) *at diffusion time $t$, for a degradation operator $\mathcal{A}$, measurement $\mathbf{y}$, and measurement noise variance $\sigma_{\mathbf{y}}^2$ is given by*

$$\text{VML}^{\text{LDM}} = D_{\text{KL}}(p(\mathbf{z}_0|\mathbf{z}_t)||p(\mathbf{z}_0|\mathbf{y})) \approx -\log p(\mathbf{z}_t) - \frac{\|\text{D}(\mathbf{z}_t,t) - \mathbf{z}_t\|^2}{2\sigma_t^2} - \frac{1}{2\sigma_t^2}\text{Tr}\left\{\text{Cov}[\mathbf{z}_0|\mathbf{z}_t]\right\}$$

$$+ \frac{\|\mathbf{y} - \mathcal{A}(\mathcal{D}(\text{D}(\mathbf{z}_t,t)))\|^2}{2\sigma_{\mathbf{y}}^2} + \frac{1}{2\sigma_{\mathbf{y}}^2}\left(\text{Tr}\left\{\frac{\partial \mathcal{A}(\mathcal{D}(\text{D}(\mathbf{z}_t,t)))}{\partial \text{D}(\mathbf{z}_t,t)}\text{Cov}[\mathbf{z}_0|\mathbf{z}_t]\frac{\partial \mathcal{A}(\mathcal{D}(\text{D}(\mathbf{z}_t,t)))^\top}{\partial \text{D}(\mathbf{z}_t,t)}\right\}\right) + \text{C}$$

*where C is a constant, independent of $\mathbf{x}_t$. Tr denotes the matrix trace, Cov denotes the covariance matrix, $\text{D}(\cdot,\cdot)$ denotes the denoiser, and $\mathcal{D}(\cdot)$ denotes the LDM decoder.*

*Proof.*

$$D_{\text{KL}}(p(\mathbf{z}_0|\mathbf{z}_t)||p(\mathbf{z}_0|\mathbf{y})) = \int_{\mathbf{z}_0} p(\mathbf{z}_0|\mathbf{z}_t)\log\frac{p(\mathbf{z}_0|\mathbf{z}_t)}{p(\mathbf{z}_0|\mathbf{y})}\text{d}\mathbf{z}_0$$

$$D_{\text{KL}}(p(\mathbf{z}_0|\mathbf{z}_t)||p(\mathbf{z}_0|\mathbf{y})) = \int_{\mathbf{z}_0} p(\mathbf{z}_0|\mathbf{z}_t)\log\frac{p(\mathbf{z}_t|\mathbf{z}_0)p(\mathbf{z}_0)p(\mathbf{y})}{p(\mathbf{z}_t)p(\mathbf{y}|\mathbf{z}_0)p(\mathbf{z}_0)}\text{d}\mathbf{z}_0$$

$$D_{\text{KL}}(p(\mathbf{z}_0|\mathbf{z}_t)||p(\mathbf{z}_0|\mathbf{y})) = \log p(\mathbf{y}) - \log p(\mathbf{z}_t) + \int_{\mathbf{z}_0} p(\mathbf{z}_0|\mathbf{z}_t)\log\frac{p(\mathbf{z}_t|\mathbf{z}_0)}{p(\mathbf{y}|\mathbf{z}_0)}\text{d}\mathbf{z}_0$$

$$D_{\text{KL}}(p(\mathbf{z}_0|\mathbf{z}_t)||p(\mathbf{z}_0|\mathbf{y})) = \log p(\mathbf{y}) - \log p(\mathbf{z}_t) + \left(\int_{\mathbf{z}_0} p(\mathbf{z}_0|\mathbf{z}_t)\log p(\mathbf{z}_t|\mathbf{z}_0)\text{d}\mathbf{z}_0\right)$$

$$- \left(\int_{\mathbf{z}_0} p(\mathbf{z}_0|\mathbf{z}_t)\log p(\mathbf{y}|\mathbf{z}_0)\text{d}\mathbf{z}_0\right)$$

$$\{\text{Note that } p(\mathbf{z}_t|\mathbf{z}_0) = \mathcal{N}(\mathbf{z}_0, \sigma_t^2\mathbf{I}) \text{ and we approximate } p(\mathbf{y}|\mathbf{z}_0) \approx \mathcal{N}(\mathcal{A}(\mathcal{D}(\mathbf{z}_0)), \sigma_{\mathbf{y}}^2\mathbf{I})\}$$

$$D_{\text{KL}}(p(\mathbf{z}_0|\mathbf{z}_t)||p(\mathbf{z}_0|\mathbf{y})) \approx -\log p(\mathbf{z}_t) - \frac{1}{2}\left(\int_{\mathbf{z}_0} p(\mathbf{z}_0|\mathbf{z}_t)\frac{\|\mathbf{z}_t - \mathbf{z}_0\|^2}{\sigma_t^2}\text{d}\mathbf{z}_0\right)$$

$$+ \frac{1}{2}\left(\int_{\mathbf{z}_0} p(\mathbf{z}_0|\mathbf{z}_t)\frac{\|\mathbf{y} - \mathcal{A}(\mathcal{D}(\mathbf{z}_0))\|^2}{\sigma_{\mathbf{y}}^2}\text{d}\mathbf{z}_0\right) + \text{C}$$

$$D_{\text{KL}}(p(\mathbf{z}_0|\mathbf{z}_t)||p(\mathbf{z}_0|\mathbf{y})) \approx -\log p(\mathbf{z}_t) - \frac{1}{2\sigma_t^2}\left(\|\mathbf{z}_t\|^2 - 2\mathbf{z}_t^\top \text{D}(\mathbf{z}_t,t) + \int_{\mathbf{z}_0} \|\mathbf{z}_0\|^2 p(\mathbf{z}_0|\mathbf{z}_t)\text{d}\mathbf{z}_0\right)$$

$$+ \frac{1}{2\sigma_{\mathbf{y}}^2}\left(-2\mathbf{y}^\top \int_{\mathbf{z}_0} \mathcal{A}(\mathcal{D}(\mathbf{z}_0))p(\mathbf{z}_0|\mathbf{z}_t)\text{d}\mathbf{z}_0 + \int_{\mathbf{z}_0} \|\mathcal{A}(\mathcal{D}(\mathbf{z}_0))\|^2 p(\mathbf{z}_0|\mathbf{z}_t)\text{d}\mathbf{z}_0\right) + \text{C}$$

We make a linear approximation of $\mathcal{A}(\mathcal{D}(\mathbf{z}_0))$ around $\hat{\mathbf{z}}_t = \text{D}(\mathbf{z}_t, t) = \int_{\mathbf{z}_0} \mathbf{z}_0 p(\mathbf{z}_0|\mathbf{z}_t)$ as follows

$$\mathcal{A}(\mathcal{D}(\mathbf{z}_0)) \approx \mathcal{A}(\mathcal{D}(\hat{\mathbf{z}}_t)) + \frac{\partial \mathcal{A}(\mathcal{D}(\hat{\mathbf{z}}_t))}{\partial \hat{\mathbf{z}}_t}(\mathbf{z}_0 - \hat{\mathbf{z}}_t)$$

$$D_{\text{KL}}(p(\mathbf{z}_0|\mathbf{z}_t)||p(\mathbf{z}_0|\mathbf{y})) \approx -\log p(\mathbf{z}_t) - \frac{1}{2\sigma_t^2}\left(\|\mathbf{z}_t\|^2 - 2\mathbf{z}_t^\top \text{D}(\mathbf{z}_t,t) + \text{Tr}\left\{\text{Cov}[\mathbf{z}_0|\mathbf{z}_t]\right\} + \|\text{D}(\mathbf{z}_t,t)\|^2\right)$$

$$+ \frac{1}{2\sigma_{\mathbf{y}}^2}\left(-2\mathbf{y}^\top\left\{\mathcal{A}(\mathcal{D}(\hat{\mathbf{z}}_t)) + \frac{\partial \mathcal{A}(\mathcal{D}(\hat{\mathbf{z}}_t))}{\partial \hat{\mathbf{z}}_t}\int_{\mathbf{z}_0}(\mathbf{z}_0 - \hat{\mathbf{z}}_t)p(\mathbf{z}_0|\mathbf{z}_t)\text{d}\mathbf{z}_0\right\} + \int_{\mathbf{z}_0} \|\mathcal{A}(\mathcal{D}(\mathbf{z}_0))\|^2 p(\mathbf{z}_0|\mathbf{z}_t)\text{d}\mathbf{z}_0\right) + \text{C}$$

$$D_{\text{KL}}(p(\mathbf{z}_0|\mathbf{z}_t)||p(\mathbf{z}_0|\mathbf{y})) \approx -\log p(\mathbf{z}_t) - \frac{\|\text{D}(\mathbf{z}_t,t) - \mathbf{z}_t\|^2}{2\sigma_t^2} - \frac{1}{2\sigma_t^2}\text{Tr}\left\{\text{Cov}[\mathbf{z}_0|\mathbf{z}_t]\right\}$$

$$+ \frac{1}{2\sigma_{\mathbf{y}}^2}\left(-2\mathbf{y}^\top \mathcal{A}(\mathcal{D}(\hat{\mathbf{z}}_t)) + \|\mathcal{A}(\mathcal{D}(\hat{\mathbf{z}}_t))\|^2 + 2\mathcal{A}(\mathcal{D}(\hat{\mathbf{z}}_t))^\top \frac{\partial \mathcal{A}(\mathcal{D}(\hat{\mathbf{z}}_t))}{\partial \hat{\mathbf{z}}_t}\int_{\mathbf{z}_0}(\mathbf{z}_0 - \hat{\mathbf{z}}_t)p(\mathbf{z}_0|\mathbf{z}_t)\mathrm{d}\mathbf{z}_0\right)$$

$$+ \frac{1}{2\sigma_{\mathbf{y}}^2}\left(\int_{\mathbf{z}_0}\left\|\frac{\partial \mathcal{A}(\mathcal{D}(\hat{\mathbf{z}}_t))}{\partial \hat{\mathbf{z}}_t}(\mathbf{z}_0 - \hat{\mathbf{z}}_t)\right\|^2 p(\mathbf{z}_0|\mathbf{z}_t)\mathrm{d}\mathbf{z}_0\right) + \mathrm{C}$$

$$D_{\mathrm{KL}}(p(\mathbf{z}_0|\mathbf{z}_t)||p(\mathbf{z}_0|\mathbf{y})) \approx -\log p(\mathbf{z}_t) - \frac{\|\mathrm{D}(\mathbf{z}_t,t) - \mathbf{z}_t\|^2}{2\sigma_t^2} - \frac{1}{2\sigma_t^2}\mathrm{Tr}\{\mathrm{Cov}[\mathbf{z}_0|\mathbf{z}_t]\} + \frac{\|\mathbf{y} - \mathcal{A}(\mathcal{D}(\hat{\mathbf{z}}_t))\|^2}{2\sigma_{\mathbf{y}}^2}$$

$$+ \frac{1}{2\sigma_{\mathbf{y}}^2}\left(\mathrm{Tr}\left\{\frac{\partial \mathcal{A}(\mathcal{D}(\hat{\mathbf{z}}_t))}{\partial \hat{\mathbf{z}}_t}\left(\int_{\mathbf{z}_0}(\mathbf{z}_0 - \hat{\mathbf{z}}_t)(\mathbf{z}_0 - \hat{\mathbf{z}}_t)^\top p(\mathbf{z}_0|\mathbf{z}_t)\mathrm{d}\mathbf{z}_0\right)\frac{\partial \mathcal{A}(\mathcal{D}(\hat{\mathbf{z}}_t))^\top}{\partial \hat{\mathbf{z}}_t}\right\}\right) + \mathrm{C}$$

$$D_{\mathrm{KL}}(p(\mathbf{z}_0|\mathbf{z}_t)||p(\mathbf{z}_0|\mathbf{y})) \approx -\log p(\mathbf{z}_t) - \frac{\|\mathrm{D}(\mathbf{z}_t,t) - \mathbf{z}_t\|^2}{2\sigma_t^2} - \frac{1}{2\sigma_t^2}\mathrm{Tr}\{\mathrm{Cov}[\mathbf{z}_0|\mathbf{z}_t]\} + \frac{\|\mathbf{y} - \mathcal{A}(\mathcal{D}(\hat{\mathbf{z}}_t))\|^2}{2\sigma_{\mathbf{y}}^2}$$

$$+ \frac{1}{2\sigma_{\mathbf{y}}^2}\left(\mathrm{Tr}\left\{\frac{\partial \mathcal{A}(\mathcal{D}(\hat{\mathbf{z}}_t))}{\partial \hat{\mathbf{z}}_t}\mathrm{Cov}[\mathbf{z}_0|\mathbf{z}_t]\frac{\partial \mathcal{A}(\mathcal{D}(\hat{\mathbf{z}}_t))^\top}{\partial \hat{\mathbf{z}}_t}\right\}\right) + \mathrm{C}$$

$$D_{\mathrm{KL}}(p(\mathbf{z}_0|\mathbf{z}_t)||p(\mathbf{z}_0|\mathbf{y})) \approx -\log p(\mathbf{z}_t) - \frac{\|\mathrm{D}(\mathbf{z}_t,t) - \mathbf{z}_t\|^2}{2\sigma_t^2} - \frac{1}{2\sigma_t^2}\mathrm{Tr}\{\mathrm{Cov}[\mathbf{z}_0|\mathbf{z}_t]\} + \frac{\|\mathbf{y} - \mathcal{A}(\mathcal{D}(\mathrm{D}(\mathbf{z}_t,t)))\|^2}{2\sigma_{\mathbf{y}}^2}$$

$$+ \frac{1}{2\sigma_{\mathbf{y}}^2}\left(\mathrm{Tr}\left\{\frac{\partial \mathcal{A}(\mathcal{D}(\mathrm{D}(\mathbf{z}_t,t)))}{\partial \mathrm{D}(\mathbf{z}_t,t)}\mathrm{Cov}[\mathbf{z}_0|\mathbf{z}_t]\frac{\partial \mathcal{A}(\mathcal{D}(\mathrm{D}(\mathbf{z}_t,t)))^\top}{\partial \mathrm{D}(\mathbf{z}_t,t)}\right\}\right) + \mathrm{C}$$

$\square$

## D.2 Simplified VML and Latent VML-MAP for LDMs

Similar to the case of pixel diffusion models, the higher-order terms involving $\mathrm{Cov}[\mathbf{z}_0|\mathbf{z}_t]$ in $\mathrm{VML}^{\mathrm{LDM}}$ converge to a constant as $t \to 0$, under mild assumptions on $\mathcal{A}$, and $\mathcal{D}$. Ignoring the higher-order terms, we define the simplified $\mathrm{VML}^{\mathrm{LDM}}$ objective and its gradient as follows.

**Simplified** $\mathrm{VML}^{\mathrm{LDM}}$ **and its gradient.** For a linear degradation matrix H, the simplified $\mathrm{VML}^{\mathrm{LDM}}$ (i.e, $\mathrm{VML}_{\mathrm{S}}^{\mathrm{LDM}}$) and its gradient is given by

$$\mathrm{VML}_{\mathrm{S}}^{\mathrm{LDM}} = -\log p(\mathbf{z}_t) - \frac{\|\mathrm{D}(\mathbf{z}_t,t) - \mathbf{z}_t\|^2}{2\sigma_t^2} + \frac{\|\mathbf{y} - \mathrm{H}\mathcal{D}(\mathrm{D}(\mathbf{z}_t,t))\|^2}{2\sigma_{\mathbf{y}}^2}$$

$$\nabla_{\mathbf{z}_t}\mathrm{VML}_{\mathrm{S}}^{\mathrm{LDM}} = \underbrace{-\frac{\partial \mathrm{D}^\top(\mathbf{z}_t,t)}{\partial \mathbf{z}_t}\frac{\partial \mathcal{D}^\top(\mathrm{D}(\mathbf{z}_t,t))}{\partial \mathrm{D}(\mathbf{z}_t,t)}\frac{\mathrm{H}^\top(\mathbf{y} - \mathrm{H}\mathcal{D}(\mathrm{D}(\mathbf{z}_t,t)))}{\sigma_{\mathbf{y}}^2}}_{\text{measurement consistency gradient}} \underbrace{-\frac{\partial \mathrm{D}^\top(\mathbf{z}_t,t)}{\partial \mathbf{z}_t}\frac{(\mathrm{D}(\mathbf{z}_t,t) - \mathbf{z}_t)}{\sigma_t^2}}_{\text{prior gradient}}$$

*Proof.*

$$\nabla_{\mathbf{z}_t}\mathrm{VML}_{\mathrm{S}}^{\mathrm{LDM}} = \left\{-\nabla_{\mathbf{z}_t}\log p(\mathbf{z}_t)\right\} - \left\{\nabla_{\mathbf{z}_t}\frac{\|\mathrm{D}(\mathbf{z}_t,t) - \mathbf{z}_t\|^2}{2\sigma_t^2}\right\} + \left\{\nabla_{\mathbf{z}_t}\frac{\|\mathbf{y} - \mathrm{H}\mathcal{D}(\mathrm{D}(\mathbf{z}_t,t))\|^2}{2\sigma_{\mathbf{y}}^2}\right\}$$

$$\nabla_{\mathbf{z}_t}\mathrm{VML}_{\mathrm{S}}^{\mathrm{LDM}} = \left\{-\frac{\mathrm{D}(\mathbf{z}_t,t) - \mathbf{z}_t}{\sigma_t^2}\right\} - \left\{\left(\frac{\partial \mathrm{D}^\top(\mathbf{z}_t,t)}{\partial \mathbf{z}_t}\frac{(\mathrm{D}(\mathbf{z}_t,t) - \mathbf{z}_t)}{\sigma_t^2}\right) - \frac{\mathrm{D}(\mathbf{z}_t,t) - \mathbf{z}_t}{\sigma_t^2}\right\}$$

$$+ \left\{-\frac{\partial \mathrm{D}^\top(\mathbf{z}_t,t)}{\partial \mathbf{z}_t}\frac{\partial \mathcal{D}^\top(\mathrm{D}(\mathbf{z}_t,t))}{\partial \mathrm{D}(\mathbf{z}_t,t)}\frac{\mathrm{H}^\top(\mathbf{y} - \mathrm{H}\mathcal{D}(\mathrm{D}(\mathbf{x}_t,t)))}{\sigma_{\mathbf{y}}^2}\right\}$$

$$\nabla_{\mathbf{z}_t}\mathrm{VML}_{\mathrm{S}}^{\mathrm{LDM}} = -\frac{\partial \mathrm{D}^\top(\mathbf{z}_t,t)}{\partial \mathbf{z}_t}\frac{\partial \mathcal{D}^\top(\mathrm{D}(\mathbf{z}_t,t))}{\partial \mathrm{D}(\mathbf{z}_t,t)}\frac{\mathrm{H}^\top(\mathbf{y} - \mathrm{H}\mathcal{D}(\mathrm{D}(\mathbf{z}_t,t)))}{\sigma_{\mathbf{y}}^2} - \frac{\partial \mathrm{D}^\top(\mathbf{z}_t,t)}{\partial \mathbf{z}_t}\frac{(\mathrm{D}(\mathbf{z}_t,t) - \mathbf{z}_t)}{\sigma_t^2}$$

$\square$

We present LatentVML-MAP (Algorithm 2) as an extension of VML-MAP (Algorithm 1) to LDMs. In principle, LatentVML-MAP minimizes $\mathrm{VML}_{\mathrm{S}}^{\mathrm{LDM}}$ at each reverse diffusion step to find $\mathbf{z}_0^* = \arg\max_{\mathbf{z}_0}\log p(\mathbf{z}_0|\mathbf{y})$ and finally, uses the decoder $\mathcal{D}$ to return $\mathbf{x}_0 = \mathcal{D}(\mathbf{z}_0^*)$.

The inputs to Algorithm 2 consists of the latent diffusion denoiser $\mathrm{D}_\theta(\cdot,\cdot)$, the decoder $\mathcal{D}(\cdot)$, the linear degradation matrix H, the measurement $\mathbf{y}$ with noise variance $\sigma_{\mathbf{y}}^2$, the diffusion noise schedule $\sigma(\cdot)$, the total number of reverse diffusion steps $N$ with the discretized time step schedule specified by $t_{i\in\{0,\dots N\}}$, where $t_0 = 0$, the gradient descent iterations per step given by $K$, and the learning rate $\gamma$. We use the notations $\sigma(t)$ and $\sigma_t$ interchangeably.

**Algorithm 2:** LatentVML-MAP

**Input:** $D_\theta(\cdot, \cdot), \mathcal{D}(\cdot), H, \mathbf{y}, \sigma_\mathbf{y}, \sigma(\cdot), t_{i \in \{0, \dots N\}}, K, \gamma$
**Output:** $\mathbf{x}_{t_0}$
**Initialize** $\mathbf{z}_{t_N} \sim \mathcal{N}(\mathbf{0}, \sigma_{t_N}^2 \mathbf{I})$
**for** $i \leftarrow N$ **to** $1$ **do**
    **for** $j \leftarrow 1$ **to** $K$ **do**
        $\mathbf{z}_{t_i} \leftarrow \mathbf{z}_{t_i} - \gamma \cdot \nabla_{\mathbf{z}_{t_i}} \text{VML}_\text{S}^\text{LDM}$             /* see Appendix D.2 */
    **end**
    $\mathbf{z}_{t_{i-1}} \sim \mathcal{N}(D_\theta(\mathbf{z}_{t_i}, t_i), \sigma_{t_{i-1}}^2 \mathbf{I})$
**end**
$\mathbf{x}_{t_0} = \mathcal{D}(\mathbf{z}_{t_0})$
**Return** $\mathbf{x}_{t_0}$

### D.3 EXPERIMENTS ON CELEBA

We conduct experiments on 100 test images from the CelebA (Liu et al., 2015) dataset on Half-mask inpainting, Box-mask inpainting, $4\times$ Super-resolution, and Deblurring tasks using the pre-trained latent diffusion model and the autoencoder from Rombach et al. (2022). For LatentVML-MAP, we fix $N = 20$, $K = 50$, and use the EDM scheduler with $\sigma_{min} = 0.002$, $\sigma_{max} = 80$, and $\rho = 7$. For Resample (Song et al., 2024), we follow the default settings with 500 time steps, and for LatentDAPS (Zhang et al., 2025), instead of the default setting of 100 neural function evaluations, we select 1000 NFEs to allow for a fair comparison, keeping other parameters fixed. We set $\sigma_y = 1e\text{-}9$ for all the methods. For LatentVML-MAP, we report the best learning rate configuration for each task as $\gamma_0 \cdot \sigma_y^2$, with $\gamma_0$ as follows: Half-mask inpainting ($\gamma_0 = 0.1$), Box-mask inpainting ($\gamma_0 = 0.1$), $4\times$ Super-resolution ($\gamma_0 = 0.9$), Deblurring ($\gamma_0 = 0.075$).

Table 7: Evaluation of LDM-based image restoration methods on Half-mask inpainting, Box-mask inpainting, $4\times$ super-resolution, Deblurring on 100 images of CelebA256. Excluding Resample, we denote the best values in **bold**, second best values underlined.

| Dataset | Method | Half-Inpaint | | Box-Inpaint | | $4\times$Sup-res | | Deblurring | |
|---------|--------|--------------|------|-------------|------|-----------|------|-----------|------|
| | | LPIPS↓ | FID↓ | LPIPS↓ | FID↓ | LPIPS↓ | FID↓ | LPIPS↓ | FID↓ |
| CelebA | Resample | 0.235 | 55.80 | 0.092 | 53.12 | 0.084 | 41.85 | 0.202 | 63.41 |
| | LatentDAPS-1K | 0.240 | 54.47 | **0.070** | 37.69 | 0.114 | 38.72 | 0.328 | 104.2 |
| | Resample w/o PO | 0.259 | 64.32 | 0.106 | 68.30 | 0.112 | 50.95 | **0.210** | **65.36** |
| | **LatentVML-MAP** | **0.208** | **47.04** | 0.074 | **34.73** | **0.099** | **36.82** | 0.212 | 71.43 |

For LatentVML-MAP, we observed that even for linear inverse problems, the optimization becomes more challenging due to the non-linearity of the LDM decoder. As a result, we noticed that the final reconstructed images are blurry and inconsistent with the measurement $\mathbf{y}$. We also observed this pattern with LatentDAPS, but not with Resample, as it uses pixel-space optimization. Note that Resample also requires the LDM encoder to project the pixel-space-optimized result back into the latent space, unlike LatentDAPS and LatentVML-MAP. For a fair comparison, in our experiments,

- we also report the performance of Resample with its pixel-space optimization replaced with latent-space optimization (see Song et al. (2024)), denoted as **Resample w/o PO**

- we project the final reconstructed images of all methods onto the measurement subspace to ensure that all the reconstructed images are consistent with the measurements.

For inpainting, the measurement subspace projection implies that we paste the observed pixels back into the reconstructed images. For super-resolution and deblurring, we paste back the observations in the spectral space of the linear operator by using the SVD accessible operators from Kawar et al. (2022). Results from Table 7 validate the effectiveness of LatentVML-MAP in practice. We believe that the performance bottleneck primarily exists due to the challenging optimization and that improved optimization techniques can further enhance the performance of LatentVML-MAP.

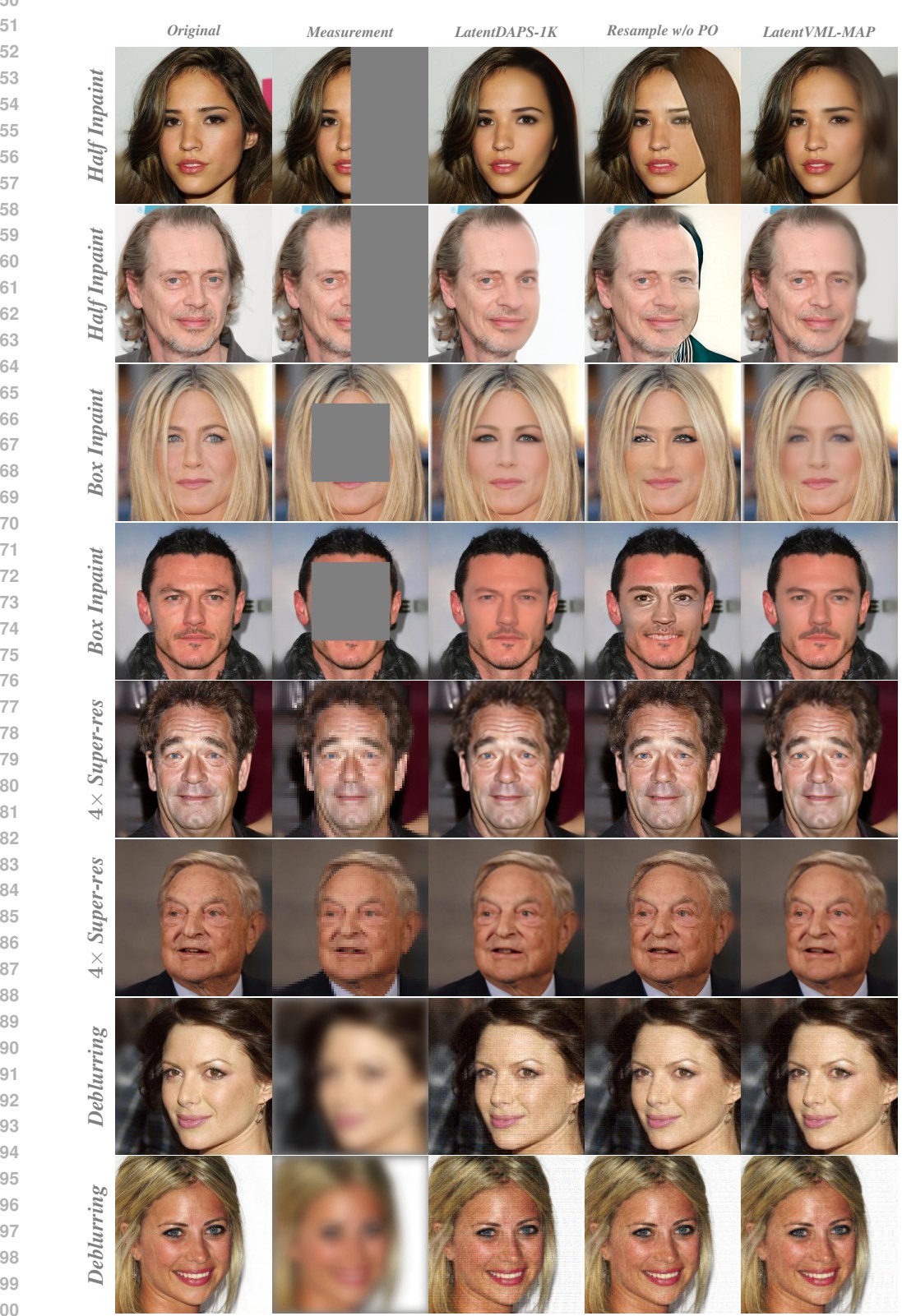

Figure 16: Left to right: original image, measurement, restored images with LatentDAPS-1K, Resample w/o PO, and LatentVML-MAP. Zoom in for the best view.

