# OpenReview forum: "Mode-seeking for inverse problems with diffusion models"
_ICLR.cc/2026/Conference — Submitted to ICLR 2026_

### Official Review · Reviewer_2tpR · 2025-10-26

**Soundness:** 2
**Presentation:** 3
**Contribution:** 2
**Rating:** 2
**Confidence:** 4

**Summary:**

The paper introduces a variational mode-seeking loss (VML) for solving inverse problems with diffusion models and derives a closed-form expression for linear operators. While the formulation is conceptually interesting and mathematically sound, the experimental validation is limited and not convincing. The paper only evaluates nearly noise-free settings and reports perceptual metrics (FID, LPIPS) without standard reconstruction measures such as PSNR or SSIM. As a result, it is difficult to assess the practical effectiveness and robustness of the proposed method. If the authors could provide stronger experimental evidence, particularly under noisy conditions and with standard reconstruction metrics, the contribution would become significantly more convincing and could merit a higher evaluation.

**Strengths:**

1. The derivation for linear inverse problems is clear and mathematically consistent, showing the authors’ understanding of the theoretical aspects.

2. The proposed algorithms are conceptually straightforward and easy to implement, requiring only a pre-trained unconditional diffusion model.

3. The writing and organization of the paper are generally clear, with helpful figures and tables that make it easier to follow the main idea.

**Weaknesses:**

1. **Limited experimental settings**:
The experiments are conducted only under an almost noise-free condition (σₓ = 1e-9). However, in diffusion-based inverse problem literature, it is standard to evaluate both noisy (e.g., σₓ = 0.05) and noise-free scenarios to assess robustness. The lack of results in noisy conditions makes it difficult to judge the algorithm’s stability and general applicability.

2. **Missing key evaluation metrics**:
While the paper reports FID and LPIPS, it omits SSIM and PSNR, which are standard quantitative metrics for measuring reconstruction fidelity in inverse problems. The absence of these metrics substantially weakens the validation of the proposed method’s effectiveness.

3. **Formatting issue**:
The submission contains blue-colored text, suggesting it was uploaded as an unclean diff version. Although minor, this formatting issue slightly affects readability and presentation quality.

**Questions:**

1. The paper only evaluates nearly noise-free settings (σᵧ = 1e−9). Could the authors provide results for noisy scenarios (e.g., σᵧ = 0.01 or 0.05) to test robustness?

2. Why are PSNR and SSIM not reported? These are standard reconstruction metrics for inverse problems.

3. The submission includes blue-colored text, suggesting an unclean version. Please ensure a clean and properly formatted final submission.

---

> ### Author Response · Authors · 2025-11-20
> **Official Response to Reviewer 2tpR (1/2)**
>
> We thank the reviewer for their valuable feedback. Below, we addressed the posed questions and concerns.
>
> $\textbf{Formatting Issue:}$
>
> We apologize for the misunderstanding regarding the blue-colored text. We’d like to clarify that the submission is indeed a clean version and that the blue-colored text was added intentionally to emphasize certain parts of the text to the reader, including key ideas and certain experiments which were deferred to the Appendix. Following the suggested comments, we have now uploaded a properly formatted version.
>
>
> $\textbf{Evaluation metrics:}$
>
> We thank the reviewer for bringing up this point. While we acknowledge the importance of PSNR and SSIM as conventional reconstruction metrics, we note that, in the context of image restoration inverse problems, these metrics do not always reflect perceptual quality. The perception-distortion tradeoff from Blau et al. CVPR 2018 (see [1]) demonstrates, both formally and empirically, the tradeoff between distortion metrics (e.g., PSNR, SSIM) and perceptual metrics, and has been well established in the literature. From  [1], a less perceptual image can often achieve a high PSNR, while the goal of image restoration is to reconstruct a plausible (perceptually better) image. [2] further solidifies the formal mechanism of the trade-off for specific distortion and perceptual metrics. In this regard, a high PSNR or SSIM does not necessarily imply a better perceptual sample, while a low LPIPS and FID is more strongly correlated with the sample's perceptual quality. In fact, the severe ill-posedness of the more interesting (and thus challenging) inpainting tasks, can exacerbate this trade-off.
>
> For more context, we quote the following text from the Introduction section of [1]:
>
> *"In fact, and perhaps counter-intuitively, algorithms that are superior in terms of perceptual quality, are often inferior in terms of e.g. PSNR and SSIM. This phenomenon is commonly interpreted as a shortcoming of the existing distortion measures, which fuels a constant search for alternative “more perceptual” criteria."*
>
> For completeness, we report PSNR and SSIM metrics below for the half-mask inpainting and $4\times$ Super-resolution experiments on ImageNet256 from the paper. The methods achieving high PSNR indeed have poor perceptual sample quality (i.e., higher LPIPS, FID), and can also be observed from qualitative visualizations (see Figures 8 and 9 in the paper), highlighting the perception-distortion tradeoff and also the need for more emphasis on perceptual metrics (see [3]). Particularly, note the worse performance of DAPS 1K,4K (CVPR 25) compared to earlier methods, e.g., DDRM (NeurIPS 2022) in terms of PSNR and SSIM. For improved context, we report PSNR and SSIM metrics for more experiments in our final submission and also add a discussion on this tradeoff in the Appendix.
>
> **Table 1.** Half-mask inpainting on 100 validation images of ImageNet256.
> Mean ± std across 4 seeds is reported. Best values are **bold**, second-best are $\underline{underlined}$.
>
> | Dataset   | Method     |       PSNR ↑       |       SSIM ↑       |       LPIPS ↓       |       FID ↓       |
> |------------|-------------|------------------|------------------|------------------|-----------------|
> |            | DDRM        | $\underline{15.01}$ ± $\color{lightgray}{0.055}$  | $\underline{0.679}$ ± $\color{lightgray}{0.000}$  | 0.393 ± $\color{lightgray}{0.001}$    | 103.5 ± $\color{lightgray}{0.657}$   |
> |            | ΠGDM        | **15.08** ± $\color{lightgray}{0.081}$| **0.681** ± $\color{lightgray}{0.001}$| 0.373 ± $\color{lightgray}{0.000}$    | 103.7 ± $\color{lightgray}{0.694}$   |
> |            | MAPGA       | 14.51 ± $\color{lightgray}{0.044}$    | 0.654 ± $\color{lightgray}{0.000}$    | $\underline{0.290}$ ± $\color{lightgray}{0.001}$  | $\underline{80.76}$ ± $\color{lightgray}{2.013}$ |
> | ImageNet   | DAPS-1K     | 14.88 ± $\color{lightgray}{0.117}$    | 0.641 ± $\color{lightgray}{0.000}$    | 0.384 ± $\color{lightgray}{0.001}$    | 98.45 ± $\color{lightgray}{1.879}$   |
> |            | DAPS-4K     | 14.59 ± $\color{lightgray}{0.115}$    | 0.637 ± $\color{lightgray}{0.001}$    | 0.365 ± $\color{lightgray}{0.004}$    | 93.62 ± $\color{lightgray}{0.856}$   |
> |            | VML-MAP | 14.34 ± $\color{lightgray}{0.067}$    | 0.631 ± $\color{lightgray}{0.001}$    | **0.262** ± $\color{lightgray}{0.002}$| **74.21** ± $\color{lightgray}{3.209}$|

---

> ### Author Response · Authors · 2025-11-20
> **Official Response to Reviewer 2tpR (2/2)**
>
> **Table 2.** 4$\times$ super-resolution on 100 validation images of ImageNet256.
> Mean ± std across 4 seeds is reported. Best values are **bold**, second-best are $\underline{underlined}$.
>
> | Dataset   | Method     |       PSNR ↑       |       SSIM ↑       |       LPIPS ↓       |       FID ↓       |
> |------------|-------------|------------------|------------------|------------------|-----------------|
> |            | DDRM        | 24.20 ± $\color{lightgray}{0.065}$  | $0.738$ ± $\color{lightgray}{0.000}$  | 0.289 ± $\color{lightgray}{0.000}$    | 89.40 ± $\color{lightgray}{0.962}$   |
> |            | ΠGDM        | $\underline{24.41}$ ± $\color{lightgray}{0.068}$| $\underline{0.745}$ ± $\color{lightgray}{0.000}$| 0.292 ± $\color{lightgray}{0.001}$    |  83.88 ± $\color{lightgray}{0.226}$   |
> |            | MAPGA       | **24.63** ± $\color{lightgray}{0.038}$    | **0.755** ± $\color{lightgray}{0.000}$    | 0.273 ± $\color{lightgray}{0.001}$  | $76.12$ ± $\color{lightgray}{0.587}$ |
> | ImageNet   | DAPS-1K     | 23.82 ± $\color{lightgray}{0.038}$    | 0.669 ± $\color{lightgray}{0.000}$    | 0.254 ± $\color{lightgray}{0.001}$    | 71.60 ± $\color{lightgray}{1.010}$   |
> |            | DAPS-4K     | 23.62 ± $\color{lightgray}{0.049}$    | 0.670 ± $\color{lightgray}{0.000}$    | $\underline{0.244}$ ± $\color{lightgray}{0.000}$    |  $\underline{69.71}$ ± $\color{lightgray}{1.554}$   |
> |            | VML-MAP | 23.41 ± $\color{lightgray}{0.049}$    | 0.729 ± $\color{lightgray}{0.000}$    | **0.194** ± $\color{lightgray}{0.001} $ | **60.08** ± $\color{lightgray}{0.687}$|
>
>
> $\textbf{Robustness to noise:}$
>
>
> Indeed, this is a fair point. Note that the VML objective converges to $-\log p(\mathbf{x}\_0|\mathbf{y}) = -\log p(\mathbf{y}|\mathbf{x}\_0) - \log p(\mathbf{x}\_0) + \text{Constant} $. In its original formulation, we find that the VML objective is sensitive to measurement noise. However, we observe that the reweighted MAP objective (downweighting the prior with $\lambda \ll 1$, i.e., $-\log p(\mathbf{y}|\mathbf{x}\_0) - \lambda \log p(\mathbf{x}\_0) $) adds to the numerical stability of the algorithm and robustness to measurement noise. Note that the reweighted objective translates to reweighting the measurement consistency and the prior terms of VML (proof trivially follows from Proposition 4). While $\lambda$ can also be learned, we treat it as a hyperparameter and fix $\lambda = \sigma^2\_{min}$ in our experiments below.
>
> **Table 3: Robustness to measurement noise. Half-mask Inpainting (ImageNet256)**
>
> | σ_y |     $\ \ \ \ \ \ \ \ \ \ \ \ \ \ $ΠGDM |  |     $\ \ \ \ \ \ \ \ \ \ \ \ \ \ $DAPS |  |      $\ \ \ \ \ \ \ \ \ \ \ \ \ \ $VML-MAP |  |
> |:---:|:--------:|:--------:|:-------:|:--------:|:-----------:|:--------:|
> |     | LPIPS ↓  | FID ↓    | LPIPS ↓ | FID ↓    | LPIPS ↓     | FID ↓    |
> | 0.00 | 0.373 | 103.8 | 0.385 | 98.72 | 0.265 | 69.21 |
> | 0.01 | 0.373 | 107.6 | 0.385 | 99.33 | 0.262 | 71.68 |
> | 0.02 | 0.376 | 113.4 | 0.388 | 101.2 | 0.278 | 81.59 |
> | 0.04 | 0.383 | 124.9 | 0.404 | 109.0 | 0.315 | 92.71 |
> | 0.05 | 0.390 | 126.4 | 0.415 | 113.8 | 0.347 | 101.3 |
>
>
> **Table 4: Robustness to measurement noise. 4$\times$ Super-resolution (ImageNet256)**
>
> | σ_y | $\ \ \ \ \ \ \ \ \ \ \ \ \ \ $ΠGDM |  | $\ \ \ \ \ \ \ \ \ \ \ \ \ \ $DAPS |  | $\ \ \ \ \ \ \ \ \ \ \ \ \ \ $VML-MAP |  |
> |:---:|:--------:|:--------:|:-------:|:--------:|:-----------:|:--------:|
> |     | LPIPS ↓  | FID ↓    | LPIPS ↓ | FID ↓    | LPIPS ↓     | FID ↓    |
> | 0.00 | 0.292 | 83.80 | 0.256 | 73.33 | 0.192 | 59.91 |
> | 0.01 | 0.306 | 100.2 | 0.256 | 72.88 | 0.204 | 61.60 |
> | 0.02 | 0.324 | 114.1 | 0.259 | 73.04 | 0.241 | 74.76 |
> | 0.04 | 0.356 | 139.0 | 0.290 | 81.24 | 0.331 | 103.1 |
> | 0.05 | 0.371 | 147.3 | 0.345 | 97.17 | 0.373 | 113.2 |
>
> Experiments show that VML-MAP is robust up to moderate levels of measurement noise ($\sigma_{\mathbf{y}} \leq 0.02$), after which the performance begins to degrade. Note that other methods exhibit a similar trend, though the effect is less pronounced. For VML, this behavior stems from the sensitivity of its gradient term to $\sigma_{\mathbf{y}}$. While heuristic adjustments (eg, DAPS sets $\sigma_{min}=0.1$ to make their method more robust to noise. See under hyperparameters overview in Appendix F.2 of DAPS paper) could mitigate this issue, a more principled and effective solution would involve higher-order or advanced optimization techniques. This is beyond the current scope and is left for future work.
>
> [1] Blau, Yochai, and Tomer Michaeli. "The perception-distortion tradeoff." CVPR 2018.
>
> [2] Freirich et al. ”A Theory of the Distortion-Perception Tradeoff in Wasserstein Space”, NeurIPS 2021
>
> [3] Achituve, Idan, et al. "Inverse problem sampling in latent space using sequential Monte Carlo." ICML 2025.

---

> > ### Comment · Reviewer_2tpR · 2025-11-22
> >
> > Thank you for the authors’ response. The revisions made to the blue-colored text have indeed improved the readability of the paper, and I have raised my score from 2 to 4. However, for the following reasons, I am unable to give a higher rating:
> >
> > 1. **Limited performance in noisy settings.**
> >    In fact, in the literature on inverse problems solved with diffusion models, it is standard practice to report results under a noise level of $\sigma_y$ = 0.05. The noiseless setting can serve as a complementary experiment to provide completeness, but it should not be the primary evaluation scenario.
> >
> > 2. **Weak performance on non-perceptual metrics.**

---

> > > ### Author Response · Authors · 2025-11-26
> > > **Author Response to Reviewer 2tpR's comments**
> > >
> > > We thank the reviewer for the updated rating. However, several concerns motivating the lower rating appear to stem from misunderstandings or factually incorrect assumptions. We address these points below.
> > >
> > > $\textbf{1. Difficulty axes in inverse problems}$
> > >
> > > A central issue in the review is the implicit assumption that inverse-problem difficulty is determined primarily by the noise level. This overlooks a fundamental point: inverse problems vary in difficulty along two independent axes (i) the observation noise level and (ii) the severity or ill-posedness of the degradation operator, even in the zero-noise regime. Our experiments intentionally focus on harder degradation operators (see Appendix C.1), including half-mask inpainting, expand-mask, and a deblurring setup with more singular values zeroed out. Only a handful of prior works evaluate methods under such challenging conditions, and our results show that competing approaches degrade substantially in these regimes. Evaluating solely through the lens of the noise level leads to an incomplete assessment of our method.
> > >
> > >
> > > The reviewer’s claim that $\sigma_{\mathbf{y}} = 0.05$ constitutes a “standard” setting is not factual. Recent ICLR and ICML papers [1,2] report the majority of experiments with $\sigma_{\mathbf{y}} < 0.05$, and there is no consensus that $0.05$ should be the canonical choice. If anything, privileging this value raises the symmetric question of why an even higher noise level would not then be considered “standard” (a case where most methods fail to achieve good performance).
> > >
> > >
> > > Regarding the expectation of robustness at high noise levels: this comparison does not account for the substantial hyperparameter tuning required by some baselines. For example, DAPS achieves its reported robustness only after increasing the EDM default $\sigma_{\min}=0.002$ to $0.1$; using the default values results in significant degradation. Below, we show the sensitivity of DAPS to this parameter for the half-mask inpainting task:
> > >
> > > | σ_y |     $\ \ \  \ \ \ \ \ $ DAPS ($\sigma_{min}=0.1$) |  |    $\ \ \  \ \ \ \ \ $ DAPS ($\sigma_{min}=0.002$) |  |
> > > |:---:|:--------:|:--------:|:-------:|:--------:|
> > > |     | LPIPS ↓  | FID ↓    | LPIPS ↓ | FID ↓ |
> > > | 0.01 | 0.385 | 99.33 | 0.475  | 98.19 |
> > > | 0.02 | 0.388 | 101.2 | 0.475 | 99.95 |
> > > | 0.04 | 0.404 | 109.0 | 0.487 | 107.7 |
> > > | 0.05 | 0.415 | 113.8 | 0.496  | 112.1 |
> > >
> > >
> > > Other methods (DDRM, MAP-GA, etc.) similarly rely on nontrivial noise-handling mechanisms, although these are rarely emphasized. In this context, we also show that a simple heuristic adjustment to VML optimization leads to substantial robustness improvements. Specifically, we follow the same VML optimization with reweighted MAP objective ($\lambda=\sigma_{\min}^2$) until the diffusion noise reaches $\sigma_t = 2*\sigma_{\mathbf{y}}$, and then continue with only one inner-loop iteration ($K=1$) and a reduced learning rate. Results for half-mask inpainting are shown below:
> > >
> > > | σ_y |    $\ \ \ \ \ \ \ \ \ \ \ \ \ $ DAPS |  |   $\ \ \ \ \ \ \ \ \ \ \ \ \ $ VML-MAP  |  |
> > > |:---:|:--------:|:--------:|:-------:|:--------:|
> > > |     | LPIPS ↓  | FID ↓    | LPIPS ↓ | FID ↓ |
> > > | 0.01 | 0.385 | 99.33 | 0.270  | 75.34 |
> > > | 0.02 | 0.388 | 101.2 | 0.278 | 78.23 |
> > > | 0.04 | 0.404 | 109.0 | 0.280 | 82.82 |
> > > | 0.05 | 0.415 | 113.8 | 0.280 | 84.70 |
> > >
> > > The improved results show that the VML framework is inherently more capable. We reiterate our message from Section 7 again: In VML, while SOTA results can be achieved with such heuristic adjustments or special modeling of the noisy case (similar to other methods), a more principled approach involves advanced optimization strategies, which we leave for future work, as it requires a substantial study of its own.
> > >
> > > $\textbf{2. On perceptual vs non-perceptual metrics:}$
> > >
> > > The review dismisses our discussion of metrics without justification. The literature strongly supports our claim: non-perceptual metrics such as PSNR and SSIM frequently diverge from perceptual quality, and recent ICML work [2] highlights the same trade-offs we observe. In image restoration tasks where perceptual fidelity is central, non-perceptual metrics can be misleading. Baselines such as DAPS, DDRM, PiGDM, and MAP-GA frequently achieve higher non-perceptual scores but produce noticeably inferior samples. In contrast, VML-MAP is designed for optimal perceptual fidelity, and the qualitative results shown throughout the paper clearly demonstrate this advantage.
> > >
> > >
> > > We are happy to provide further clarification wherever needed.
> > >
> > >
> > > [1] Solving Inverse Problems with Latent Diffusion Models via Hard Data Consistency (ICLR 2024)
> > >
> > >
> > > [2] Achituve, Idan, et al. "Inverse problem sampling in latent space using sequential Monte Carlo." ICML 2025.

---

### Official Review · Reviewer_8Wgc · 2025-10-31

**Soundness:** 4
**Presentation:** 3
**Contribution:** 2
**Rating:** 4
**Confidence:** 4

**Summary:**

This paper proposes a novel inference-time guidance strategy called Variational Mode-Seeking Loss for solving inverse problems using pre-trained unconditional diffusion models. The core idea is to minimize the reverse KL divergence at each reverse diffusion step, which encourages the intermediate sample $\mathbf{x}_t$ to converge toward the Maximum a Posteriori estimate as $t \to 0$. The authors derive a closed-form expression for VML in the case of linear inverse problems and propose a simplified version $VML_S$ by omitting higher-order covariance terms. They also introduce a preconditioner to handle ill-conditioned linear operators. Extensive experiments on image restoration tasks (inpainting, super-resolution, deblurring) across multiple datasets demonstrate that and its preconditioned variant outperform existing methods in terms of LPIPS and FID metrics, often with lower computational cost.

**Strengths:**

- **Originality:** The formulation of VML as a mode-seeking loss is novel and well-motivated from a variational perspective.
- **Empirical Validation:** Extensive experiments on multiple tasks and datasets show consistent improvements in both perceptual quality (LPIPS) and sample fidelity (FID).
- **Generality:** The method is extended to latent diffusion models, showing promise beyond pixel-space models.

**Weaknesses:**

- **Complexity:** Although the simplified VML is used, the gradient computation still requires Jacobians of the denoiser, which can be computationally expensive, especially for high-resolution images or complex degradation operators.
- **Limited Non-Linear Extension:** The extension to non-linear inverse problems (via LDMs) is preliminary and suffers from optimization challenges due to the non-linearity of the decoder.

**Questions:**

1. In Equation (8), the term $(1 - \Sigma^+ \Sigma)$ uses "1" instead of the identity matrix $\mathbf{I}$. Is this a typo, or is it meant to be a scalar or broadcasted operation? Clarification is needed.
2. The optimization of $D_{\text{KL}}(p(\mathbf{x}_0|\mathbf{x}_t) \| p(\mathbf{x}_0|\mathbf{y}))$ is non-convex and may not guarantee convergence to the true MAP. How does the authors' gradient-based optimizer handle local minima, especially in early reverse steps?
3. The preconditioner requires SVD of $\mathbf{H}$, which may be infeasible for very large or non-linear operators. Are there scalable alternatives for such cases?

---

> ### Author Response · Authors · 2025-11-20
> **Official Response to Reviewer 8Wgc**
>
> We thank the reviewer for their valuable feedback and comments. Below, we addressed the posed questions and concerns.
>
>
> $\textbf{W1 Complexity:}$
>
> We would like to clarify that the gradient computations of VML (and also the preconditioned variant) do not require explicit Jacobian computation but only need the vector-Jacobian products (VJP), which modern autodiff frameworks handle efficiently in practice. This requirement is consistent with the majority of related works (eg, DPS, PiGDM, MAP-GA, DMPlug, etc.). We generally agree that methods requiring VJPs can typically be more computationally expensive than those that do not require VJPs or even a differentiable degradation operator. However, in such cases, a more informative comparison is the overall performance-computation tradeoff. For this purpose, we refer to Figure 7 from the paper, which reveals the performance-computation tradeoff of different methods, highlighting that VML-MAP achieves significantly better performance than other methods for a similar/lower compute time.
>
> $\textbf{W2 Limited Non-Linear Extension:}$
>
> We would like to clarify that the difficulty associated with non-linear inverse problems (via LDMs) is not specific to VML, but rather inherent to the LDM framework itself: any linear inverse problem becomes non-linear once passed through the decoder. As a result, optimization challenges arise for most methods, including ours. For the VML extension to LDMs, these challenges can be amplified when the degradation operator is ill-conditioned, since it compounds with the non-linearity of the decoder. However, if the operator is reasonably well-conditioned (e.g., inpainting), Latent VML-MAP remains competitive with other methods. We also emphasize that our approach uses only the LDM decoder. Other methods, such as Resample and PSLD, additionally rely on the LDM encoder or optimize in both latent and pixel spaces, highlighting that LDM cases need special treatment. While such heuristics can improve performance, exploring them is beyond the scope of this work and is left for future research.
>
> Note that the optimization of VML can be challenging, especially for ill-conditioned or non-linear degradation operators, or both (eg, in the case of LDMs). Higher-order optimization techniques (eg, Newton or quasi-Newton methods) could certainly help, but memory-efficient implementations typically require iterative schemes, eg, Conjugate-gradient or Generalized minimal residual (GM-RES) or L-BFGS methods, which adds to the computational complexity, especially when optimization has to be performed at each diffusion step. Therefore, designing efficient optimizers that are also computationally feasible remains an important open problem. For completeness, we refer to Sec. 7 of our paper.
>
>
> $\textbf{Q1:}$  We thank the reviewer for pointing this out. The “1” in Eqn 8 was a typo and should denote the identity matrix rather than a scalar or broadcast operation. This has been corrected in the latest version.
>
> $\textbf{Q2:}$ Yes, as with any non-convex optimization problem, a practical optimizer cannot guarantee recovery of the global minimizer, and this is a fundamental practical limitation. In VML-MAP, the optimization of $\mathrm{D}\_{KL}(p(x\_0|x_t)||p(x\_0|y))$ is performed at each reverse diffusion step $t$ where the optimizer returns a local solution $x^{*}\_t$, which is used to initialize the next step (see line 228 in Algorithm 1). Because $p(x\_0|x\_t=\gamma)$ converges to $\delta(x\_0-\gamma)$ as $t \to 0$, these iterates converge to a local solution (i.e., a mode of the posterior) rather than the global MAP. See Sec. 3 (Lines 158-160) of the paper.
>
> Since we use a gradient descent optimizer, its convergence behavior depends on the initialization $x_T$ at the start of reverse diffusion. This was analyzed in Appendix C.4 (see Lines 1140-1145), where we evaluated VML-MAP under different random seeds. These experiments show that quantitative results are largely unaffected by initialization (see Tables 5 and 6), while qualitative variations do appear (see Fig. 15)
>
> $\textbf{Q3:}$ The non-preconditioned variant (VML-MAP) does not require any SVD and is a viable option in most cases. Our proposed preconditioner applies to linear operators with affordable SVD, and may be infeasible in other cases. However, note that preconditioners are not unique; different designs are possible, depending on the operator and the availability of SVD. Our proposed preconditioner is only one such design.
>
> In Tab. 2, Vanilla VML-MAP (non-preconditioned, no SVD) outperforms DAPS (no SVD) on nearly all restoration tasks. In Tab. 3, for inpainting tasks, where the preconditioner $\mathrm{P} = \mathbf{I}$, and  VML-MAP, VML-MAP$\_{pre}$​ are essentially equivalent, VML-MAP still significantly outperforms all other methods. This shows that while preconditioning can further improve performance, VML-MAP is already more effective than other methods even when SVD is unavailable.

---

### Official Review · Reviewer_M7Xj · 2025-11-01

**Soundness:** 3
**Presentation:** 3
**Contribution:** 3
**Rating:** 4
**Confidence:** 4

**Summary:**

This paper proposes a variational mode seeking loss (VML) as a principled guidance mechanism for solving inverse problems using unconditional diffusion models. The VML objective, defined as the KL divergence between $p(x_0 | x_t)$ and the measurement posterior is minimized at each reverse diffusion step to steer samples toward the MAP estimate.

**Strengths:**

Main strengths of the paper:

- the paper is well written, easy to read from a technical perspective.
- mathematical details are provided and cleanly described
- a fair bit of experimental results are provided - which demonstrates the method across many scenarios

**Weaknesses:**

I believe the paper misses some relevant comparisons as their method is also somewhat a second-order method (at least the optimal solutions in special cases would be so). So comparisons to some relevant methods (see below) would improve the work.

I think the main weakness is that it is not clear why VML would be preferable to standard methods. Given that there are several approximations in the way the VML is used in practice, I think the claim that this is more principled is a bit weak. I'd encourage authors to both provide practical and mathematical justification of VML compared to existing works.

**Questions:**

- In the motivation part, I think it wasn't made clear what the *practical* motivation of this work is.

- Please define the argument of the loss when you define VML. What is the main "input variable" of VML?

- The authors say *we hypothesize that these higher-order terms may not be crucial in practice*, does this not reduce your approach to standard guidance approaches?

- The authors derive in Props 1 & 2, their VML. As noted by the authors, this takes the form that is similar to the usual guidance. Can the authors compare their approach mathematically to second-order guidance approaches such as Boys et al (2024) or other moment-matching based methods. Especially the approach based on preconditioners.

- Similarly, given the relationship, I'd also expect TMPD (Boys et al, 2024) or similar approaches to be included in the experimental comparisons.

- Is the algorithm more robust to noise compared to standard approaches? Perhaps an experiment with increasing noise levels would be helpful here. This is a bit like your Figure 7 but x axis would be increasing noise levels.

- Is there any scope or reason to use anything else than KL?

Style comment: I think having blue text in the main text is not standard and appropriate. I wonder if these were meant to highlight this is a previous submission. In any case, I'd ask authors to get rid of text colouring (except for instructive cases like equation highlighting etc).

---

> ### Author Response · Authors · 2025-11-20
> **Official Response to Reviewer M7Xj (1/3)**
>
> We thank the reviewer for their valuable feedback. We answered the posed questions below and would be happy to provide further clarifications.
>
> $\textbf{Styling:}$ We apologize for the misunderstanding and would like to clarify that it is not a previous submission. The blue-colored text was added intentionally to emphasize certain parts of the text to the reader, which we now removed, following the suggestion.
>
> $\textbf{Q1 to Q5:}$
>
> **TMPD is a second-order, score-based, posterior sampling method, and makes several restrictive approximations, unlike VML**
>
> We respectfully clarify that VML differs fundamentally from TMPD and other guidance approaches, both in its theory and method. TMPD is a posterior sampling method that approximates the conditional score $\nabla\_{\mathbf{x}\_t} \log p(\mathbf{x}_t|\mathbf{y})$ using strong modeling assumptions (i.e., a Gaussian approximation of $p(\mathbf{x}\_0|\mathbf{x}\_t)$), covariance simplifications, etc. In contrast, VML is an optimization-based MAP estimation framework derived from a principled KL objective, requiring no Gaussian or score-modeling assumptions, and no covariance modeling, unlike TMPD. Note that TMPD aims to approximate the conditional score, while VML does not deal with the conditional scores at all.
>
>
> TMPD assumes $p(\mathbf{x}\_0|\mathbf{x}\_t) \approx \mathcal{N}(\mathrm{D} (\mathbf{x}\_t,t), \mathrm{Cov}[\mathbf{x}\_0|\mathbf{x}\_t])$, where $\mathrm{D}(\mathbf{x}\_t,t)$ is the denoiser. This is a very restrictive assumption, as it leads to a Gaussian treatment of $p(\mathbf{x}\_0)$.
>
> With the above, TMPD derives:
> $\nabla_{\mathbf{x}\_t} \log p(\mathbf{x}\_t|\mathbf{y}) \approx \frac{\partial \mathrm{D}^{\top}(\mathbf{x}\_t,t)}{\partial \mathbf{x}\_t} \mathrm{H}^{\top} \left( \mathrm{H} \mathrm{Cov}[\mathbf{x}\_0|\mathbf{x}\_t] \mathrm{H}^{\top} + \sigma^2\_{\mathbf{y}}\mathbf{I} \right)^{-1} \left( \mathbf{y} - \mathrm{H}\mathrm{D}(\mathbf{x}\_t,t) \right) + \frac{\mathrm{D}(\mathbf{x}\_t,t) - \mathbf{x}\_t}{\sigma^2\_t}$
>
> by making several modeling and empirical approximations (eg, treating $\mathrm{Cov}[\mathbf{x}\_0|\mathbf{x}\_t]$ independent of $\mathbf{x}\_t$ and estimating only the diagonal by further approximating with a heuristic row-sum of Jacobian, etc.)
>
> **Motivation of MAP estimation vs Posterior sampling methods**
>
> It is perhaps more accurate to evaluate posterior sampling schemes based on their capacity to cover the true posterior, but this has not been the focus in the literature and understandably so, due to the limited availability of ground truth posterior samples. Also, many inverse tasks (eg, image restoration) often require generating a plausible image consistent with the measurement, fueling the current evaluation criteria towards plausibility of estimates rather than covering the posterior. This is inconsistent with the principle of posterior sampling and aligns more closely with MAP estimation, since a MAP estimate is the most plausible (probable) posterior sample. This has been the main motivation for MAP estimation approaches, including VML.
>
> **VML is a first-order, optimization-based, MAP estimation method, and does not make restrictive modeling approximations, unlike TMPD**
>
> Our work introduces the VML objective at time $t$ i.e., $ \mathrm{VML}\_{t}(\mathbf{x}\_t,t) = \mathrm{D}\_{KL}(p(\mathbf{x}\_0|\mathbf{x}\_t)||p(\mathbf{x}\_0|\mathbf{y}))$, and show that optimizing it for $\mathbf{x}^{\*}\_t$ steers these towards the MAP estimate as $t \to 0$. We prove under mild assumptions, $\mathrm{VML}\_t(\cdot, t)$ converges to the negative log posterior (i.e., the MAP objective) as $t \to 0$. We observe that $\mathrm{VML}\_{t}(\mathbf{x}\_t,t)$ has certain higher-order terms involving $\mathrm{Cov}[\mathbf{x}\_0|\mathbf{x}\_t]$ whose gradient computation is infeasible in practice. However, we show that these terms converge to zero as $t\to 0$, implying VML and the simplified-VML converge to the same function as $t\to 0$.
>
> Note that if VML and simplified-VML are well-behaved and have certain convergence criteria, their global minimizers could also converge to the same value, i.e., the MAP estimate. Proving this formally is beyond the scope of our work, however, we argue for the same from a practical perspective: since we do not have a perfect optimizer in practice, at higher time $t$, even with the full VML objective, the optimizer cannot guarantee finding the global minimizer, and as $t \to 0$, the simplified-VML gets closer to full VML and so we remark that it may not be crucial to ignore the higher-order loss terms either way.

---

> ### Author Response · Authors · 2025-11-20
> **Official Response to Reviewer M7Xj (2/3)**
>
> The negative gradient of the simplified VML, i.e.,
> $-\nabla\_{\mathbf{x}\_t} \mathrm{VML}\_{\mathrm{S}} = \underbrace{ \frac{\partial \mathrm{D}^{\top}(\mathbf{x}\_t,t)}{\partial \mathbf{x}\_t} \frac{\mathrm{H}^{\top}(\mathbf{y}-\mathrm{H}\mathrm{D}(\mathbf{x}\_t,t))}{\sigma^2\_\mathbf{y}} }\_{\textit{measurement consistency gradient}} \underbrace{ + \frac{\partial \mathrm{D}^{\top}(\mathbf{x}\_t,t)}{\partial \mathbf{x}\_t} \frac{(\mathrm{D}(\mathbf{x}\_t,t)-\mathbf{x}\_t)}{\sigma^2\_t} }\_{\textit{prior gradient}}$
>
> Please note the different formulations of TMPD and VML above, and recall that TMPD is score-based (i.e., single evaluation/use of the conditional score per diffusion step), while VML is optimization-based (i.e., solves an optimization problem with multiple gradient updates per diffusion step). Also note that both VML-MAP and the pre-conditioned variant are first-order, and do not require estimation of $\mathrm{Cov}[\mathbf{x}_0|\mathrm{x}_t]$, unlike TMPD.
>
> We further provide an empirical comparison of VML-MAP vs TMPD below.
>
>  **Table 1.** Half-mask inpainting on 100 validation images of ImageNet256.
> Mean ± std across 4 seeds is reported. Best values are **bold**, second-best are $\underline{underlined}$.
>
> | Dataset   | Method     |       LPIPS ↓       |       FID ↓       |
> |------------|-------------|------------------|-----------------|
> |            | TMPD     | 0.384 ± $\color{lightgray}{0.004}$    | 138.9 ± $\color{lightgray}{1.532}$   |
> |   ImageNet   | VML-MAP | **0.262** ± $\color{lightgray}{0.002}$| **74.21** ± $\color{lightgray}{3.209}$|
> |            | VML-MAP$\_{pre}$ | **0.262** ± $\color{lightgray}{0.002}$|**74.21** ± $\color{lightgray}{3.209}$|
>
> **Table 2.** 4$\times$ super-resolution on 100 validation images of ImageNet256.
> Mean ± std across 4 seeds is reported. Best values are **bold**, second-best are $\underline{underlined}$.
>
> | Dataset   | Method     |       LPIPS ↓       |       FID ↓       |
> |------------|-------------|------------------|-----------------|
> |            | TMPD     | 0.230 ± $\color{lightgray}{0.002}$    |  77.56 ± $\color{lightgray}{0.502}$   |
> |   ImageNet   | VML-MAP | **0.194** ± $\color{lightgray}{0.001} $ | $\underline{60.08}$ ± $\color{lightgray}{0.687}$|
> |            | VML-MAP$\_{pre}$ | $\underline{0.196}$ ± $\color{lightgray}{0.003} $ | **58.60** ± $\color{lightgray}{2.076}$|
>
> **Table 3.** Uniform deblurring on 100 validation images of ImageNet256.
> Mean ± std across 4 seeds is reported. Best values are **bold**, second-best are $\underline{underlined}$.
>
> | Dataset   | Method     |       LPIPS ↓       |       FID ↓       |
> |------------|-------------|------------------|-----------------|
> |            | TMPD     | 0.534 ± $\color{lightgray}{0.004}$    |  245.8 ± $\color{lightgray}{3.422}$   |
> |   ImageNet   | VML-MAP | $\underline{0.509}$ ± $\color{lightgray}{0.003} $ | $\underline{200.4}$ ± $\color{lightgray}{2.812}$|
> |            | VML-MAP$\_{pre}$ | **0.367** ± $\color{lightgray}{0.002} $ | **165.2** ± $\color{lightgray}{2.037}$|

---

> ### Author Response · Authors · 2025-11-20
> **Official Response to Reviewer M7Xj (3/3)**
>
> $\textbf{Q6: Robustness to noise}$
>
> We agree that this is an important point. Note that the VML objective converges to $-\log p(\mathbf{x}\_0|\mathbf{y}) = -\log p(\mathbf{y}|\mathbf{x}\_0) - \log p(\mathbf{x}\_0) + \text{Constant} $. In its original formulation, we find that the VML objective is sensitive to measurement noise. However, we observe that the reweighted MAP objective (downweighting the prior with $\lambda \ll 1$, i.e., $-\log p(\mathbf{y}|\mathbf{x}\_0) - \lambda \log p(\mathbf{x}\_0) $ adds to the numerical stability of the algorithm and robustness to measurement noise. Note that the reweighted objective translates to reweighting the measurement consistency and the prior terms of VML (Proof trivially follows from Proposition 4). While $\lambda$ can also be learned, we treat it as a hyperparameter and fix $\lambda = \sigma^2\_{min}$ in our experiments below.
>
> **Table 4: Robustness to measurement noise. Half-mask Inpainting (ImageNet256)**
>
> | σ_y |     $\ \ \ \ \ \ \ \ \ \ \ \ \ \ $ΠGDM |  |     $\ \ \ \ \ \ \ \ \ \ \ \ \ \ $DAPS |  |      $\ \ \ \ \ \ \ \ \ \ \ \ \ \ $VML-MAP |  |
> |:---:|:--------:|:--------:|:-------:|:--------:|:-----------:|:--------:|
> |     | LPIPS ↓  | FID ↓    | LPIPS ↓ | FID ↓    | LPIPS ↓     | FID ↓    |
> | 0.00 | 0.373 | 103.8 | 0.385 | 98.72 | 0.265 | 69.21 |
> | 0.01 | 0.373 | 107.6 | 0.385 | 99.33 | 0.262 | 71.68 |
> | 0.02 | 0.376 | 113.4 | 0.388 | 101.2 | 0.278 | 81.59 |
> | 0.04 | 0.383 | 124.9 | 0.404 | 109.0 | 0.315 | 92.71 |
> | 0.05 | 0.390 | 126.4 | 0.415 | 113.8 | 0.347 | 101.3 |
>
>
> **Table 5: Robustness to measurement noise. 4× Super-resolution (ImageNet256)**
>
> | σ_y | $\ \ \ \ \ \ \ \ \ \ \ \ \ \ $ΠGDM |  | $\ \ \ \ \ \ \ \ \ \ \ \ \ \ $DAPS |  | $\ \ \ \ \ \ \ \ \ \ \ \ \ \ $VML-MAP |  |
> |:---:|:--------:|:--------:|:-------:|:--------:|:-----------:|:--------:|
> |     | LPIPS ↓  | FID ↓    | LPIPS ↓ | FID ↓    | LPIPS ↓     | FID ↓    |
> | 0.00 | 0.292 | 83.80 | 0.256 | 73.33 | 0.192 | 59.91 |
> | 0.01 | 0.306 | 100.2 | 0.256 | 72.88 | 0.204 | 61.60 |
> | 0.02 | 0.324 | 114.1 | 0.259 | 73.04 | 0.241 | 74.76 |
> | 0.04 | 0.356 | 139.0 | 0.290 | 81.24 | 0.331 | 103.1 |
> | 0.05 | 0.371 | 147.3 | 0.345 | 97.17 | 0.373 | 113.2 |
>
> Experiments show that VML-MAP is robust up to moderate levels of measurement noise ($\sigma_{\mathbf{y}} \leq 0.02$), after which the performance begins to degrade. Note that other methods exhibit a similar trend, though the effect is less pronounced. For VML, this behavior stems from the sensitivity of its gradient term to $\sigma_{\mathbf{y}}$. While heuristic adjustments (eg, DAPS sets $\sigma_{min}=0.1$ to make their method more robust to noise. See under hyperparameters overview in Appendix F.2 of DAPS paper) could mitigate this issue, a more principled and effective solution would involve higher-order or advanced optimization techniques. This is beyond the current scope and is left for future work.
>
> $\textbf{Q7: Scope to use other losses than KL}$
>
> Great question, in fact, while KL loss makes for a simplified derivation, the general framework is not restricted to it, i.e., any loss/divergence measure that encourages/captures a mode-seeking behavior of distributions can be used. However, changing the loss or divergence might require an adaptation of the optimization scheme in practice, and therefore, we leave it to future work. Finally, note that the simplified-VML objective proposed in the paper precisely does such an update of the loss function.

---

### Official Review · Reviewer_JeSW · 2025-11-10

**Soundness:** 2
**Presentation:** 3
**Contribution:** 2
**Rating:** 6
**Confidence:** 3

**Summary:**

This paper presents a new inference-time optimization method, called Variational Mode-Seeking Loss (VML), for solving inverse problems using pre-trained unconditional diffusion models. Unlike prior approaches that approximate the conditional score or solve complex ODEs for posterior sampling or MAP estimation, this work formulates a loss function derived from minimizing the Kullback–Leibler divergence between the diffusion posterior $p(x_0 \mid x_t)$ and the measurement posterior $p(x_0 \mid y)$.

The authors show that, for linear inverse problems, this loss can be expressed in closed form, avoiding the need for approximations. Based on this, they propose VML-MAP, an algorithm that iteratively minimizes the loss during each reverse diffusion step, guiding samples toward the MAP estimate. They further propose a preconditioned variant to address optimization instability in ill-conditioned problems. Extensive experiments on several image restoration tasks—such as inpainting, super-resolution, and debluring—demonstrate improved perceptual quality and competitive computational efficiency compared to methods like DDRM, ΠGDM, and MAPGA.

**Strengths:**

1. The derivation of the variational mode-seeking loss is mathematically rigorous and well connected to Bayesian reasoning. The paper provides clear intuition for why minimizing the reverse KL divergence aligns the diffusion trajectory with the posterior mode.
2. By deriving an analytical expression for VML under linear degradations, the paper avoids the heuristic approximations common in posterior sampling methods. This provides both conceptual clarity and computational efficiency.
3. The evaluation spans multiple datasets and degradation operators. The results consistently show that VML-MAP and its preconditioned variant outperform or match strong baselines across tasks, confirming robustness.

**Weaknesses:**

The paper is well executed but the conceptual novelty feels moderate. The mode-seeking idea is mainly a reformulation of MAP estimation within the diffusion framework. While mathematically elegant, it is not entirely clear what fundamentally distinguishes VML from existing guidance-based or posterior sampling methods beyond its derivation.

1. The practical improvements over prior MAP-based solvers are relatively small in some tasks, and the results mostly show incremental gains in LPIPS or FID rather than large performance leaps. The paper could analyze why the VML-based formulation improves results—whether due to better gradient alignment, reduced variance, or implicit regularization.

2. The derivations and proofs rely heavily on the assumption of a linear degradation operator and Gaussian noise. It is unclear how VML could generalize to nonlinear or learned operators, which are becoming increasingly common in inverse problems. The brief mention of extending to latent diffusion models acknowledges this, but the results there remain limited.

Overall, the work is technically impressive and experimentally solid, but it would benefit from stronger conceptual clarity on how the proposed formulation fundamentally advances beyond reweighted MAP estimation with standard priors.

**Questions:**

1. Can the proposed method be interpreted as an implicit form of score correction or gradient projection? If so, is there a way to visualize or quantify how the diffusion trajectory differs from existing guided diffusion methods?
2. The derivation assumes a linear operator H and Gaussian measurement noise. Could you discuss how the approach might extend to nonlinear or non-Gaussian settings?
3. The VML objective relies on minimizing $\mathrm{KL}(p(x_0\mid x_t) \| p(x_0\mid y))$. Why is the reverse KL chosen instead of the forward direction, and how does this affect the sampling trajectory compared to posterior averaging methods?

---

> ### Author Response · Authors · 2025-11-20
> **Official Response to Reviewer JeSW (1/2)**
>
> We thank the reviewer for their valuable comments. We answered the posed questions below and would be happy to provide further clarifications.
>
> $\textbf{Q1:}$ MAP estimation methods (including ours), in contrast to most guidance-based or posterior sampling methods (eg, DPS, PiGDM, TMPD), fundamentally do not estimate the guidance score or the conditional score and are methodologically different with an entirely different theoretical objective. While it may be possible to interpret VML as score correction under very special cases, in general, VML fundamentally differs from score-based approaches. Please see the discussion below under “conceptual novelty of VML” that highlights the distinctions further.
>
> $\textbf{Q2:}$ Non-linear operators and non-Gaussian settings would require approximations for the full VML objective. However, an equivalent simplified-VML converging to the same function as the full VML may be possible in non-linear settings, just like for the linear setting we showed in the paper. Importantly, the VML extension to non-linear operators (via LDMs) has already been explored in Appendix D. Note that it is trivial to use the same extension with pixel diffusion models (see Lines 451-456). We would like to emphasize that, unlike VML, existing MAP solvers do not provide theoretical justification even in the linear setting.
>
> $\textbf{Q3:}$ The distribution $p(\mathbf{x}\_0|\mathbf{x}\_t)$ gets peaky as $t \to 0$. Use of forward KL would steer the sample towards the posterior mean (the well-known moment matching behavior), which is undesirable, as the posterior mean may not have a high density or even lie on the data manifold. We chose reverse KL so that the samples are steered towards the MAP estimate (or modes in practice), which is our main objective (i.e., MAP estimation). Please see Section 3, which explains the case. Also, note that our proposed framework is not restricted to using the reverse KL loss. While it makes for a simplified derivation, other losses/divergence measures that encourage mode-seeking behavior of distributions can also be used.
>
> $\textbf{W1:}$
>
> $\textbf{Incremental improvements:}$
> Note that we intentionally chose very challenging inverse tasks in the paper (see Appendix C.1), and VML still achieves significant improvements compared to all other methods in several cases (see Super-resolution and Deblurring cases in Tab.2). MAP-GA, in a few cases (especially inpainting tasks), is the second-best and close, but note that LPIPS $\in [0,1]$, and even a relatively smaller numerical improvement in LPIPS is significant (see qualitative examples in Appendix C.3). Also, most importantly, VML is ~3.5x faster than MAP-GA (See Fig.7).
>
> $\textbf{Why VML formulation improves results:}$
> We believe the efficacy of the VML-based formulation arises from the following: 1. The main goal is to find the MAP estimate (unlike guidance-based or posterior sampling schemes that do not explicitly aim for it) 2. No modeling approximations of VML in theory (unlike other MAP solvers). Please see more details under “Conceptual novelty of VML” below, which further distinguishes VML from existing methods and justifies why VML could be more effective.
>
>
> $\textbf{W2:}$
> $\textbf{Generalization to non-linear operators:}$
> Note that, as long as the degradation operator is differentiable, VML can be generalized to non-linear or learned operators via approximations similar to our proposed non-linear extension (via LDMs) in Appendix D.
>
> We would like to clarify that the difficulty associated with non-linear inverse problems (via LDMs) is not specific to VML, but rather inherent to the LDM framework itself: any linear inverse problem becomes non-linear once passed through the decoder. As a result, optimization challenges arise for most methods, including ours. For the VML extension to LDMs, these challenges can be amplified when the degradation operator is ill-conditioned, since it compounds with the non-linearity of the decoder. However, if the operator is reasonably well-conditioned (e.g., inpainting), Latent VML-MAP performs better than existing methods. We also emphasize that our approach uses only the LDM decoder, while other methods (eg, Resample, PSLD) additionally rely on the LDM encoder or optimize in both latent and pixel spaces, highlighting that LDM cases need special treatment. While such heuristics can improve performance, exploring them is beyond the scope of this work and is left for future research.
>
>
> Also note that better optimization techniques with the same VML formulation have the potential to improve performance significantly for non-linear operators, just as the preconditioner improves the performance for linear tasks. See Section 7.

---

> ### Author Response · Authors · 2025-11-20
> **Official Response to Reviewer JeSW (2/2)**
>
> $\textbf{Conceptual novelty of VML vs existing methods:}$
>
> Posterior sampling methods aim to sample from the posterior $p(\mathbf{x}\_0|\mathbf{y})$. In diffusion models, samples generated by the reverse SDE: $\mathrm{d} \mathbf{x}\_t = \\{ \mathbf{f}(\mathbf{x}\_t, t) - g^2(t) \nabla\_{\mathbf{x}\_t} \log p(\mathbf{x}\_t|\mathbf{y}) \\} \mathrm{d} t + g(t) \mathrm{d} \bar{\mathbf{w}}\_t$ covers the true posterior. However, having only the unconditional score  $\nabla\_{\mathbf{x}\_t} \log p(\mathbf{x}\_t)$, the conditional score $\nabla\_{\mathbf{x}\_t} \log p(\mathbf{x}\_t|\mathbf{y})$ is hard to estimate and is intractable. So several works propose approximations of the conditional score (eg, DPS, PiGDM, TMPD, etc.). Note that other works (eg, DAPS) propose approaches that circumvent the need to estimate this conditional score. However, the main objective of all posterior sampling methods remains the same, which is to sample from (i.e., cover) the posterior.
>
> MAP estimation approaches aim to find the MAP estimate (i.e., $\arg \max\_{\mathbf{x}\_0}p(\mathbf{x}\_0|\mathbf{y})$), and fundamentally differ in theory and method compared to posterior sampling schemes. As the MAP estimate is a point estimate, MAP solvers do not deal with the conditional score function (like posterior samplers), but rather find different approaches to optimize the MAP objective.
>
> Earlier works on MAP estimation, like DiffPIR, adapt variable splitting optimization algorithms such as HQS or ADMM (which relaxes the original MAP objective into several proximal subproblems) to the diffusion framework. Later, more efficient works, such as DMPlug and MAP-GA, considered the direct mapping from noise to data given by the PFODE or the consistency model and solved for the optimal noise using gradient-based optimizers on the reparameterized MAP objective. Though free from modeling approximations, these approaches require a consistency model. So, multiple diffusion denoiser steps are used to approximate the full PFODE solution, resulting in expensive gradient computations.
>
> Our paper proposes a different MAP estimation approach than the above methods. We introduce the VML objective at time $t$ i.e., $ \mathrm{VML}\_{t}(\mathbf{x}\_t,t) = \mathrm{D}\_{KL}(p(\mathbf{x}\_0|\mathbf{x}\_t)||p(\mathbf{x}\_0|\mathbf{y}))$, and show that optimizing this sequence of functions $\mathrm{VML}\_t(\cdot,\cdot)$ at each time for optimal $\mathbf{x}^{*}\_t$ will steer these towards the MAP estimate as $t \to 0$. Under mild assumptions, we prove $\mathrm{VML}\_t(\cdot, t)$ actually converges to the negative log posterior (i.e., the MAP objective) as $t \to 0$.
>
> Our approach differs from existing MAP estimation methods, and certainly from posterior sampling methods (for we do not deal with the conditional score or guidance scores at all). To the best of our knowledge, VML is the first approximation-free modeling of MAP estimation with formal analysis for the linear operator setting, in the context of solving inverse problems using only the pre-trained diffusion model. Note that VML has empirical approximations when we use the simplified-VML in practice; however, it is based on a thorough theoretical justification (see Appendix B).

---

### Meta-Review · Area_Chair_LdeE · 2026-01-06

**Summary:**

This paper proposes a variational mode-seeking loss (VML) for inference-time MAP estimation with pre-trained diffusion models. The key idea is to minimize a reverse-KL objective between the diffusion posterior p(x0|xt) and the measurement posterior p(x0|y) at each reverse step, yielding a practical algorithm (VML-MAP) and a preconditioned variant for ill-conditioned linear operators. Reviewers generally agree the derivations for linear inverse problems are clean and the empirical results are broad, but the overall reception is mixed because (i) the conceptual novelty over existing MAP-style solvers / guidance-like objectives is debated, (ii) the method still relies on practical approximations and non-convex optimization whose behavior is not fully characterized, and (iii) the evaluation choices (metrics and noise regimes) were initially seen as incomplete by the most critical reviewer. Overall: interesting work, but not clearly above the bar given the current scores and remaining disagreement.

**Reviewer Concerns:**

Concerns that were addressed by the rebuttal:
1) Missing comparisons and robustness-to-noise questions: the authors added comparisons against TMPD and provided additional noisy-setting results with a reweighted objective, which directly responds to requests for noise robustness and relevant baselines.
2) Complexity/Jacobian concern: the authors clarified they rely on vector-Jacobian products rather than explicit Jacobian construction, and argued the runtime–quality tradeoff is favorable.
3) Presentation issues: the blue-colored text and a typo around the identity matrix were acknowledged and corrected.

Concerns still outstanding:
1) Novelty remains borderline for part of the committee. Even if the reverse-KL framing is principled, several reviewers still view the practical algorithm as close to existing “optimize a MAP-like objective during sampling” approaches, with differences more in formulation than in a clearly new capability.
2) Evaluation remains somewhat contentious. The paper makes a reasonable case for perceptual metrics, but one reviewer’s request for standard fidelity metrics and broader noisy settings is only partially satisfied, and the decision still depends on what the conference wants to prioritize for inverse problems.
3) The method’s behavior is tied to non-convex optimization per step; the rebuttal acknowledges this and frames it as a general limitation, but it remains a real uncertainty for reliability and generality (especially beyond linear operators / beyond the specific settings tested).

**Reviewer Scores:**

1) Reviewer JeSW (score 6): likely unchanged at 6.
2) Reviewer M7Xj (score 4): likely increase to 5.
3) Reviewer 8Wgc (score 4): likely unchanged at 4.
4) Reviewer 2tpR (score 2): likely unchanged at 2.

---

### Decision · Program_Chairs · 2026-01-26

Reject